# Hessian-Free Online Certified Unlearning

**Xinbao Qiao**[1]    **Meng Zhang**[1,*]    **Ming Tang**[2]    **Ermin Wei**[3]
[1]Zhejiang University    [2]Southern University of Science and Technology    [3]Northwestern University
xinbaoqiao@zju.edu.cn

## Abstract

Machine unlearning strives to uphold the data owners' right to be forgotten by enabling models to selectively forget specific data. Recent advances suggest pre-computing and storing statistics extracted from second-order information and implementing unlearning through Newton-style updates. However, the Hessian matrix operations are extremely costly and previous works conduct unlearning for empirical risk minimizer with the convexity assumption, precluding their applicability to high-dimensional over-parameterized models and the nonconvergence condition. In this paper, we propose an efficient Hessian-free unlearning approach. The key idea is to maintain a statistical vector for each training data, computed through affine stochastic recursion of the difference between the retrained and learned models. We prove that our proposed method outperforms the state-of-the-art methods in terms of the unlearning and generalization guarantees, the deletion capacity, and the time/storage complexity, under the same regularity conditions. Through the strategy of recollecting statistics for removing data, we develop an online unlearning algorithm that achieves near-instantaneous data removal, as it requires only vector addition. Experiments demonstrate that our proposed scheme surpasses existing results by orders of magnitude in terms of time/storage costs with millisecond-level unlearning execution, while also enhancing test accuracy.

## 1 Introduction

Recent data protection regulations, e.g. General Data Protection Regulation, aim to safeguard individual privacy. These regulations emphasize the provisions of the *Right to be Forgotten*, which grants individuals the ability to request the removal of personal data. This regulation mandates that the data controller shall erase personal data without undue delay when data providers request to delete it. Within the context of the right to be forgotten, data owners can request the removal of their personal data's contribution from trained models. This can be achieved through the concept of *machine unlearning*, which involves erasing data that pertains to the data owner's personal information.

A straightforward unlearning algorithm is to retrain a learned model from scratch, which is however impractical as it incurs unaffordable costs. It is especially challenging in time-sensitive applications such as fraud detection or online learning, as delayed updates upon receiving a deletion request would impede the system's ability to respond effectively to subsequent inputs, compromising its performance and reliability. Moreover, attaining a model entirely identical to the one achieved through retraining for an unlearning method represents a notably demanding criterion. Therefore, many approximate methods have been proposed aiming to minimize the impact of forgetting data. Inspired by differential privacy, Guo et al. (2020) introduced a relaxed definition of certified unlearning, which aims to make the output distribution of unlearned algorithm indistinguishable from that of the retraining.

Second-order unlearning methods have been recently studied due to their rigorous unlearning certification and the generalization guarantee. Sekhari et al. (2021) initially proposed the Newton update method and introduced the definition of deletion capacity to understand the relationship between generalization performance and the amount of deleted samples. These method and concepts were later extended, e.g. Suriyakumar et al. (2022); Liu et al. (2023); Mahadevan & Mathioudakis (2021); Chien et al. (2022). The key idea of these works is to extract Hessian information of the loss function and achieve computational and memory efficiency by pre-computing and storing these second-order statistics prior to any deletion request. In particular, Sekhari et al. (2021) and Suriyakumar et al.

---

*Corresponding Author

Table 1: **Summary of Results.** Here, $d$, $n$, and $m$ denote the model parameters size, training dataset size, and forgetting dataset size, respectively. The unlearning guarantee is derived under the same regularity conditions as prior second-order unlearning studies, with $\rho$ being a constant less than 1.

| Methods | Computation time | Storage | Unlearning time | Unlearning guarantee |
|---|---|---|---|---|
| Sekhari et al. (2021) | $\mathcal{O}(nd^2)$ | $\mathcal{O}(d^2)$ | $\mathcal{O}(d^3 + md^2 + md)$ | $\mathcal{O}(m^2/n^2)$ |
| Suriyakumar et al. (2022) | $\mathcal{O}(d^3 + nd^2)$ | $\mathcal{O}(d^2)$ | $\mathcal{O}(d^2 + md)$ | $\mathcal{O}(m^2/n^2)$ |
| Proposed method | $\mathcal{O}(n^2d)$ | $\mathcal{O}(nd)$ | $\mathcal{O}(md)$ | $\mathcal{O}(m\rho^n)$ |

(2022) revealed that for models that approximate the minimization of convex and sufficiently smooth empirical minimization objectives, unlearning can be efficiently executed through a Newton style step, facilitated by pre-storing Hessian matrix. These methods and theoretical analyses are subsequently extended to graph learning (Chien et al. (2022)) and minimax learning (Liu et al. (2023)) scenarios.

While these works have demonstrated the effectiveness of the second-order method to approximate retrained models with limited updates, they do have a few fundamental challenges: existing second-order algorithms (i) lack the capability to handle multiple deletion requests in online because they require explicit precomputing Hessian (or its inverse), and (ii) the learned model needs to be empirical risk minimizer, (iii) with assumption of convexity to ensure the invertibility of Hessian. Consequently, these limitations preclude their applicability to high-dimensional models (e.g., over-parameterized neural networks) for computing the Hessian. For non-convex problems (e.g., deep networks), we cannot guarantee the invertibility of the Hessian, and it is non-trivial for a learned model to converge to an empirical risk minimizer. Even for convex problems, the non-stationarity of data (Hazan (2016)) is another common issue in the incremental model update scenarios (e.g., continual learning) where training is sequential but not concurrent, making it challenging for the model to adapt and converge to the optimal solution. Consequently, we are motivated to answer the following key question:

> *(Q) How can efficient second-order unlearning be achieved without incurring the high cost of Hessian operations, and how can effective certified unlearning be implemented under non-convex assumptions for the objectives and non-convergence conditions for the models?*

In this paper, we take the first step towards tackling *(Q)* from an innovative Hessian-free perspective. In contrast to existing schemes that extract second-order information exclusively from the optimal learned model, we leverage details from learning models during training procedure. We analyze the difference between the learning model and the retraining model by recollecting the model update trajectory discrepancy. Subsequently, we demonstrate that the discrepancies can be characterized through an affine stochastic recursion. We summarize our main contributions below and in Table 1.

- **Hessian-Free Model Update Method.** We propose a method to extract the Hessian information by analyzing each sample's impact on the training trajectory without explicitly computing the Hessian matrix. Our Hessian-free method enables efficient handling of deletion requests in an online manner. Moreover, the proposed method does not require that the learned model is empirical risk minimizer, which enjoys better scalability compared to existing second-order methods. (Section 3).

- **Improved Guarantees and Complexity.** Under identical regularity conditions, we analytically show that our proposed method outperforms the state-of-the-art theoretical studies in terms of unlearning guarantee (See Corollary 5), generalization guarantee (See Theorem 6), deletion capacity (See Theorem 7), and computation/storage complexity (See Subsection 4.4). Moreover, the unlearning guarantee can be applied to both convex and non-convex settings. (Section 4).

- **Instantaneous Unlearning Algorithm.** Building upon the aforementioned analysis, we develop a Hessian-free certified unlearning algorithm. To our knowledge, our algorithm is *near-instantaneous* and is currently the most efficient as it requires only a vector addition operation. (Section 4).

- **Extensive Evaluation.** We conduct experimental evaluations using a wider range of metrics compared to previous theoretical second-order studies, and release our open source code.[1] The experimental results verify our theoretical analysis and demonstrate that our proposed approach surpasses previous certified unlearning works. In particular, our algorithm incurs millisecond-level unlearning runtime to forget per sample with minimal performance degradation. (Sections 5 and 6).

---

[1]Our code is available at ⬡ Hessian Free Certified Unlearning

## 1.1 Related Work

Machine unlearning can be traced back to Cao & Yang (2015), which defines "the completeness of forgetting", demanding the models obtained through retraining and unlearning exhibit consistency. **Exact Unlearning**, e.g., Bourtoule et al. (2021); Brophy & Lowd (2021); Schelter et al. (2021); Yan et al. (2022); Chen et al. (2022b;a), typically train the submodels based on different dataset partitions and aggregation strategies (See elaborated exact unlearning background in Appendix A). **Approximate Unlearning**, e.g. Wu et al. (2020); Nguyen et al. (2020); Golatkar et al. (2020a); Izzo et al. (2021); Mehta et al. (2022); Wu et al. (2022); Tanno et al. (2022); Becker & Liebig (2022); Jagielski et al. (2023); Warnecke et al. (2023); Tarun et al. (2023), seeks to minimize the impact of forgetting data to an acceptable level to tradeoff for computational efficiency, reduced storage costs, and flexibility (See elaborated approximate unlearning background in Appendix A). In particular, Guo et al. (2020) introduces a relaxed definition of $(\epsilon, \delta)$-unlearning, in anticipation of ensuring the output distribution of an unlearning algorithm is indistinguishable from that of the retraining. Building upon this framework, diverse methodologies, e.g. Sekhari et al. (2021); Suriyakumar et al. (2022); Gupta et al. (2021); Neel et al. (2021); Liu et al. (2023); Chien et al. (2022), have been devised.

In the certified unlearning definition, the Hessian-based approaches (Sekhari et al. (2021); Suriyakumar et al. (2022); Liu et al. (2023)) are aligned with our methodology, which involves pre-computing data statistics that extract second-order information from the Hessian and using these precomputed "recollections" to facilitate data forgetting. Specifically, (1) **Newton Step (*NS*).** Sekhari et al. (2021) proposed, through the precomputation and storage of the Hessian offline, to achieve forgetting upon the arrival of a forgetting request using a Newton step. However, *NS* is limited by its high computational complexity of $\mathcal{O}(d^3)$, making it impractical for large-scale deep models. Furthermore, it can only perform certified unlearning for empirical minimizers under convexity assumption. (2) **Infinitesimal Jackknife (*IJ*).** Suriyakumar et al. (2022) extended the work of *NS*, attaining strong convexity through a non-smooth regularizer by leveraging the proximal *IJ* with reduced unlearning time $\mathcal{O}(d^2)$. However, a few important challenges still remain, i.e., existing second-order methods still require expensive Hessian computations and perform forgetting for empirical risk minimizer with the convexity assumption. In contrast, our method overcomes the limitations of existing second-order methods by achieving near-instantaneous unlearning and performing forgetting for models in any training step. Hence, our approach is applicable to scenarios where models are continuously updated.

## 2 Problem Formulation

**Learning.** Let $\ell(\mathbf{w}; z)$ be loss function for a given parameter $\mathbf{w}$ over parameter space $\mathcal{W}$ and sample $z$ over instance space $\mathcal{Z}$. Learning can be expressed as a population risk minimization problem:

$$\min_{\mathbf{w}} \quad F(\mathbf{w}) := \mathbb{E}_{z \sim \mathcal{D}}[\ell(\mathbf{w}; z)]. \tag{1}$$

Addressing this problem directly is challenging, as the probability distribution $\mathcal{D}$ is usually unknown. For a finite dataset $\mathcal{S} = \{z_i\}_{i=1}^n$, one often solves the following empirical risk minimization problem:

$$\min_{\mathbf{w}} \quad F_{\mathcal{S}}(\mathbf{w}) := \frac{1}{n} \sum_{i=1}^{n} \ell(\mathbf{w}; z_i). \tag{2}$$

A standard approach to solve the problem (2) is the stochastic gradient descent (SGD), iterating towards the direction with stepsize $\eta_{e,b}$ of the negative gradient (stochastically):

$$\mathbf{w}_{e,b+1} \leftarrow \mathbf{w}_{e,b} - \frac{\eta_{e,b}}{|\mathcal{B}_{e,b}|} \sum_{i \in \mathcal{B}_{e,b}} \nabla \ell\left(\mathbf{w}_{e,b}; z_i\right), \tag{3}$$

for the index of epoch update $e \in \{0, ..., E+1\}$ and batch update $b \in \{0, ..., B+1\}$, where $E$ is total epochs and $B$ is batches per epoch. Here, $\mathcal{B}_{e,b}$ denotes the set that contains the data sampled in the $e$-th epoch's $b$-th update, and $|\mathcal{B}_{e,b}|$ is the size of $\mathcal{B}_{e,b}$, where $|\mathcal{B}|$ is the maximum batchsize.

**Unlearning.** Consider a scenario where a user requests to remove a forgetting dataset $U = \{u_j\}_{j=1}^m \subseteq \mathcal{S}$. To unlearn $U$, retraining from scratch is considered a naive unlearning method, which aims to minimize $F_{\mathcal{S} \setminus U}(\mathbf{w})$. Specifically, for ease of understanding, each sample appears only in one (mini-)batch per epoch, we define the data point $u_j$ in $U$ to be sampled in $\mathcal{B}_{e,b(u_j)}$ during the $e$-th

epoch's $b(u_j)$-th update in the learning process, where $b(u_j)$ represents the batch update index when $u_j$ is sampled. In accordance with the linear scaling rule discussed in Goyal et al. (2017), i.e. in order to effectively leverage batch sizes, one has to adjust the step size in tandem with batch size. In other words, the step size to batch size ratio during retraining is consistent with the learning process. Therefore, the update rule of retraining process during $e$-th epoch is given by,

$$\mathbf{w}_{e,b+1}^{-U} \leftarrow \mathbf{w}_{e,b}^{-U} - \frac{\eta_{e,b}}{|\mathcal{B}_{e,b}|} \sum_{i \in \mathcal{B}_{e,b}} \nabla \ell(\mathbf{w}_{e,b}^{-U}; z_i). \tag{4}$$

Specifically, for the training data $u_j$ sampled into $\mathcal{B}_{e,b(u_j)}$ in the learning process, the removal of sample $u_j$ in the retraining process results in the following update:

$$\mathbf{w}_{e,b(u_j)+1}^{-U} \leftarrow \mathbf{w}_{e,b(u_j)}^{-U} - \frac{\eta_{e,b(u_j)}}{|\mathcal{B}_{e,b(u_j)}|} \sum_{i \in \mathcal{B}_{e,b(u_j)} \setminus \{u_j\}} \nabla \ell(\mathbf{w}_{e,b(u_j)}^{-U}; z_i). \tag{5}$$

To avoid the unaffordable costs incurred by the retraining algorithm, we aim to approximate the retrained model through limited updates on the learned model $\mathbf{w}_{e,b+1}$. We thus consider the relaxed definition of model indistinguishability defined by Sekhari et al. (2021).

**Definition 1** (($\epsilon, \delta$)-certified unlearning). *Let $S$ be a training set and $\Omega : \mathcal{Z}^n \to \mathcal{W}$ be an algorithm that trains on $S$ and outputs a model $\mathbf{w}$, $\mathcal{T}(S)$ represents the additional data statistics that need to be stored (typically not the entire dataset). Given an output of the learning algorithm $\Omega \in \mathcal{W}$ and a set of data deletion requests $U$, we obtain the results of the unlearning algorithm $\bar{\Omega}(\Omega(S), \mathcal{T}(S)) \in \mathcal{W}$ and the retraining algorithm $\bar{\Omega}(\Omega(S \setminus U), \emptyset)$ through a removal mechanism $\bar{\Omega}$. For $0 < \epsilon \leq 1$, $\delta > 0$, and $\forall \mathcal{W} \subseteq \mathcal{R}^d, S \subseteq \mathcal{Z}$, we say that the removal mechanism $\bar{\Omega}$ satisfies ($\epsilon, \delta$)-certified unlearning if*

$$\begin{aligned} \mathrm{P}(\bar{\Omega}(\Omega(S), \mathcal{T}(S)) \leq e^\epsilon \, \mathrm{P}(\bar{\Omega}(\Omega(S \setminus U), \emptyset))) + \delta, \text{ and} \\ \mathrm{P}(\bar{\Omega}(\Omega(S \setminus U), \emptyset) \leq e^\epsilon \, \mathrm{P}(\bar{\Omega}(\Omega(S), \mathcal{T}(S)) + \delta. \end{aligned} \tag{6}$$

Although classical Gaussian mechanisms (Dwork & Roth (2014)) assume $0 < \epsilon \leq 1$, Balle & Wang (2018) addressed $\epsilon > 1$ by using Gaussian cumulative density function instead of a tail bound approximation. For consistency with prior works, we use classical mechanisms in the main text.

## 3 MAIN INTUITION AND SOLUTION JUSTIFICATION

Before detailing our analysis, let us informally motivate our main method. Specifically, we focus on studying the discrepancy between the learning and the retraining model by analyzing the impact of each sample on the update trajectory. Our main observation is that the difference between the retraining and learning models analyzed through SGD recursion can be described by an *affine stochastic recursion*. Suppose retraining and learning processes share the same initialization, such as using the same random seed or pre-trained model. We now commence the main procedure as follows:

**Unlearning a single data sample.** We start with designing an approximator for a single sample (i.e., $U = \{u\}$) on the training (the learning and retraining) updates, considering the difference between the learning model in Equation (3) and the retraining model obtained without the knowledge of $u$ in Equation (4). In particular, we consider the following two cases of model updates in the $e$-th epoch.

***Case*** **1** ($u$ **not in** $\mathcal{B}_{e,b}$): The difference between retraining and learning updates in the $e$-th epoch is

$$\mathbf{w}_{e,b+1}^{-u} - \mathbf{w}_{e,b+1} = \mathbf{w}_{e,b}^{-u} - \mathbf{w}_{e,b} - \frac{\eta_{e,b}}{|\mathcal{B}_{e,b}|} \sum_{i \in \mathcal{B}_{e,b}} \left( \nabla \ell(\mathbf{w}_{e,b}^{-u}; z_i) - \nabla \ell(\mathbf{w}_{e,b}; z_i) \right). \tag{7}$$

Suppose loss function $\ell \in G^2(\mathbb{R}^d)$, i.e. $\ell$ is twice continuously differentiable. Therefore, from the Taylor expansion of $\nabla \ell(\mathbf{w}_{e,b}^{-u}; z_i)$ around $\mathbf{w}_{e,b}$, we have $\nabla \ell(\mathbf{w}_{e,b}^{-u}; z_i) \approx \nabla \ell(\mathbf{w}_{e,b}; z_i) + \nabla^2 \ell(\mathbf{w}_{e,b}; z_i)(\mathbf{w}_{e,b}^{-u} - \mathbf{w}_{e,b})$. Let $\mathbf{H}_{e,b} = 1/|\mathcal{B}_{e,b}| \sum_{i \in \mathcal{B}_{e,b}} \nabla^2 \ell(\mathbf{w}_{e,b}; z_i)$ denote the Hessian matrix of the loss function, we can then approximate the recursion of Equation (7) as

$$\mathbf{w}_{e,b+1}^{-u} - \mathbf{w}_{e,b+1} \approx (\mathbf{I} - \eta_{e,b} \mathbf{H}_{e,b})(\mathbf{w}_{e,b}^{-u} - \mathbf{w}_{e,b}). \tag{8}$$

***Case* 2** ($u$ **in** $\mathcal{B}_{e,b(u)}$): The difference between retraining and learning updates in the $e$-th epoch is:

$$\mathbf{w}_{e,b(u)+1}^{-u} - \mathbf{w}_{e,b(u)+1} \approx (\mathbf{I} - \eta_{e,b} \mathbf{H}_{e,b})(\mathbf{w}_{e,b(u)}^{-u} - \mathbf{w}_{e,b(u)}) + \underbrace{\frac{\eta_{e,b(u)}}{|\mathcal{B}_{e,b(u)}|} \nabla \ell(\mathbf{w}_{e,b(u)}; u)}_{\text{Approximate impact of } u \text{ in epoch } e}, \tag{9}$$

where $\mathbf{I}$ denotes identity matrix of appropriate size. Combining the observations in Cases 1 and 2, along with same initialization, we now define the *recollection* matrix $\mathbf{M}_{e,b(u)}$ and the *approximator* $\mathbf{a}_{E,B}^{-u}$, backtracking the overall difference between the retrained model and learned model:

$$\mathbf{a}_{E,B}^{-u} := \sum_{e=0}^{E} \mathbf{M}_{e,b(u)} \cdot \nabla\ell(\mathbf{w}_{e,b(u)}; u), \text{ where } \mathbf{M}_{e,b(u)} := \frac{\eta_{e,b(u)}}{|\mathcal{B}_{e,b(u)}|}\hat{\mathbf{H}}_{E,B-1\to e,b(u)+1}. \quad (10)$$

We define $\hat{\mathbf{H}}_{E,B-1\to e,b(u)+1} = (\mathbf{I}-\eta_{E,B-1}\mathbf{H}_{E,B-1})(\mathbf{I}-\eta_{E,B-2}\mathbf{H}_{E,B-2})...(\mathbf{I}-\eta_{e,b(u)+1}\mathbf{H}_{e,b(u)+1})$, which represents the product of $\mathbf{I}-\eta\mathbf{H}$ from $E$-th epoch's $B-1$-th update to $e$-th epoch's $b(u)+1$-th update.[2] This provides an approximation of the impact of sample $u$ on the training trajectory:

$$\mathbf{w}_{E,B}^{-u} - \mathbf{w}_{E,B} \approx \mathbf{a}_{E,B}^{-u} \quad (11)$$

Colloquially, a simple vector $\mathbf{a}_{E,B}^{-u}$ addition to model $\mathbf{w}_{E,B}$ can approximate retrained model $\bar{\mathbf{w}}_{E,B}^{-u}$ for forgetting the sample $u$. We define $\Delta_{E,B}^{-u} = \mathbf{w}_{E,B}^{-u} - \mathbf{w}_{E,B} - \mathbf{a}_{E,B}^{-u}$ as the approximation error.

The approximator designed in Equation (11) enables us to exploit the Hessian vector product (HVP) technique (Pearlmutter (1994)) to compute it without explicitly computing Hessian. When computing the product of the Hessian matrix and an arbitrary vector for the neural networks, the HVP technique first obtains gradients during the first backpropagation, multiplying the gradients with the vector, and then performing backpropagation again. This results in a time complexity of $\mathcal{O}(d)$ (Dagréou et al. (2024)), which has the same order of magnitude as computing a gradient. We defer the detailed calculation process of Equation (11) using HVP to Appendix B.3. We note that the existing second-order methods require proactively storing a Hessian (or its inverse) (Sekhari et al. (2021); Suriyakumar et al. (2022); Liu et al. (2023)). It is thus unable to deploy HVP in their settings.

Note that our proposed analysis does not require to invert Hessian matrix, distinguishing it from existing Newton-like methods (Sekhari et al. (2021); Suriyakumar et al. (2022); Liu et al. (2023)). This eliminates the need for assuming convexity to ensure the invertibility of the Hessian matrix. Moreover, when it comes to forgetting a set of samples, we have,

**Theorem 1** (**Additivity**). *When $m$ sequential deletion requests arrive, the sum of $m$ approximators is equivalent to performing batch deletion simultaneously. Colloquially, for $u_1, \ldots, u_m$ sequence of continuously arriving deletion requests, we demonstrate that $\mathbf{a}_{E,B}^{-\sum_{j=1}^{m} u_j} = \sum_{j=1}^{m} \mathbf{a}_{E,B}^{-u_j}$.*

See complete analysis and proofs of Section 3 in Appendix C.1 and Appendix C.2. This elucidates that by storing the impact of updates for each sample, nearly instantaneous data removal can be achieved through vector additions. Note that it is non-trivial to retrain and store a leave-one-out model for each sample, as such a process cannot accommodate sequential deletion requests. If we were to consider all possible combinations of forgetting data, retraining need to pretrain $\mathcal{O}(2^n)$ models, which yields computation complexity $\mathcal{O}(2^n EBd)$. The additivity enables our Hessian-free method to handle multiple deletion requests in an online manner. Previous Hessian-based methods, lacking additivity, fail to utilize HVP as the forgetting sample $u$ is unknown before a deletion request arrives.

## 4 THEORETICAL ANALYSIS AND ALGORITHM DESIGN

In this section, we theoretically analyze the ❶ **Unlearning Guarantee** (Theorem 4), ❷ **Generalization Guarantee** (Theorem 6), and ❸ **Deletion Capacity Guarantee** (Theorem 8). Particularly, we demonstrate that under the Assumption 1, our method can outperform existing theoretical certified unlearning methods (Sekhari et al. (2021); Suriyakumar et al. (2022); Liu et al. (2023)) in terms of unlearning guarantee, generalization guarantee, and deletion capacity. In contrast to these works, the analysis of unlearning guarantee is applicable to both convex and non-convex settings, as our proposed method does not involve Hessian inversion. Based on the foregoing analysis, we further develop an almost instantaneous online unlearning algorithm, which is one of the most efficient, as it requires only a vector addition operation to delete each training data. Finally, we conduct storage and computation complexity analysis and show that in most scenarios where over-parameterized models and the ratio of $E$ to $B$ are much smaller than the parameters, it is more efficient than previous works.

---

[2]Similar analysis has been conducted in other lines of work to investigate the behavior of SGD dynamics, e.g. Gürbüzbalaban et al. (2021) focused on the properties of the matrix $\mathbf{I} - \eta\mathbf{H}$, which is termed as multiplicative noise, serving as the main source of heavy-tails in SGD dynamics and governing the update behavior of $\mathbf{w}_{e,b}$.

## 4.1 APPROXIMATION ERROR ANALYSIS

To analyze the approximation error $\Delta_{E,B}^{-U}$, we introduce the following lemmata. We first show that the geometrically decaying stepsize strategy facilitates us to derive an upper bound for the error term.

**Assumption 1.** *For any $z$ and $\mathbf{w}$, loss $\ell(\mathbf{w};z)$ is $\lambda$-strongly convex, $L$-Lipschitz and $M$-smooth in $\mathbf{w}$.*

**Lemma 2.** *For any $e, b$ and $z \in \mathcal{B}_{e,b}$, there exists $G$ such that $G = \max \|\nabla \ell(\mathbf{w}_{e,b}; z)\| < \infty$. Consider geometrically decaying stepsizes satisfying $\eta_{e,b+1} = q\eta_{e,b}$, where $0 < q < 1$ and $\eta = \eta_0$. During the $e$-th epoch's $b$-th batch update, we have:*

$$\|\mathbf{w}_{e,b}^{-U} - \mathbf{w}_{e,b}\| \le 2\eta G \frac{1 - q^{eB+b}}{1 - q}. \tag{12}$$

See proof in Appendix C.3. Then, we further analyze the eigenvalue of the Hessian matrix:

**Lemma 3.** *For any $e, b$ and vector $\mathbf{v} \in \mathbb{R}^d$, there exists $\lambda$ and $M$ such that $\lambda \mathbf{I} \preceq \nabla^2 \ell(\mathbf{w}_{e,b}; z_i) \preceq M\mathbf{I}$, and let $\rho = max\{|1 - \eta_{e,b}\lambda|, |1 - \eta_{e,b}M|\}$ be the spectral radius of $\mathbf{I} - \eta_{e,b}\mathbf{H}_{e,b}$, we have $\|(\mathbf{I} - \eta_{e,b}\mathbf{H}_{e,b})\mathbf{v}\| \le \rho\|\mathbf{v}\|$.*

Combining Equation (11), Lemma 2 and Lemma 3, we have the approximation error bound:

**Theorem 4 (Unlearning Guarantee).** *For any subset $U = \{u_j\}_{j=1}^m$ to be forgotten, with $0 \le E < \infty$ and $q < \rho$, the unlearning model $\bar{\mathbf{w}}_{E,B}^{-U} = \mathbf{w}_{E,B} + \mathbf{a}_{E,B}^{-U}$ yields an approximation error bounded by*

$$\|\Delta_{E,B}^{-U}\| = \|\mathbf{w}_{E,B}^{-U} - \bar{\mathbf{w}}_{E,B}^{-U}\| \le \mathcal{O}(\eta Gm\rho^{nE/|\mathcal{B}|}), \tag{13}$$

*where the total number of batch updates per epoch $B = \lceil n/|\mathcal{B}| \rceil$.*

See proofs and analysis in Appendix C.4. We note that in general, Theorem 4 does not require the objectives to be strongly convex as it does not involve inverting the Hessian. Therefore, our method accommodates most non-convex and non-$M$-smooth functions. On the other hand, considering that previous theoretical works require the convexity assumption, for further comparing with these works and better analyzing the following generalization results in Subsection 4.2, we show that $L^2$-regularized convex problems with random regularization coefficients fall into the scope of Theorem 4.

To achieve a tighter bound, we consider the standard Assumption 1, which are similar to those in unlearning works (Suriyakumar et al. (2022); Liu et al. (2023)). We have the following conclusion,

**Corollary 5.** *Suppose Assumption 1 is satisfied, and we choose $\eta < 2/(M + \lambda)$ to ensure gradient descent convergence and $\rho < 1$. Then, Theorem 4 yields an approximation error bound by $\mathcal{O}(m\rho^n)$.*

Under same Assumption 1, the derived bound surpasses the state-of-the-art $\mathcal{O}(m^2/n^2)$ in Suriyakumar et al. (2022); Liu et al. (2023), representing an improvement for unlearning guarantee. Moreover, Corollary 5 implies that the unlearning model can closely approximate the retrained model.

## 4.2 GENERALIZATION PROPERTIES ANALYSIS

Despite that our proposed method is applicable to both convex and non-convex settings, we temporarily impose Assumption 1 for the analytical simplicity of generalization guarantee. This assumption will be relaxed for algorithm design, unlearning analyses, and experiments. We further analyze the generalization guarantee and compare the results with previous works.

**Theorem 6 (Generalization Guarantee).** *Let Assumption 1 hold. Choose $\eta \le \frac{2}{M+\lambda}$ such that $\rho < 1$ for $0 \le E < \infty$. Let $\mathbf{w}^*$ be the minimizer of the population risk in Equation (1). Considering noise perturbation to ensure $(\epsilon, \delta)$-unlearning, the unlearned model $\tilde{\mathbf{w}}_{E,B}^{-U}$ satisfies the excess risk bound,*

$$F(\tilde{\mathbf{w}}_{E,B}^{-U}) - \mathbb{E}[F(\mathbf{w}^*)] = \mathcal{O}\left(\frac{4L^2}{\lambda(n-m)} + \rho^{nE/|\mathcal{B}|}\left(\frac{2L^2}{\lambda} + \frac{2mL\eta G\sqrt{d}\sqrt{\ln(1/\delta)}}{\epsilon}\right)\right). \tag{14}$$

See proof in Appendix C.6. The result of our proposed method yields the order $\mathcal{O}(d^{\frac{1}{2}}\rho^n)$. Under these conditions and the fact that $n = \Omega(m)$, our proposed method outperforms the performance of the state-of-the-art methods (Suriyakumar et al. (2022); Liu et al. (2023)), which are $\mathcal{O}(d^{\frac{1}{2}}/n^2)$. Moreover, to measure the maximum number of samples to be forgotten while ensuring a good excess risk guarantee, we provide our proposed scheme's *deletion capacity*, following the analysis in Sekhari et al. (2021). Due to the space limit, we defer the definition and analysis to Appendix B.2.

---

**Algorithm 1:** Hessian-Free Online Unlearning (**HF**) Algorithm

1  **Stage I**: **Learning** model $\mathbf{w}_{E,B}$ on dataset $S = \{z_i\}_{i=1}^n$:
2  **for** $e = 0, 1 \cdots, E+1$ **do**
3     | **for** $b = 0, 1 \cdots, B+1$ **do**
4     |    | Gradient descent: $\mathbf{w}_{e,b+1} \leftarrow \mathbf{w}_{e,b} - \eta_{e,b} \frac{1}{|\mathcal{B}_{e,b}|} \sum_{i \in \mathcal{B}_{e,b}} \nabla \ell (\mathbf{w}_{e,b}; z_i)$
5     | **end**
6  **end**
7  **Stage II**: **Pre-computing and Pre-storing** unlearning statistics $\mathcal{T}(\mathcal{S}) = \{\mathbf{a}_{E,B}^{-u_j}\}_{j=1}^n$:
8  **for** $j = 1, 2 \cdots, n$ **do**
9     | Recursive HVP based on Algorithm 2: $\mathbf{a}_{E,B}^{-u_j} \leftarrow \sum_{e=0}^E \mathbf{M}_{e,b(u)} \cdot \nabla \ell(\mathbf{w}_{e,b(u)}; u)$.
10 **end**
11 **Stage III**: **Unlearning** when user requests to forget the subset $U = \{u_j\}_{j=1}^m$ on model $\mathbf{w}_{E,B}$:
12 Compute: $\bar{\mathbf{w}}_{E,B}^{-U} \leftarrow \mathbf{w}_{E,B} + \mathbf{a}_{E,B}^{-U}$, $\sigma \leftarrow \frac{\|\Delta_{E,B}^{-U}\|\sqrt{2\ln(1.25/\delta)}}{\epsilon}$, Delete unlearning statistics: $\{\mathbf{a}_{E,B}^{-u_j}\}_{j=1}^m$
13 Sample: $\mathbf{n} \sim \mathcal{N}(0, \sigma \mathbf{I})$, Return: $\tilde{\mathbf{w}}_{E,B}^{-U} = \bar{\mathbf{w}}_{E,B}^{-U} + \mathbf{n}$

---

### 4.3 EFFICIENT UNLEARNING ALGORITHM DESIGN

Building on the foregoing techniques and analysis of Theorem 4, we propose a memory and computationally efficient unlearning scheme in Algorithm 1. To the best of our knowledge, our algorithm is one of the most efficient method, as it only requires a vector addition by utilizing pre-computed statistics. This is particularly advantageous in time-sensitive scenarios and non-optimal models.

We present our approach in Algorithm 1. In **Stage I**, we perform conventional model training. Upon completing the model training, we proceed to **Stage II**, where we compute the unlearning statistics vector $\mathbf{a}_{E,B}^{-u}$ for each sample $u$. To reduce computational complexity, we leverage the HVP technique as described in Appendix B.3. The computed approximators $\{\mathbf{a}_{E,B}^{-u_j}\}_{j=1}^n$ are then stored for further use.[3] A deletion request initiates **Stage III**. In this stage, the indistinguishability of the $(\epsilon, \delta)$-unlearning, as defined in Definition 1, is achieved through a single vector addition with noise perturbation. Notably, Algorithm 1 does not necessitate accessing any datasets during Stage III.

### 4.4 COMPARISON TO EXISTING APPROACHES

We provide a brief overview of the second-order unlearning algorithms and compare the complexity of our method with these works in terms of **pre-computation**, **storage**, and **unlearning time**.
*Newton Step (NS)*. Sekhari et al. (2021) proposed a certified unlearning algorithm by pre-computing Hessian $\sum_{i=1}^n \nabla^2 \ell(\hat{\mathbf{w}}; z_i)$. When a request arrives to forget $U = \{u_j\}_{j=1}^m$, the update for *NS* is as follows:

$$\bar{\mathbf{w}}_{NS}^{-U} = \hat{\mathbf{w}} + \frac{1}{n-m}\Big( \frac{1}{n-m}\big( \sum_{i=1}^n \nabla^2 \ell(\hat{\mathbf{w}}; z_i) - \sum_{j \in U} \nabla^2 \ell(\hat{\mathbf{w}}; u_j) \big) \Big)^{-1} \sum_{j \in U} \nabla \ell(\hat{\mathbf{w}}; u_j). \tag{15}$$

*Infinitesimal Jackknife (IJ)*. Suriyakumar et al. (2022) built upon the work of *NS* and facilitated forgetting by pre-computing and storing the inverse Hessian. The update for *NS* thus is as follows:

$$\bar{\mathbf{w}}_{IJ}^{-U} = \hat{\mathbf{w}} + \frac{1}{n}\Big( \frac{1}{n} \sum_{i=1}^n \nabla^2 \ell(\hat{\mathbf{w}}; z_i) \Big)^{-1} \sum_{j \in U} \nabla \ell(\hat{\mathbf{w}}; u_j). \tag{16}$$

These methods fail to use HVP, for Hessian $\sum_{j \in U} \nabla^2 \ell(\hat{\mathbf{w}}; u_j)$ in *NS* and gradient $\nabla \ell(\hat{\mathbf{w}}, u_j)$ in both *NS* and *IJ*, (i) forgetting sample $u_j$ is unknown before deletion request arrives, and (ii) the learned empirical risk minimzer $\hat{\mathbf{w}}$ will be deleted after processing a deletion request, while subsequent unlearned model is also unknown as forgetting sample $u_j$ is unpredictable. Therefore, explicit pre-computing is required for $\sum_{i=1}^n \nabla^2 \ell(\hat{\mathbf{w}}; z_i)$ in *NS* and $\big(\sum_{i=1}^n \nabla^2 \ell(\hat{\mathbf{w}}; z_i)\big)^{-1}$ in *IJ*.

---

[3]Storing additional data-dependent vectors may incur supplementary privacy risks. A straightforward solution is to have the users retain custody of these vectors. Analogous to Ghazi et al. (2023), upon submitting a deletion request, the user is then required to furnish the corresponding approximators as part of the deletion request.

**Comparison of Precomputation.** For a single data, Algorithm 1 entails computing the HVP $\mathcal{O}(En/|\mathcal{B}|)$ times. To precompute the Hessian-free (HF) approximators $\mathbf{a}_{\text{HF}}^{-u}$ for all $n$ samples, the complexity is $\mathcal{O}(n^2 dE/|\mathcal{B}|)$. Typically, the ratio of $E$ to $|\mathcal{B}|$ is much smaller than $d$ or $n$ (Smith (2018)), which simplifies the complexity to $\mathcal{O}(n^2 d)$. HF method is highly efficient when dealing with overparameterized models, especially when $d \gg n$.[4] This is because computing the full Hessian for *NS* requires $\mathcal{O}(nd^2)$ time, and computing and inverting the Hessian $\mathbf{H}_{\text{IJ}}^{-1}$ for *IJ* requires $\mathcal{O}(d^3 + nd^2)$.

**Comparison of Storage.** When we compute the approximator, it requires storing $\mathcal{O}(En/|\mathcal{B}|)$ models in Stage I, leading to a maximum complexity of $\mathcal{O}(ndE/|\mathcal{B}| + nd)$, which can simplify to $\mathcal{O}(2nd)$. After complete precomputing, Algorithm 1 only needs the storage of $\mathcal{O}(nd)$. This is lower than the overhead introduced by storing matrices for *NS* and *IJ* in overparameterized models, which is $\mathcal{O}(d^2)$.

**Comparison of Unlearning Computation.** When an unlearning request with $m$ forgetting data arrives, Algorithm 1 utilizes precomputed and prestored approximators to achieve forgetting, requiring a simple vector addition that takes $\mathcal{O}(md)$ time. For *NS*, it requires computing and inverting different Hessian that depends on the user requesting the deletion. This necessitates substantial computational cost of $\mathcal{O}(d^3 + md^2 + md)$ each time a deletion request arrives. Although *IJ* reduces this to $\mathcal{O}(d^2 + md)$ by precomputing the inverse Hessian, it is still not as efficient compared to our method.

## 5 EXPERIMENTS FROM DEVELOPER'S PERSPECTIVE

We devised the following experimental components: ❶ **Verification** and ❷ **Application.** Verification experiments are centered on evaluating the disparity between the unlearning model $\bar{\mathbf{w}}_{E,B}^{-U}$ and retrained models $\mathbf{w}_{E,B}^{-U}$, across both convex and non-convex settings. Application experiments are geared towards evaluating the performance of different unlearning algorithms after perturbing $\tilde{\mathbf{w}}_{E,B}^{-U}$ with noise $\mathbf{n}$ with a specific focus on determining the accuracy, pre-computation/storage, and unlearning time. Considering the space limit, we defer a portion of experiments to the Appendix D.

### 5.1 VERIFICATION EXPERIMENTS

Our principal aim is to validate the differences between the proposed Hessian-free (HF) approximator $\mathbf{a}_{\text{HF}}^{-U}$ and the actual $\mathbf{w}_{E,B}^{-U} - \mathbf{w}_{E,B}$, where a smaller difference indicates the unlearned model is closer to retrained model and also reflects the degree of data forgetting. Therefore, we evaluate the degree of forgetting from two perspectives: *distance* and *correlation*. Specifically, we use the $L^2$-norm $\|\mathbf{w}_{E,B}^{-U} - \mathbf{w}_{E,B} - \mathbf{a}^{-U}\|$ to measure the distance metric. A smaller distance metric indicates the approximators accurately capture the disparity between the retrained and learned models. Additionally, we utilize the Pearson (Wright (1921)) and Spearman (Spearman (1961)) coefficients to assess the correlation between $\mathbf{a}^{-U}$ and $\mathbf{w}_{E,B}^{-U} - \mathbf{w}_{E,B}$ by mapping them from high-dimensional space to scalar loss values, i.e. the approximate loss change $\ell(\mathbf{w}_{E,B} + \mathbf{a}^{-U}; U) - \ell(\mathbf{w}_{E,B}; U)$ and the actual change $\ell(\mathbf{w}_{E,B}^{-U}; U) - \ell(\mathbf{w}_{E,B}; U)$ on the forgetting dataset $U$. The correlation scores range from -1 to 1, where the higher score, the more the unlearned method performs like the retrained model.

**Configurations:** We conduct experiments in both convex and non-convex scenarios. Specifically, we trained a multinomial Logistic Regression (LR) with total parameters $d = 7850$ and a simple convolutional neural network (CNN) with total parameters $d = 21840$ on MNIST dataset (Deng (2012)) for handwriting digit classification. We apply a cross-entropy loss and the inclusion of an $L^2$-regularization coefficient of 0.5 to ensure that the loss function of LR is strongly convex. For LR, training was performed for 15 epochs with a stepsize of 0.05 and a batch of 32. For CNN, training was carried out for 20 epochs with a stepsize of 0.05 and a batch size of 64. Given these configurations, we separately assess the distance and correlation between approximators $\mathbf{a}_{\text{HF}}^{-U}$, $\mathbf{a}_{\text{NS}}^{-U}$, $\mathbf{a}_{\text{IJ}}^{-U}$ at deletion rates in the set $\{1\%, 5\%, 10\%, 15\%, 20\%, 25\%, 30\%\}$. Following the suggestion in Basu et al. (2021), a damping factor of 0.01 is added to the Hessian to ensure its invertibility when implementing *NS* and *IJ*. Besides, we conducted experiments with 7 random seeds to obtain average results. For the sake of clear visualizations, we use the minimum and maximum values as error bars.

---

[4]Deep networks for computation vision tasks are typically trained with a number of model parameters that far exceeds the size of the training dataset (Chang et al. (2021)). For general neural language model, the empirical data-to-parameter scaling law is also expected to remain at $n \sim d^{0.74}$, as suggested in Kaplan et al. (2020).

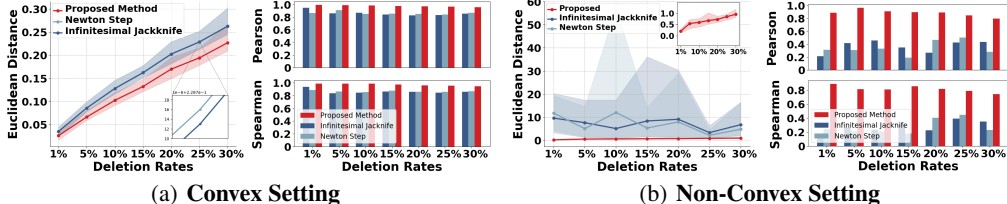

(a) **Convex Setting**          (b) **Non-Convex Setting**

Figure 1: **Verification experiments I** on (a) LR and (b) CNN, respectively. The left of (a)(b) shows the distance to the retrained model, i.e. $\|\mathbf{w}_{E,B}^{-U} - \mathbf{w}_{E,B} - \mathbf{a}^{-U}\|$. The right of (a)(b) shows the correlation between the approximate loss change and actual loss change on the forgetting dataset.

Table 2: **Application experiments I.** An online scenario with 20% samples to be forgotten, where each unlearning request involves forgetting a single data point with each execution. A performance gap against Retrain is provided in ($\cdot$). We compute the total computational/storage cost for all samples.

|  | Model | Unlearning Computation | | Pre-Computaion | Storage | Test Accuracy (%) |
|---|---|---|---|---|---|---|
|  |  | Runtime (Sec) | Speedup | Runtime (Sec) | (GB) | Unlearned model |
| NS | LR | $5.09 \times 10^2$ | $1.34\times$ | $2.55 \times 10^3$ | 0.23 | 86.25 (-1.50) |
|  | CNN | $5.81 \times 10^3$ | $0.12\times$ | $2.91 \times 10^4$ | 1.78 | 83.50 (-10.25) |
| IJ | LR | $0.23 \times 10^2$ | $29.64\times$ | $2.55 \times 10^3$ | 0.23 | 86.25 (-1.50) |
|  | CNN | $1.91 \times 10^2$ | $3.63\times$ | $2.91 \times 10^4$ | 1.78 | 82.75 (-11.00) |
| **HF** | LR | **0.0073** | **93,376$\times$** | **$2.20 \times 10^2$** | **0.03** | **87.50** (-0.25) |
|  | CNN | **0.0268** | **25,821$\times$** | **$5.34 \times 10^2$** | **0.08** | **91.50** (-2.25) |

**The convex setting**. As shown in Figure 1 (a), our approach outperforms previous works. First, by distance evaluation, the approximation error $\|\Delta_{E,B}^{-U}\|$ of the proposed method is lower than that of the previous *NS* and *IJ* works. At a deletion rate of 30%, the error is 0.2097, lower than that of *NS* and *IJ*, which have errors of 0.246554 and 0.246557, respectively. Second, in assessing the loss change for the forgotten dataset $U$, our proposed method more accurately captures the actual changes. Particularly when removing 30% of the samples, the proposed method maintains a high correlation, with Pearson and Spearman being 0.96 and 0.95, respectively. These evaluations demonstrate the efficacy of both Hessian-free and Hessian-based methods in accurately approximating the retrained model. Our proposed approach not only achieves a closer approximation to the retrained model but also exhibits lower complexity compared to previous methods. This is substantiated by the results of Application Experiments I, which provide empirical evidence supporting our theoretical findings.

**The non-convex setting**. As illustrated in Figure 1 (b), the proposed *HF* method demonstrates superior performance, exhibiting less dependency on random variations compared to *NS* and *IJ*. When 30% of the samples are removed, our method maintains a lower approximation error of 0.90, surpassing the performance of *NS* and *IJ*. Our methods achieve Spearman and Pearson correlation coefficients of 0.81 and 0.74, respectively, accurately capturing actual loss changes, in contrast to *NS* and *IJ* which are far from the actual values in non-convex settings. In Appendix D.2, we explain the volatility of the results in the prior works on non-convex setting, and provide more large-scale results, which is not achievable in *NS* and *IJ* (Sekhari et al. (2021); Suriyakumar et al. (2022); Liu et al. (2023)) due to out-of-memory to precisely computing the Hessian matrix (or its inverse).

## 5.2 APPLICATION EXPERIMENTS

We compare the cost and performance of our method with the previous certified unlearning methods. Evaluations are conducted from two perspectives: *runtime and utility*. Specifically, runtime focuses on the time spent pre-computing unlearning statistics and the speedup of the unlearning algorithm compared to the retraining. Moreover, we evaluate the utility by the test accuracy of unlearned model to ensure generalization guarantee. We train LR and CNN with 20% data to be forgotten, where the configuration is identical to the foregoing verification experiments I. We conduct experiments with the unlearning guarantee of $\epsilon = 1, \delta = 10^{-3}$ for LR and $\epsilon = 100, \delta = 0.1$ for CNN. We provide larger scale results in Appendix D.3, and we have not evaluated *NS* and *IJ* due to out-of-memory.

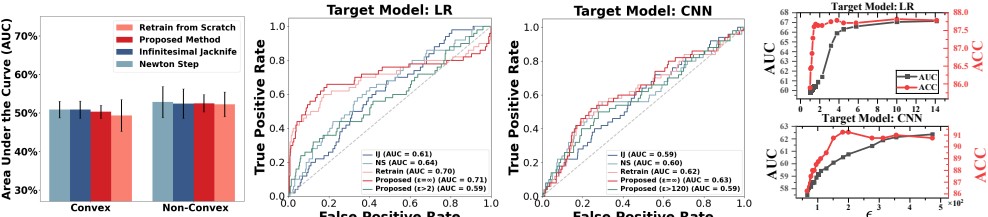

Figure 2: **Membership Inference Attack I.** The first plot shows the MIA-L and error bars show 95% confidence intervals; the second and third plots depict the MIA-U results; and the fourth plot illustrates the privacy-utility tradeoff. Proposed method successfully defends MIA-L and mitigates excessive privacy leakage from MIA-U without sacrificing performance when applied with appropriate noise.

**Application experiment results.** Table 2 demonstrates that the proposed algorithm outperforms other baselines by a significant margin. Strikingly, the proposed algorithm outperforms benchmarks of certified unlearning regarding all metrics with millisecond-level unlearning and minimal performance degradation in both convex and non-convex settings. When models exceed the complexity of LR, precisely estimating the full Hessian becomes infeasible. This conclusion is supported by Table 2 and Mehta et al. (2022). Therefore, we demonstrate that the proposed Hessian-free perspective method exhibits great potentials for over-parameterized models compared to the Hessian-based approaches.

## 6 EXPERIMENTS FROM ADVERSARY'S PERSPECTIVE

To further investigate the effectiveness of data removal for auditing, we perform membership inference attack (MIA) experiments from the attacker's perspective, including two state-of-the-art MIA: ❶ **MIA against Learning** (**MIA-L**) (LiRA MIA Carlini et al. (2022)) and ❷ **MIA against Unlearning** (**MIA-U**) (Chen et al. (2021)). Specifically, MIA-L enables an adversary to query a trained ML model to predict whether a specific example was included in the model's training dataset; MIA-U involves querying two versions of the ML model to infer whether a target sample is part of the training set of the learned model but has been removed from the corresponding unlearned model (primarily targeting retrain algorithm). See Appendix D.4 for a detailed introduction and more experimental results.

**MIA results** with 5% data to be forgotten. As shown in the first plot of Figure 2, our proposed method effectively defends against MIA-L even in low-privacy scenarios, as evidenced by effective approximation of the retrained model in the verification experiment. However, MIA-U exploits discrepancies between the outputs of the model's retrained and original versions. The verification experiment also reveals that our unlearned model's loss change correlation coefficient is closely aligned with that of the retrained model, indicating that, like the retrained model, it is similarly vulnerable to such attacks. As depicted in the second and third plots, MIA-U shows high AUC values for the retrained method (light red line) and our method (red line) without perturbation ($\epsilon = \infty$). Nevertheless, adding appropriate noise to the model can mitigate MIA-U. As demonstrated in the fourth plot, introducing an appropriate noise level ($\epsilon > 2$ for LR and $\epsilon > 120$ for CNN) effectively reduces the excessive privacy leakage without compromising performance. Besides, further increasing the model perturbation to tradeoff privacy-utility does not significantly reduce the effectiveness of such attacks. To better defend MIA-U, we will consider measures like noised SGD in future work.

## 7 CONCLUSION

We proposed a novel Hessian-free certified unlearning method. Our analysis relies on an affine stochastic recursion to characterize the model update discrepancy between the learned and retrained models, which does not depend on inverting Hessian and model optimality. We provide results of unlearning, generalization, deletion capacity guarantees, and time/memory complexity that are superior to state-of-the-art methods. We then develop an algorithm that achieves near-instantaneous unlearning. We show that the proposed method can reduce the precomputation time, storage, and unlearning time to $\mathcal{O}(n^2 d)$, $\mathcal{O}(nd)$, $\mathcal{O}(d)$, respectively. Experimental results validate that our work benefits certified unlearning with millisecond-level unlearning time and improved test accuracy.

## ACKNOWLEDGMENTS

This work was supported in part by NSF grants CMMI-2024774 and ECCS-2030251, and in part by the National Natural Science Foundation of China under Grant 62202427.

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

CONTENTS OF APPENDIX

## A    AN EXPANDED VERSION OF THE RELATED WORK

**Machine Unlearning**, the terminology can be traced back to the paper of Cao & Yang (2015), and their work defined the notion of "the completeness of forgetting". There are two main categories based on the completeness of forgetting: Exact Unlearning and Approximate Unlearning.

**Exact Unlearning** aims to accelerate the speed of retraining and completely eliminate the influence of forgotten data on learned model. The majority of existing Exact Unlearning methods are derived from the work of Bourtoule et al. (2021), which proposed the notable SISA framework, grounded in ensemble learning techniques. Specifically, the training dataset is partitioned into mutually disjoint shards, and then sub-models are trained on these shards. Upon unlearning, SISA only needs to retrain the sub-model correlated with the relevant data shard and then make the final prediction by assembling the knowledge of each sub-model, which significantly lowers retraining computational costs. Under the SISA framework, other Exact Unlearning methods have also been proposed, e.g., Chen et al. (2022b) extended SISA to graph learning by partitioning the original training graph into disjoint shards, parallelly training a set of shard models, and learning an optimal importance score for each shard model. RecEraser (Chen et al. (2022a)) extended SISA to recommendation systems. KNOT (Su & Li (2023)) adopted the SISA for client-level asynchronous federated unlearning during training and introduced a clustered mechanism that divides clients into multiple clusters. Other methods, such as DaRE (Brophy & Lowd (2021)) and HedgeCut (Schelter et al. (2021)), addressed the issue that SISA cannot be applied to decision trees and proposed exact unlearning solutions; and Ginart et al. (2019) aimed to achieve forgetting in K-means clustering. Furthermore, ARCANE (Yan et al. (2022)) introduced a class-based data partitioning strategy to enhance the performance of SISA.

However, the SISA framework adopts a structure that completely separates the data, resembling ensemble learning approaches where predictions or models are aggregated only at the end. This separation restricts the model training process from fully leveraging the entirety of the segmented datasets and also hampers its extensibility, leading to lower accuracy and performance compared to a model trained on the entire unsegmented dataset. In an experimental evaluation conducted by Koch & Soll (2023), it was found that simply down-sampling the data set to achieve a more balanced single shard outperforms SISA in terms of unlearning speed and performance.

**Approximate Unlearning** thus seeks to minimize the impact of forgetting data to an acceptable level to trade off computational efficiency, reduced storage costs, and flexibility. There are many approaches to approximate unlearning, such as Kurmanji et al. (2024) developed a distillation-based method; Halimi et al. (2022) adopted a gradient ascent approach on forgotten data to achieve forgetting. Among these, second-order methods have been widely studied, with most of these methods (e.g., Sekhari et al. (2021); Suriyakumar et al. (2022); Liu et al. (2023); Chien et al. (2022); Wu et al. (2023); Mehta et al. (2022); Warnecke et al. (2023); Wu et al. (2022)) inspired by influence function studies (Koh & Liang (2017); Basu et al. (2021; 2020); Hara et al. (2019)), aiming to extract information from the Hessian of loss function and achieve forgetting through limited model updates. In particular, inspired by differential privacy (Dwork (2006)), Guo et al. (2020) introduced certified unlearning $(\epsilon, \delta)$, anticipating ensuring that the output distribution of an unlearning algorithm is indistinguishable from that of retraining. Moreover, due to costly Hessian operations, Fisher-based methods (Golatkar et al. (2020a;b; 2021)) have emerged to approximate Hessian. Our paper concentrates on the Hessian-based approach, yet obviates the need for Hessian inversion and explicit calculation of the Hessian.

Among these Hessian-based certified unlearning methods, Sekhari et al. (2021); Suriyakumar et al. (2022); Liu et al. (2023) established the theoretical foundation for Hessian-based methods and provided the generalization result in terms of the population risk, as well as derived the deletion capacity guarantee. In the definition of certified unlearning, their works align with our method, which lies in pre-computing statistics of data that extract second-order information from the Hessian and using these 'recollections' to forget data. However, existing second-order methods still require expensive matrix computations and can only perform forgetting for optimal models under convexity assumption. Although Zhang et al. (2024) offered certified unlearning for deep model through adding sufficient large regularization to guarantee local strong convexity, but the optimality problem remains unsolved, particularly due to the non-convexity of modern deep models, making it impractical to find the empirical risk minimizer. Even for convex models, in most non-concurrent training scenarios, such as online learning or continual learning, the model cannot converge to the optimal solution.

# B DISCUSSIONS

## B.1 LIMITATIONS AND FUTURE WORKS

**Limitations.** Although we have made the above contributions, there are some significant limitations:

① Our proposed algorithm necessitates $\mathcal{O}(n^2 d)$ precomputation and $\mathcal{O}(nd)$ storage, which remains costly, particularly in scenarios involving extensive data and larger-scale models. Consequently, conducting experiments concurrently with both large-scale datasets and models is anticipated to significantly extend the time required for precomputing unlearning statistics.

② As demonstrated by the results in Theorem 4, it is important to recognize that the proposed method is highly sensitive to the the stability of the training process or well-conditioned Hessian on non-convex setting. The addition of a large explicit $L^2$-regularizer to the objective or utilizing small step sizes can ensure a meaningful bound. However, we are not aware of any popular non-convex machine learning problem that can satisfy all the conditions of Theorem 4. If the learning process is unstable, unlearning will introduce significant error. For instance, in Appendix E.1, employing an aggressive step size ($\eta > 0.1$) can lead to a substantial increase in approximation error.

③ While we propose a new method that does not rely on convexity, our analysis builds upon the theoretical framework of previous works for comparison purposes and much of our theoretical results rely on convexity—except for our unlearning-related conclusions. Providing a nonconvex theoretical framework is meaningful, and we will explore potential extensions in further research.

**Future work.** Therefore, to solve the problems mentioned above, we briefly propose some solutions.

① Specifically, our proposed framework offers the feasibility to unlearn an arbitrary subset of the entire datasets, allowing a trade-off that further reduces the precomputation and storage complexity, for instance, consider a typical scenario at the user level, where there are $k$ users ($k << n$). By focusing on the unlearning of a single user's dataset or a subset of users' datasets, we can maintain $k$ vectors for $k$ users' dataset instead of maintaining $n$ vectors for $n$ all data points. Therefore, we can further reduce precomputation and storage to $\mathcal{O}(knd)$ and $\mathcal{O}(kd)$, respectively.

② Moreover, based on our analysis in Theorem 4 and Appendix E.1, the main source of the approximation error is $\rho$, which mainly consists of two components: the step size $\eta$ and the maximum eigenvalue $\lambda_1$ of Hessian. Reducing the step size is straightforward method, but we cannot expect every training process to have a sufficiently small step size. Therefore, reducing $\lambda_1$ during optimization, will not only improve the generalization ability of the learned model but also reduce the approximation error in the unlearning stage. Possible approaches are through stepsize warm-up strategies (Gilmer et al. (2022)) or sharpness-aware minimization (Wen et al. (2023)) to effectively reduce $\lambda_1$.

## B.2 DELETION CAPACITY GUARANTEE

We analyze the *deletion capacity* of our method based on the analysis of Sekhari et al. (2021), which formalizes how many samples can be deleted from the model parameterized by the original empirical risk minimizer while maintaining reasonable guarantees on the test loss. We establish the relationship between generalization performance and the amount of deleted samples through deletion capacity, and our analysis indicates that our proposed method outperforms previous works in both generalization and deletion capacity. The lower bound on deletion capacity can be directly derived from the generalization guarantee (Theorem 6). Below, we first formalize the deletion capacity.

**Definition 2 (Definition of Deletion Capacity** (Sekhari et al. (2021))**).** *Let $\epsilon, \delta \geq 0$. Let $S$ be a dataset of size $n$ drawn i.i.d. from $\mathcal{D}$. For a pair of learning and unlearning algorithms $\Omega, \bar{\Omega}$ that are $(\epsilon, \delta)$-certified unlearning, the deletion capacity $m_{\epsilon,\delta}^{\Omega,\bar{\Omega}}(d, n)$ is defined as the maximum number of samples $U$ to be forgotten, while still ensuring an excess population risk is $\gamma$. Specifically,*

$$m_{\epsilon,\delta,\gamma}^{\Omega,\bar{\Omega}}(d, n) := \max \left\{ m \mid \mathbb{E} \left[ \max_{U \subseteq \mathcal{S}: |U| \leq m} F(\bar{\Omega}(\Omega(\mathcal{S}); T(\mathcal{S})) - F(\mathbf{w}^*) \right] \leq \gamma \right\}, \quad (17)$$

*where the expectation above is with respect to $S \sim \mathcal{D}^n$ and output of the algorithms $\Omega$ and $\bar{\Omega}$.*

Similar to Sekhari et al. (2021), we provide upper bound and lower bound of deletion capacity for our algorithm. We first give the upper bound of deletion capacity.

**Theorem 7 (Upper Bound of Deletion Capacity** Sekhari et al. (2021)**).** *Let $\delta \leq 0.005$ and $\epsilon = 1$. There exists a 4-Lipschitz and 1-strongly convex loss function $f$, and a distribution $\mathcal{D}$ such that for any learning algorithm $A$ and removal mechanism $M$ that satisfies $(\epsilon, \delta)$-unlearning and has access to all remaining samples $\mathcal{S}\backslash U$, then the deletion capacity is:*

$$m_{\epsilon,\delta,\gamma}^{A,M}(d,n) \leq cn \tag{18}$$

*where $c$ depends on the properties of function $f$ and is strictly less than 1.*

A comprehensive proof of Theorem 7 is available in Sekhari et al. (2021). Furthermore, based on Theorem 6, we present the lower bound of deletion capacity below:

**Theorem 8 (Deletion Capacity Guarantee).** *There exists a learning algorithm $\Omega$ and an unlearning algorithm $\bar{\Omega}$ such that for any convex (and hence strongly convex), L-Lipschitz, and M-Smoothness loss $f$ and distribution $\mathcal{D}$. Choose step size $\eta \leq \frac{2}{M+\lambda}$ and $\mathbf{w}_{E,B}^{-U}$ is the empirical risk minimizer of objective function $F_{\mathcal{S}\backslash U}(\mathbf{w})$, then we have*

$$m_{\epsilon,\delta,\gamma}^{\Omega,\bar{\Omega}}(d,n) = \mathcal{O}\left((1/\rho)^n/d^{\frac{1}{2}}\right). \tag{19}$$

*Proof of Theorem 8.* Based on Theorem 6, we have

$$F(\tilde{\mathbf{w}}_{E-1,B}^{-U}) - \mathbb{E}[F(\mathbf{w}^*)] = \mathcal{O}\left(\frac{4L^2}{\lambda(n-m)} + \rho^{nE/|\mathcal{B}|}\frac{2L^2}{\lambda} + \rho^{nE/|\mathcal{B}|}\frac{2L\eta G\sqrt{d}\sqrt{\ln(1/\delta)}}{\epsilon}\right). \tag{20}$$

Note that $\rho < 1$. The above upper bound on the excess risk thus implies that we can delete at least

$$m = \mathcal{O}\left(\frac{\epsilon}{\eta\sqrt{\ln(1/\delta)}d^{\frac{1}{2}}\rho^n}\right) \tag{21}$$

samples while still ensuring an excess risk guarantee of $\gamma$. Accordingly, it yields an order of $\mathcal{O}\left((1/\rho)^n/d^{\frac{1}{2}}\right)$. Therefore, we have completed the proof. $\qquad\square$

### B.3 HESSIAN-VECTOR PRODUCTS TECHNIQUE IN STAGE II OF ALGORITHM 1

We provide the detailed Hessian-Vector Products (HVP) procedure in Algorithm 2. Notably, when recursively computing the product of the matrix and an arbitrary vector $(\mathbf{I} - \eta\mathbf{H})\mathbf{v}$, we can derive it from $\mathbf{v} - \eta\mathbf{H}\mathbf{v}$. Here, $\mathbf{H}\mathbf{v}$ involves a HVP through modern automatic differentiation frameworks such as JAX or PyTorch. For more explanation of the HVP, please refer to Dagréou et al. (2024).

---

**Algorithm 2:** Matrix-Vector Products in Stage II of Algorithm 1

---

1 **Stage II: Pre-computing and Pre-storing** statistics $\mathcal{T}(\mathcal{S}) = \{\mathbf{a}_{E,B}^{-u_j}\}_{j=1}^n$:

2 initialization $\{\mathbf{a}_{E,B}^{-u_j}\}_{j=1}^n = 0$, $\{\mathbf{v}_{e,b}^{-u_j}\}_{j=1}^n = 0$;

3 **for** $e = 0, \cdots, E-1, E$ **do**

4     **for** $b = 0, \cdots, B-1, B$ **do**

5         **for** $j = 1, 2 \cdots, n$ **do**

6             Compute: $\mathbf{a}_{e,b}^{-u_j} \leftarrow (\mathbf{I} - \eta_{e,b}\mathbf{H}_{e,b})\mathbf{a}_{e,b}^{-u_j} + \frac{\eta_{e,b}}{|\mathcal{B}_{e,b}|}(\mathbf{I} - \eta_{e,b}\mathbf{H}_{e,b})\mathbf{v}_{e,b}^{-u_j}$

7             **if** *gradient of sample $u_j$ computed in this update step* **then**

8                 $\mathbf{v}_{e,b}^{-u_j} \leftarrow \nabla\ell(\mathbf{w}_{e,b}^{-u_j}; z_i)$

9             **else**

10                 $\mathbf{v}_{e,b}^{-u_j} \leftarrow 0$

11             **end**

12         **end**

13     **end**

14 **end**

---

### B.4 Unlearning-Repairing Strategy in Stage III of Algorithm 1

While our approach does not require accessing any datasets in Stage III, we can mitigate the performance degradation caused by excessive deletions and make our algorithm more robust through a simple fine-tuning strategy if we can still access the remaining dataset in Stage III. We only consider this as a preliminary attempt in this section; other experiments do not include this Finetune strategy. This strategy allows our method to be used as a heuristic algorithm in non-privacy scenarios, without the need for theoretical certified guarantees, such as in bias removal.

Specifically, when the continuous incoming deletion requests to forget data, we can rapidly implement data removal through vector addition. This involves augmenting the current model with approximators, where errors accumulate during this process and thus cause performance degradation. Suppose, upon the arrival of a new round of deletion requests for a particular model and dataset, the model's accuracy significantly drops due to the removal of a substantial amount of data. As a remedy in Algorithm 3, we can address this performance degradation by fine-tuning the current model for $E_r$ epochs on the remaining dataset, referred to as the 'repairing' process. Additionally, by computing the approximators for the remaining dataset during the repairing process, and adding these approximators during the repairing process to the pre-stored approximators during the learning process, we obtain the new set of approximators for the remaining data. In Figure 3, we conduct an experiment on ResNet-18 using the CelebA dataset to demonstrate the efficacy of the repairing strategy.

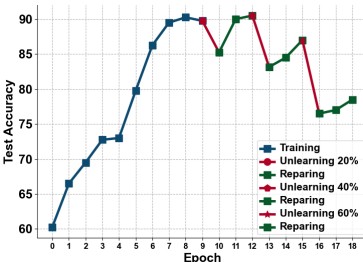

Figure 3: **Unlearning-Repairing Strategy**. The blue, red, and green lines respectively represent the learning, unlearning, and repairing process. When a significant amount of data is removed, leading to a loss in model performance, performing fine-tuning on the remaining dataset and recording the approximators during this period can help avoid performance degradation.

---

**Algorithm 3:** Hessian-Free Unlearning-Repairing Strategy in Stage III of Algorithm 1

---

1   **Stage III: Unlearning** when user requests to forget the subset $U=\{u_j\}_{j=1}^m$ on model $\mathbf{w}_{E,B}$:

2   Compute: $\bar{\mathbf{w}}_{E,B}^{-U} \leftarrow \mathbf{w}_{E,B}+\mathbf{a}_{E,B}^{-U}, \ \ c \leftarrow \|\Delta_{E,B}^{-U}\|\frac{\sqrt{2\ln(1.25/\delta)}}{\epsilon}$   Delete statistics: $\{\mathbf{a}_{E,B}^{-u_j}\}_{j=1}^m$

3   Sample: $\mathbf{n} \sim \mathcal{N}(0, \sigma\mathbf{I})$, Return: $\tilde{\mathbf{w}}_{E,B}^{-U} = \bar{\mathbf{w}}_{E,B}^{-U} + \mathbf{n}$

4   **Reparing** (Fine-tuning) the model $\tilde{\mathbf{w}}_{E,B}^{-U}$ on the remaining dataset $S\backslash U = \{z_r\}_{r=1}^{n-m}$:

5   **for** $e = 0, ..., E_r$ **do**

6      **for** $b = 0, 1 \cdots, B$ **do**

7          Compute gradient: $\mathbf{g} \leftarrow \frac{1}{|\mathcal{B}_{e,b}|}\sum_{r\in\mathcal{B}_{e,b}}\nabla\ell\left(\mathbf{w}_{e,b}; z_r\right),$

8          Gradient descent: $\mathbf{w}_{e,b+1} \leftarrow \mathbf{w}_{e,b} - \eta_{e,b}\mathbf{g}$

9      **end**

10   **end**

11   **Pre-computing** statistics $\mathcal{T}(\mathcal{S}) = \{\mathbf{a}_{E,B}^{-u_r}\}_{r=1}^{n-m}$ of the remaining dataset $S\backslash U = \{z_r\}_{r=1}^{n-m}$:

12   **for** $r = 1, 2 \cdots, n-m$ **do**

13      Recursive computation by using HVP based on Algorithm 2:

14      $\mathbf{a}_{E+E_r,B}^{-u_r} \leftarrow \sum_{e=0}^{E_r}\mathbf{M}_{e,b(u_r)} \cdot \nabla\ell(\mathbf{w}_{e,b(u_r)}; u) + \hat{\mathbf{H}}_{E_r,B-1\to0,b(u_r)+1} \cdot \mathbf{a}_{E,B}^{-u_r}$

15      where: $\mathbf{M}_{e,b(u_r)} = \frac{\eta_{e,b(u_r)}}{|\mathcal{B}_{e,b(u_r)}|}\hat{\mathbf{H}}_{E,B-1\to e,b(u_r)+1}$

16   **end**

---

## C PROOFS

### C.1 DETAILED ANALYSIS OF EQUATION (11)

Before proceeding with the analysis of our method, we will first provide the analysis of previous works (Suriyakumar et al. (2022); Liu et al. (2023); Mahadevan & Mathioudakis (2021); Chien et al. (2022); Liu et al. (2023)) and discuss the circumstances that lead to its failure.

We first assume that the empirical risk in Equation (2) is twice-differentiable and strictly convex in the parameter space $\mathcal{W}$. The original empirical risk minimizer can be expressed as

$$\hat{\mathbf{w}} := \underset{\mathbf{w} \in \mathcal{W}}{\arg\min} \quad F_{\mathcal{S}} = \underset{\mathbf{w} \in \mathcal{W}}{\arg\min} \quad \frac{1}{n} \sum_{i=1}^{n} \ell(\mathbf{w}; z_i). \tag{22}$$

We reweight the sample $u$ in the original empirical risk with a weighting factor $\omega$,

$$\hat{\mathbf{w}}^{-u} := \underset{\mathbf{w} \in \mathcal{W}}{\arg\min} \quad \frac{1}{n} \sum_{i=1}^{n} \ell(\mathbf{w}; z_i) + \omega \ell(\mathbf{w}; u),$$

$$\text{where} \quad \omega = -\frac{1}{n}. \tag{23}$$

When weighting factor $\omega = -\frac{1}{n}$, Equation (23) is equivalent to the empirical risk minimizer without samle $u$, i.e. removing a training point $u$ is similar to up-weighting its corresponding weight by $\omega = -\frac{1}{n}$. Following the classical result of Weisberg & Cook (1982), Koh & Liang (2017) aimed to approximate $\hat{\mathbf{w}}^{-u}$ by the first-order Taylor expansion around the optimal learned model $\hat{\mathbf{w}}$, given by,

$$\hat{\mathbf{w}}^{-u} - \hat{\mathbf{w}} \approx -\frac{1}{n} \left. \frac{d\hat{\mathbf{w}}^{-u}}{d\omega} \right|_{\omega=0} = \left( \frac{1}{n} \sum_{i=1}^{n} \nabla^2 \ell(\hat{\mathbf{w}}; z_i) \right)^{-1} \nabla \ell(\hat{\mathbf{w}}; u). \tag{24}$$

However, we note that these works rely on several key assumptions: the convexity of the objective function to ensure the invertibility of the Hessian matrix, and the optimality of the learned model. These assumptions are often stringent and may not be satisfied in practice, particularly in modern machine learning scenarios where the objective function may be non-convex and the empirical risk minimizer may not be attainable. In addition, in over-parameterized models, the computation of the Hessian matrix also requires a significant memory overhead.

In fact, aside from the optimal model, the training process contains more information, and our method aims to utilize the information from the training process to address both issues simultaneously. Here, we provide a more detailed explanation for Equation (11). Specifically, we first consider the impact of sample $u$ during the $e$-th epoch, and then extend the analysis to all epochs. It should be noted that a necessary setting for our method is that the retraining and learning process have the same initialization, model architecture, and training process.

***Analysis of Equation (11).*** For ease of understanding, each sample appears only in one (mini-)batch per epoch. The update rule for the model parameters is typically given by,

$$\mathbf{w}_{E,B+1} \leftarrow \mathbf{w}_{E,B} - \frac{\eta_{E,B}}{|\mathcal{B}_{E,B}|} \sum_{i \in \mathcal{B}_{E,B}} \nabla \ell(\mathbf{w}_{E,B}; z_i). \tag{25}$$

Without loss of generality, for the $e$-th epoch, we define the sample $u$ to be sampled in the $e$-th epoch's $b(u)$-th update. Reweighting the sample $u$ **in the each update process (rather than the empirical risk)** with a weighting factor $\omega$, Equation (3) and Equation (4) can be represented by

$$\mathbf{w}_{E,B+1} \leftarrow \mathbf{w}_{E,B} - \frac{\eta_{E,B}}{|\mathcal{B}_{E,B}|} \sum_{i \in \mathcal{B}_{E,B}} \left( \nabla \ell(\mathbf{w}_{E,B}; z_i) + \mathbf{1}_{b=b(u)} \omega \nabla \ell(\mathbf{w}_{E,B}; u) \right). \tag{26}$$

where $\mathbf{1}_{b=b(u)}$ is an indicator function that equals 1 if the condition $b = b(u)$ is true (that is, sample $u$ is to be involved (sampled) in the $b(u)$-th update of the $e$-th epoch), and 0 otherwise. We define that when $\omega = 0$, Equation (26) represents the original learning update, whereas $\omega = -1$ represents the

update after forgetting training sample $u$. To obtain the difference between the learning model and retraining model, we calculate the derivative of $\mathbf{w}_{E,B+1}$ w.r.t the weighting factor at $\omega = 0$, given by,

$$\mathbf{w}_{E,B+1}^{-u} - \mathbf{w}_{E,B+1} = \mathbf{w}_{E,B+1}\big|_{\omega=-1} - \mathbf{w}_{E,B+1}\big|_{\omega=0} \approx -\frac{d\mathbf{w}_{E,B+1}}{d\omega}\bigg|_{\omega=0}. \tag{27}$$

Based on the chain rule, we have,

$$\frac{d\mathbf{w}_{E,B+1}}{d\omega}\bigg|_{\omega=0} = \frac{d\mathbf{w}_{E,B+1}}{d\mathbf{w}_{E,B}}\frac{d\mathbf{w}_{E,B}}{d\omega}\bigg|_{\omega=0}. \tag{28}$$

Let $\mathbf{H}_{e,b} = 1/|\mathcal{B}_{e,b}|\sum_{i\in\mathcal{B}_{e,b}}\nabla^2\ell(\mathbf{w}_{e,b}; z_i)$ denote the Hessian matrix of the loss function. For Equation (26), we have that $\frac{d\mathbf{w}_{E,B+1}}{d\mathbf{w}_{E,B}} = \mathbf{I} - \eta_{E,B}\mathbf{H}_{E,B}$. Substituting into Equation (28),

$$\frac{d\mathbf{w}_{E,B+1}}{d\omega}\bigg|_{\omega=0} = (\mathbf{I} - \eta_{E,B}\mathbf{H}_{E,B})\frac{d\mathbf{w}_{E,B}}{d\omega}\bigg|_{\omega=0}, \tag{29}$$

which is equivalent to Equation (8) in the main text (*Case* 1), that is,

$$\mathbf{w}_{e,b+1}^{-u} - \mathbf{w}_{e,b+1} \approx (\mathbf{I} - \eta_{e,b}\mathbf{H}_{e,b})(\mathbf{w}_{e,b}^{-u} - \mathbf{w}_{e,b}). \tag{30}$$

Recursively applying the chain rule until $\mathbf{1}_{b=b(u)} = 1$ during $E$-th epoch,

$$\frac{d\mathbf{w}_{E,B+1}}{d\omega}\bigg|_{\omega=0} = \prod_{b=B}^{b(u)+1}(\mathbf{I} - \eta_{E,b}\mathbf{H}_{E,b})\frac{d\mathbf{w}_{E,b(u)+1}}{d\omega}\bigg|_{\omega=0}. \tag{31}$$

When the indicator function $\mathbf{1}_{b=b(u)} = 1$, for Equation (26), we obtain,

$$\frac{d\mathbf{w}_{E,b(u)+1}}{d\omega}\bigg|_{\omega=0} = \frac{d\mathbf{w}_{E,b(u)}}{d\omega}\bigg|_{\omega=0} - \eta_{E,b(u)}\mathbf{H}_{E,b(u)}\frac{d\mathbf{w}_{E,b(u)}}{d\omega}\bigg|_{\omega=0} - \frac{\eta_{E,b}}{|\mathcal{B}_{E,b}|}\nabla\ell\left(\mathbf{w}_{E,b(u)}; u\right), \tag{32}$$

which is equivalent to Equation (9) in the main text (*Case* 2), that is,

$$\mathbf{w}_{e,b(u)+1}^{-u} - \mathbf{w}_{e,b(u)+1} \approx (\mathbf{I} - \eta_{e,b(u)}\mathbf{H}_{e,b(u)})(\mathbf{w}_{e,b(u)}^{-u} - \mathbf{w}_{e,b(u)}) + \frac{\eta_{e,b(u)}}{|\mathcal{B}_{e,b(u)}|}\nabla\ell(\mathbf{w}_{e,b(u)}; u). \tag{33}$$

Substituting Equation (32) into Equation (31),

$$\frac{d\mathbf{w}_{E,B+1}}{d\omega}\bigg|_{\omega=0} = \prod_{b=B}^{b(u)}(\mathbf{I} - \eta_{E,b}\mathbf{H}_{E,b})\frac{d\mathbf{w}_{E,b(u)}}{d\omega}\bigg|_{\omega=0} - \frac{\eta_{e,b}}{|\mathcal{B}_{e,b}|}\prod_{b=B}^{b(u)+1}(\mathbf{I} - \eta_{E,b}\mathbf{H}_{E,b})\nabla\ell\left(\mathbf{w}_{E,b(u)}; u\right). \tag{34}$$

Therefore, the impact of sample $u$ on the model update trajectory during the $E$-th round is,

$$\begin{aligned} \mathbf{w}_{E,B}^{-u} - \mathbf{w}_{E,B} \approx{} & \prod_{b=B-1}^{0}(\mathbf{I} - \eta_{E,b}\mathbf{H}_{E,b})(\mathbf{w}_{E,0}^{-u} - \mathbf{w}_{E,0}) \\ & + \prod_{b=B-1}^{b(u)+1}(\mathbf{I} - \eta_{E,b}\mathbf{H}_{E,b})\frac{\eta_{E,b(u)}}{|\mathcal{B}_{E,b(u)}|}\nabla\ell(\mathbf{w}_{E,b(u)}; u). \end{aligned} \tag{35}$$

Since initially the retraining model $\mathbf{w}_{0,0}^{-u}$ and the learning model $\mathbf{w}_{0,0}$ are the same, the first term of Equation (35) will recursively reduce to the initial model and will equal 0, and as sample $u$ participates in updates once per epoch, the second term is the sum over $E$ epochs.[5] Apply the above process recursively to complete the proof and, we obtain the approximator for sample $u$.

$$\mathbf{a}_{E,B}^{-u} := \sum_{e=0}^{E}\mathbf{M}_{e,b(u)}\cdot\nabla\ell(\mathbf{w}_{e,b(u)}; u), \text{ where } \mathbf{M}_{e,b(u)} = \frac{\eta_{e,b(u)}}{|\mathcal{B}_{e,b(u)}|}\hat{\mathbf{H}}_{E,B-1\to e,b(u)+1}. \tag{36}$$

We define $\hat{\mathbf{H}}_{E,B-1\to e,b(u)+1} = (\mathbf{I} - \eta_{E,B-1}\mathbf{H}_{E,B-1})...(\mathbf{I} - \eta_{e,b(u)+1}\mathbf{H}_{e,b(u)+1})$, which means that the multiplication should begin with recent update and continue backward through previous updates.

$\square$

---

[5] Note that we assume sample $u$ participates in the gradient calculation exactly once per epoch. Therefore, in Equation (36), the number of summations $E$ represents that sample $u$ has participated in $E$ gradient calculations. In general, due to the randomness of sampling, sample $u$ may be sampled multiple times or not at all within a single epoch. Nevertheless, we can derive the results for these situations in the same manner. To facilitate understanding and draw relevant conclusions, all results in this work are based on the simplified assumption.

## C.2 DETAILED PROOF OF THEOREM 1

Now, we demonstrate that in the case of continuous arrival of deletion requests, our algorithm is capable of streaming removing samples. Intuitively, this property can be mainly explained by Taylor's linear expansion. For ease of exposition, we simplify our proof to a scenario with only two samples, $u_1$ and $u_2$. That is, our goal is to prove that when $u_1$ initiates a deletion request and the algorithm is executed, the arrival of a deletion request from $u_2$ after $u_1$ (*Streaming Deletion*), is fully equivalent to the simultaneous execution of deletion requests from both $u_1$ and $u_2$ (*Batch Deletion*).

Without loss of generality, we assume that $b(u_1) < b(u_2)$. Based on Equation (11), the approximators for $u_1$ and $u_2$ are denoted as $\mathbf{a}_{E,B}^{-u_1}$ and $\mathbf{a}_{E,B}^{-u_2}$, respectively, as follows:

$$
\begin{aligned}
\mathbf{a}_{E,B}^{-u_1} &:= \sum_{e=0}^{E} \mathbf{M}_{e,b(u_1)} \cdot \nabla\ell(\mathbf{w}_{e,b(u_1)}; u_1), \text{ where } \mathbf{M}_{e,b(u_1)} = \frac{\eta_{e,b(u_1)}}{|\mathcal{B}_{e,b(u_1)}|}\hat{\mathbf{H}}_{E,B-1\to e,b(u_1)+1}, \\
\mathbf{a}_{E,B}^{-u_2} &:= \sum_{e=0}^{E} \mathbf{M}_{e,b(u_2)} \cdot \nabla\ell(\mathbf{w}_{e,b(u_2)}; u_2), \text{ where } \mathbf{M}_{e,b(u_2)} = \frac{\eta_{e,b(u_2)}}{|\mathcal{B}_{e,b(u_2)}|}\hat{\mathbf{H}}_{E,B-1\to e,b(u_2)+1}
\end{aligned}
\tag{37}
$$

When we simultaneously delete $u_1$ and $u_2$, the difference is,

$$
\mathbf{w}_{e,b+1}^{-u_2-u_1} - \mathbf{w}_{e,b+1} \leftarrow \mathbf{w}_{e,b}^{-u_2-u_1} - \mathbf{w}_{e,b} - \frac{\eta_{e,b}}{|\mathcal{B}_{e,b}|} \sum_{i\in\mathcal{B}_{e,b}} \left( \nabla\ell(\mathbf{w}_{e,b}^{-u_2-u_1}; z_i) - \nabla\ell(\mathbf{w}_{e,b}; z_i) \right).
\tag{38}
$$

Consider the Taylor expansion during the $E$-th epoch, we have,

$$
\mathbf{w}_{E,b+1}^{-u_2-u_1} - \mathbf{w}_{E,b+1} \approx (\mathbf{I} - \frac{\eta_{E,b}}{|\mathcal{B}_{E,b}|}\mathbf{H}_{E,b})(\mathbf{w}_{E,b}^{-u_2-u_1} - \mathbf{w}_{E,b}).
\tag{39}
$$

When $u_1$ and $u_2$ are removed respectively, we have the following approximation:

$$
\begin{aligned}
\mathbf{w}_{E,b(u_2)+1}^{-u_2-u_1} - \mathbf{w}_{E,b(u_2)+1} &\approx (\mathbf{I} - \frac{\eta_{E,b(u_2)}}{|\mathcal{B}_{E,b(u_2)}|}\mathbf{H}_{E,b(u_2)})(\mathbf{w}_{E,b(u_2)}^{-u_2-u_1} - \mathbf{w}_{E,b(u_2)}) \\
&\quad + \underbrace{\frac{\eta_{E,b(u_2)}}{|\mathcal{B}_{E,b(u_2)}|}\nabla\ell(\mathbf{w}_{E,b(u_2)}; u_2)}_{\text{Approximate impact of } u_2 \text{ in } e\text{-th epoch}}, \\
\mathbf{w}_{E,b(u_1)+1}^{-u_2-u_1} - \mathbf{w}_{E,b(u_1)+1} &\approx (\mathbf{I} - \frac{\eta_{E,b(u_1)}}{|\mathcal{B}_{E,b(u_1)}|}\mathbf{H}_{E,b(u_1)})(\mathbf{w}_{E,b(u_1)}^{-u_2-u_1} - \mathbf{w}_{E,b(u_1)}) \\
&\quad + \underbrace{\frac{\eta_{E,b(u_2)}}{|\mathcal{B}_{E,b(u_2)}|}\nabla\ell(\mathbf{w}_{E,b(u_2)}; u_2)}_{\text{Approximate impact of } u_2 \text{ in } e\text{-th epoch}} + \underbrace{\frac{\eta_{E,b(u_1)}}{|\mathcal{B}_{E,b(u_1)}|}\nabla\ell(\mathbf{w}_{E,b(u_1)}; u_1)}_{\text{Approximate impact of } u_1 \text{ in } e\text{-th epoch}}.
\end{aligned}
\tag{40}
$$

Therefore, in the $E$-th epoch, we have the following affine stochastic recursion,

$$
\begin{aligned}
\mathbf{w}_{E,B}^{-u_2-u_1} - \mathbf{w}_{E,B} &\approx \prod_{b=B-1}^{0} (\mathbf{I} - \frac{\eta_{E,b}}{|\mathcal{B}_{E,b}|}\mathbf{H}_{E,b})(\mathbf{w}_{E,0}^{-u_2-u_1} - \mathbf{w}_{E,0}) + \\
&\prod_{b=B-1}^{b(u_1)+1}(\mathbf{I}-\frac{\eta_{E,b}}{|\mathcal{B}_{E,b}|}\mathbf{H}_{E,b})\frac{\eta_{E,b(u_1)}}{|\mathcal{B}_{E,b(u_1)}|}\nabla\ell(\mathbf{w}_{E,b(u_1)};u_1) + \prod_{b=B-1}^{b(u_2)+1}(\mathbf{I}-\frac{\eta_{E,b}}{|\mathcal{B}_{E,b}|}\mathbf{H}_{E,b})\frac{\eta_{E,b(u_2)}}{|\mathcal{B}_{E,b(u_2)}|}\nabla\ell(\mathbf{w}_{E,b(u_2)};u_2).
\end{aligned}
\tag{41}
$$

Apply it recursively to complete the proof, and we can simultaneously compute the impact of $u_1$ and $u_2$ on training for all epochs.

$$
\mathbf{w}_{E,B}^{-u_2-u_1} - \mathbf{w}_{E,B} \approx \mathbf{a}_{E,B}^{-u} := \sum_{e=0}^{E} \mathbf{M}_{e,b(u_1)} \cdot \nabla\ell(\mathbf{w}_{e,b(u_1)}; u_1) + \sum_{e=0}^{E} \mathbf{M}_{e,b(u_2)} \cdot \nabla\ell(\mathbf{w}_{e,b(u_2)}; u_2) + .
\tag{42}
$$

According to Eqs. (37) and (42), we have completed the proof of $\mathbf{a}_{E,B}^{-u_2-u_1} = \mathbf{a}_{E,B}^{-u_1} + \mathbf{a}_{E,B}^{-u_2}$.

### C.3 DETAILED PROOF OF LEMMA 2

**Proof of Lemma 2.** Now we begin the proof that the remainder term $o(\mathbf{w}_{e,b}^{-u} - \mathbf{w}_{e,b})$ in the $e$-th epoch, $b$-th round is bounded. Specifically, recalling Equation (4), we have,

$$\mathbf{w}_{e,b+1}^{-u} - \mathbf{w}_{e,b+1} \approx \mathbf{w}_{e,b}^{-u} - \mathbf{w}_{e,b} - \frac{\eta_{e,b}}{|\mathcal{B}_{e,b}|} \sum_{i \in \mathcal{B}_{e,b}} \left( \nabla\ell(\mathbf{w}_{e,b}^{-u}; z_i) - \nabla\ell(\mathbf{w}_{e,b}; z_i) \right),$$

$$\begin{aligned}
\mathbf{w}_{e,b(u)+1}^{-u} - \mathbf{w}_{e,b(u)+1} &\approx \mathbf{w}_{e,b(u)}^{-u} - \mathbf{w}_{e,b(u)} \\
&\quad - \frac{\eta_{e,b(u)}}{|\mathcal{B}_{e,b(u)}|} \sum_{i \in \mathcal{B}_{e,b(u)} \setminus \{u\}} \left( \nabla\ell(\mathbf{w}_{e,b(u)}^{-u}; z_i) - \nabla\ell(\mathbf{w}_{e,b(u)}; z_i) \right) + \frac{\eta_{e,b(u)}}{|\mathcal{B}_{e,b(u)}|} \nabla\ell(\mathbf{w}_{e,b(u)}; u).
\end{aligned} \tag{43}$$

For all $e, b$ and $z \in \mathcal{B}_{e,b}$, suppose that exists $G$ such that $G = \max \|\nabla\ell(\mathbf{w}_{e,b}; z)\| < \infty$.

$$\|\mathbf{w}_{e,b}^{-u} - \mathbf{w}_{e,b}\| \leq \|\mathbf{w}_{e,b-1}^{-u} - \mathbf{w}_{e,b-1}\| + 2\eta_{e,b-1}G. \tag{44}$$

Consider a simple geometrically step size decay strategy $\eta_{e,b} = q\eta_{e,b-1}$, where $0 < q < 1$ is the decay rate, and let $\eta$ be the initial stepsize.[6] Recalling that $t = eB + b$, then we get

$$\|\mathbf{w}_{e,b}^{-u} - \mathbf{w}_{e,b}\| \leq \|\mathbf{w}_{e,b-1}^{-u} - \mathbf{w}_{e,b-1}\| + 2\eta G q^{t-1}. \tag{45}$$

Note that the initial retraining model and learning model are identical. Therefore, based on the above recursion, we can obtain the geometric progression as follows,

$$\begin{aligned}
\|\mathbf{w}_{e,b}^{-u} - \mathbf{w}_{e,b}\| &\leq 2\eta G q^{t-1} + 2\eta G q^{t-2} + \dots + 2\eta G q^0 \\
&= 2\eta G \sum_{k=0}^{t-1} q^k \\
&= 2\eta G \frac{q^t - 1}{q - 1}.
\end{aligned} \tag{46}$$

Therefore, we have completed the proof of Lemma 2. $\qquad \square$

### C.4 DETAILED PROOF OF THEOREM 4

**Proof of Theorem 4.** Let us recall the approximator of Equation (11) in Section 3, we have,

$$\mathbf{a}_{E,B}^{-u} := \sum_{e=0}^{E} \mathbf{M}_{e,b(u)} \cdot \nabla\ell(\mathbf{w}_{e,b(u)}; u), \quad \text{where } \mathbf{M}_{e,b(u)} = \frac{\eta_{e,b(u)}}{|\mathcal{B}_{e,b(u)}|} \hat{\mathbf{H}}_{E,B-1 \to e,b(u)+1}. \tag{47}$$

For the $E$-th epoch' $B$-th update, we have the following Taylor approximation,

$$\begin{aligned}
\mathbf{w}_{E,B}^{-u} - \mathbf{w}_{E,B} - \mathbf{a}_{E,B}^{-u} &= (\mathbf{I} - \eta_{E,B-1}\mathbf{H}_{E,B-1})(\mathbf{w}_{E,B-1}^{-u} - \mathbf{w}_{E,B-1}) \\
&\quad + \eta_{E,B-1} o(\mathbf{w}_{E,B-1}^{-u} - \mathbf{w}_{E,B-1}) - \mathbf{a}_{E,B}^{-u}.
\end{aligned} \tag{48}$$

For convenience, we define $\Delta\mathbf{w}_{E,B}^{-u} = \mathbf{w}_{E,B}^{-u} - \mathbf{w}_{E,B}$, and $\hat{\mathbf{H}}_{E,B-1} = \mathbf{I} - \eta_{E,B-1}\mathbf{H}_{E,B-1}$. Here, we begin to analyze the error generated by each update,

$$\Delta\mathbf{w}_{E,B}^{-u} - \mathbf{a}_{E,B}^{-u} = \hat{\mathbf{H}}_{E,B-1}\Delta\mathbf{w}_{E,B-1}^{-u} + \eta_{E,B-1} o(\Delta\mathbf{w}_{E,B-1}^{-u}) - \mathbf{a}_{E,B}^{-u}, \tag{49}$$

---

[6]Li et al. (2021) demonstrated that SGD with exponential stepsizes achieves (almost) optimal convergence rate for smooth non-convex functions, thus not affecting the convergence of learning process in our paper.

Furthermore, we observe that during the $E$-th epoch, Equation (49) can be obtained from affine stochastic recursion as follows,

$$
\begin{aligned}
\|\Delta \mathbf{w}_{E,B}^{-u} - \mathbf{a}_{E,B}^{-u}\| \leq \| &\prod_{b=0}^{B-1} \hat{\mathbf{H}}_{E,b} \Delta \mathbf{w}_{E,0}^{-u} - \mathbf{a}_{E-1,B}^{-u}\| \\
&+ \underbrace{\frac{\eta q^{T-B+b(u)}}{|\mathcal{B}_{E,b(u)}|} \| \prod_{b=b(u)+1}^{B-1} \hat{\mathbf{H}}_{E,b} \left( \nabla\ell(\mathbf{w}_{E,b(u)}^{-u}; u) - \nabla\ell(\mathbf{w}_{E,b(u)}; u)\right) \|}_{\text{Approximation error term } C_2 \text{ during the } E\text{-th epoch}} \\
&+ \underbrace{\eta q^{T-1}\|\Delta \mathbf{w}_{E,B-1}^{-u}\| + \eta q^{T-2}\|\hat{\mathbf{H}}_{E,B-1}\Delta \mathbf{w}_{E,B-2}^{-u}\| + ... + \eta q^{T-B}\| \prod_{b=0}^{B-1} \hat{\mathbf{H}}_{E,b}\Delta \mathbf{w}_{E,0}^{-u}\|}_{\text{Approximation error term } C_1 \text{ during the } E\text{-th epoch}}.
\end{aligned}
$$
(50)

Through the above process, we obtain the result of the $E$-the epoch approximation. Now let's bound the last two terms, i.e., the approximation error term $C_1$ and $C_2$ during the $E$-th epoch.

Recalling Lemma 2, we have that for all $t = eB + b$,

$$
\|\Delta \mathbf{w}_{e,b}^{-u}\| = \|\mathbf{w}_{e,b}^{-U} - \mathbf{w}_{e,b}\| \leq 2\eta G \frac{1 - q^t}{1 - q}.
$$
(51)

Recalling Lemma 3, we have that for all $e, b$ and $\rho$ is the spectral radius of $\mathbf{I} - \eta_{e,b}\mathbf{H}_{e,b}$,

$$
\|\hat{\mathbf{H}}_{e,b}\Delta \mathbf{w}_{e,b}^{-u}\| = \|(\mathbf{I} - \eta_{e,b}\mathbf{H}_{e,b})\Delta \mathbf{w}_{e,b}^{-u}\| \leq \rho\|\Delta \mathbf{w}_{e,b}^{-u}\| \leq \rho \cdot 2\eta G \frac{1 - q^t}{1 - q},
$$
(52)

Therefore, we can bound the approximation error terms $C_1$ and $C_2$ during the $E$-th epoch as follows:

**(1)** we first bound the first approximation error term $C_1$ during the $E$-th epoch. For clarity, We define $T$ as the total number of steps of SGD updates performed when obtaining $\mathbf{a}_{E,B}$, where $T = EB + B$.

$$
\begin{aligned}
q^{T-1}\|\Delta \mathbf{w}_{E,B-1}^{-u}\| &+ q^{T-2}\|\hat{\mathbf{H}}_{E,B-1}\Delta \mathbf{w}_{E,B-2}^{-u}\| + ... + q^{T-B}\| \prod_{b=0}^{B-1} \hat{\mathbf{H}}_{E,b}\Delta \mathbf{w}_{E,0}^{-u}\| \\
&\leq 2\eta G \frac{q^{T-1} - q^{2(T-1)}}{1-q} + 2\eta G \frac{q^{T-2} - q^{2(T-2)}}{1-q}\rho + ... + 2\eta G \frac{q^{T-B} - q^{2(T-B)}}{1-q}\rho^{B-1} \\
&= \frac{2\eta G}{1-q} \sum_{k=T-1}^{T-B} \rho^{T-1-k}(q^k - q^{2k}).
\end{aligned}
$$
(53)

**(2)** we then bound the second approximation error term $C_2$ during the $E$-th epoch. Suppose the stochastic gradient norm $\|\nabla\ell(\mathbf{w}_{e,b}; u)\|$ is bounded by $G$, and we thus have the following upper bound,

$$
\frac{\eta q^{T-B+b(u)}}{|\mathcal{B}_{E,b(u)}|} \| \prod_{b=b(u)+1}^{B-1} \hat{\mathbf{H}}_{E,b}(\nabla\ell(\mathbf{w}_{E,b(u)}^{-u}; u) - \nabla\ell(\mathbf{w}_{E,b(u)}; u))\| \leq \frac{2G\eta}{|\mathcal{B}|} q^{T-B+b(u)}\rho^{B-b(u)-1}.
$$
(54)

Similarly, for each epoch, we have the aforementioned approximation error bound. Moreover, it is noted that when $e = 0$, the initial models for the retraining process and the learning process are same.

Therefore, we have the following overall approximate error across all epochs:

$$
\|\mathbf{w}_{E,B}^{-u} - \bar{\mathbf{w}}_{E,B}^{-u}\| \le \eta \left( q^{T-1}\|\Delta\mathbf{w}_{E,B-1}^{-u}\| + q^{T-2}\|\hat{\mathbf{H}}_{E,B-1}\Delta\mathbf{w}_{E,B-2}^{-u}\| + ... + q^0 \prod_{k=0}^{E}\prod_{b=0}^{B-1}\hat{\mathbf{H}}_{k,b}\Delta\mathbf{w}_{0,0}^{-u} \right)
$$

$$
+ \frac{\eta q^{T-B+b(u)}}{|\mathcal{B}_{E,b(u)}|}\|\prod_{b=b(u)+1}^{B-1}\hat{\mathbf{H}}_{E,b}(\nabla\ell(\mathbf{w}_{E,b(u)}^{-u};u) - \nabla\ell(\mathbf{w}_{E,b(u)};u))\| + ...
$$

$$
+ \frac{\eta q^{b(u)}}{|\mathcal{B}_{0,b(u)}|}\|\prod_{e=0}^{E}\prod_{b=b(u)+1}^{B-1}\hat{\mathbf{H}}_{E,b}(\nabla\ell(\mathbf{w}_{0,b(u)}^{-u};u) - \nabla\ell(\mathbf{w}_{0,b(u)};u))\|
$$

$$
\le 2\eta^2 G\frac{q^{T-1} - q^{2(T-1)}}{1-q} + 2\eta^2 G\frac{q^{T-2} - q^{2(T-2)}}{1-q}\rho + ... + 2\eta^2 C\frac{q^0 - q^{2\times(0)}}{1-q}\rho^{T-1}
$$

$$
+ \frac{2\eta G}{|\mathcal{B}|}q^{T-B+b(u)}\rho^{B-b(u)-1} + ... + \frac{2G\eta}{|\mathcal{B}|}q^{b(u)}\rho^{T-b(u)-1}
$$

$$
= \frac{2\eta^2 G}{1-q}\sum_{k=0}^{T-1}\rho^{T-k-1}(q^k - q^{2k}) + \frac{2\eta G}{|\mathcal{B}|}\sum_{e=0}^{E}\rho^{T-eB-b(u)-1}q^{eB+b(u)}.
$$

(55)

Suppose $q < \rho$. Through the polynomial multiplies geometric progression, we obtain,

$$
\|\mathbf{w}_{E,B}^{-u} - \bar{\mathbf{w}}_{E,B}^{-u}\| = \frac{2\eta^2 G}{1-q}\rho^{T-1}\sum_{k=0}^{T-1}\left((\frac{q}{\rho})^k - (\frac{q^2}{\rho})^k\right) + \frac{2\eta G}{|\mathcal{B}|}\rho^{T-b(u)-1}q^{b(u)}\sum_{e=0}^{E}(\frac{q}{\rho})^{eB}
$$

$$
= \frac{2\eta^2 G}{1-q}\left(\frac{\rho^T - q^T}{\rho - q} - \frac{\rho^T - q^{2T}}{\rho - q^2}\right) + \frac{2\eta G}{|\mathcal{B}|}\frac{\rho^T - q^T}{\rho^B - q^B}\rho^{B-b(u)-1}q^{b(u)},
$$

(56)

which yields an upper bound on the approximation error. For ease of comparison, we subsequently simplify the aforementioned result, as follows:

$$
\|\mathbf{w}_{E,B}^{-u} - \bar{\mathbf{w}}_{E,B}^{-u}\| \le \frac{2\eta^2 G}{1-q}\left(\frac{\rho^T - q^{2T}}{\rho - q} - \frac{\rho^T - q^{2T}}{\rho - q^2}\right) + \frac{2\eta G}{|\mathcal{B}|}\frac{\rho^T - q^{2T}}{\rho^B - q^B}\rho^{B-1}
$$

$$
\le 2\eta G(\rho^T - q^{2T})\left(\frac{\eta q}{(\rho - q)(\rho - q^2)} + \frac{1}{|\mathcal{B}|(\rho - q)}\right).
$$

(57)

At this point, we have completed the proof of Theorem 4 as follows,

$$
\|\mathbf{w}_{E,B}^{-u} - \bar{\mathbf{w}}_{E,B}^{-u}\| \le \mathcal{O}(2\eta G\rho^T).
$$

(58)

If Theorem 4 further satisfies Assumption 1, and we choose $\eta \le 2/(M + \lambda)$ to ensure gradient descent convergence as well as $\rho < 1$, we thus obtain a tight upper bound of $\mathcal{O}(\rho^n)$ for each sample. $\qquad\square$

## C.5 ADDITIONAL DEFINITIONS OF ASSUMPTION 1

We provide the following standard definitions of $\lambda$-strong convexity, $L$-Lipschitzness, $M$-Smoothness.

**Definition 3** ($\lambda$-Strong convexity). *The loss function $\ell(\mathbf{w}; z)$ is $\lambda$-strongly convex with respect to $\mathbf{w}$, i.e., there exists a constant $\lambda > 0$ such that for any $z \in \mathcal{Z}$ and $\mathbf{w}_1$, $\mathbf{w}_2 \in \mathbb{R}^d$, it holds that*

$$
\ell(\mathbf{w}_1; z) \ge \ell(\mathbf{w}_2; z) + \langle\nabla\ell(\mathbf{w}_2; z), \mathbf{w}_1 - \mathbf{w}_2\rangle + \frac{\lambda}{2}\|\mathbf{w}_1 - \mathbf{w}_2\|^2,
$$

(59)

*or equivalently, $\|\nabla^2\ell(\mathbf{w}; z)\| \ge \lambda$ for all $\mathbf{w} \in \mathbb{R}^d$.*

**Definition 4** ($L$-Lipschitzness). *The loss function $\ell(\mathbf{w}; z)$ is $L$-Lipschitz with respect to $\mathbf{w}$, i.e., there exists a constant $L > 0$ such that for any $z \in \mathcal{Z}$ and $\mathbf{w}_1$, $\mathbf{w}_2 \in \mathbb{R}^d$, it holds that*

$$
|\ell(\mathbf{w}_1; z) - \ell(\mathbf{w}_2; z)| \le L\|\mathbf{w}_1 - \mathbf{w}_2\|.
$$

(60)

*or equivalently, $\|\nabla\ell(\mathbf{w}; z)\| \le L$ for all $\mathbf{w} \in \mathbb{R}^d$.*

**Definition 5** ($M$-Smoothness). *The loss function $\ell(\mathbf{w}; z)$ is $M$-smooth with respect to $\mathbf{w}$, i.e., there exists a constant $M > 0$ such that for any $z \in \mathcal{Z}$ and $\mathbf{w}_1$, $\mathbf{w}_2 \in \mathbb{R}^d$, it holds that*

$$
\|\nabla\ell(\mathbf{w}_1; z) - \nabla\ell(\mathbf{w}_2; z)\| \le M\|\mathbf{w}_1 - \mathbf{w}_2\|,
$$

(61)

*or equivalently, $\|\nabla^2\ell(\mathbf{w}; z)\| \le M$ for all $\mathbf{w} \in \mathbb{R}^d$.*

### C.6 DETAILED PROOF OF THEOREM 6

Before we commence with our proof, we present some necessary lemmata.

**Lemma 9** (Shalev-Shwartz et al. (2009)). *Let $\mathcal{S} = \{z_i\}_{i=1}^n$ where $S \sim \mathcal{D}^n$. Let $\mathbf{w}^*$ denote a minimizer of the population risk in Equation (1) and $\hat{\mathbf{w}}^{-U}$ denote a minimizer of the empirical risk without the knowledge of $U$ in Equation (2) which aims to minimize $F_{\mathcal{S}\backslash U}(\mathbf{w})$, where $|U| = m$. Let Assumption 1 hold, we have,*

$$\mathbb{E}\left[F\left(\hat{\mathbf{w}}^{-U}\right) - F\left(\mathbf{w}^*\right)\right] \leq \frac{4L^2}{\lambda(n-m)}. \tag{62}$$

*Proof of Lemma 9.* A comprehensive proof of Lemma 9 is available in Shalev-Shwartz et al. (2009). □

For brevity and to save space, we present our generalization guarantee only for GD. Nevertheless, the results for SGD can be analogously obtained by similar process.

**Lemma 10.** *For any $z \in \mathcal{Z}$, the loss function $\ell(\mathbf{w}; z)$ is $\lambda$-strongly convex, $L$-Lipschitz and $M$-smooth. Let $\hat{\mathbf{w}}^{-U} = \arg\min_{\mathbf{w}} F_{\mathcal{S}\backslash U}(\mathbf{w})$ without the knowledge of $U$ in problem Equation (2). For all $e, b$ and vector $\mathbf{v} \in \mathbb{R}^d$, the spectral radius of $\mathbf{I} - \eta_{e,b}\mathbf{H}_{e,b}$ is defined as $\rho$ which is largest absolute eigenvalue of these matrices. We have that after $T$ steps of GD with initial stepsize $\eta \leq \frac{2}{\lambda+M}$,*

$$\|\mathbf{w}_T^{-U} - \hat{\mathbf{w}}^{-U}\| \leq \rho^T \frac{2M}{\lambda}, \text{ where } \rho = \max\{|1-\eta\lambda|, |1-\eta M|\} < 1. \tag{63}$$

*Proof of Lemma 10.* Recalling that, according to the strong convexity and smoothness, $\lambda, M > 0$, we have $\lambda\mathbf{I} \preceq \nabla^2\ell(\mathbf{w}; z) \preceq M\mathbf{I}$. Therefore, we have

$$(1-\eta_T\lambda)\mathbf{I} \preceq \mathbf{I} - \frac{\eta_T}{|\mathcal{B}_T|}\sum_{i\in\mathcal{B}_T}\nabla^2\ell(\mathbf{w}_T^{-U}; z_i) \preceq (1-\eta_T M)\mathbf{I}. \tag{64}$$

Let $\eta \leq \frac{2}{\lambda+M}$. Therefore, $\eta_T \leq \frac{2}{\lambda+M}$, which implies that for any $T \in [0, \infty)$:

$$|1-\eta_T\lambda| < 1, \ |1-\eta_T M| < 1. \tag{65}$$

It is seen from the fundamental theorem of calculus that

$$\nabla F_{\mathcal{S}\backslash U}\left(\mathbf{w}_T^{-U}\right) = \nabla F_{\mathcal{S}\backslash U}\left(\mathbf{w}_T^{-U}\right) - \underbrace{\nabla F_{\mathcal{S}\backslash U}\left(\hat{\mathbf{w}}^{-U}\right)}_{=0} = \left(\int_0^1 \nabla^2 F_{\mathcal{S}\backslash U}\left(\mathbf{w}_\tau^{-U}\right) \mathrm{d}\tau\right)\left(\mathbf{w}_T^{-U} - \hat{\mathbf{w}}^{-U}\right), \tag{66}$$

where $\mathbf{w}_\tau^{-U} := \mathbf{w}_T^{-U} + \tau\left(\hat{\mathbf{w}}^{-U} - \mathbf{w}_T^{-U}\right)$. Here, $\{\mathbf{w}_\tau^{-U}\}_{0\leq\tau\leq 1}$ forms a line segment between $\mathbf{w}_T^{-U}$ and $\hat{\mathbf{w}}^{-U}$. Therefore, According to the GD update rule and based on Lemma 3, we have,

$$\begin{aligned}
\|\mathbf{w}_{T+1}^{-U} - \hat{\mathbf{w}}^{-U}\| &= \left\|\left(\mathbf{I} - \eta_T\int_0^1\nabla^2 F_{\mathcal{S}\backslash U}\left(\mathbf{w}_\tau^{-U}\right)\mathrm{d}\tau\right)\left(\mathbf{w}_T^{-U} - \hat{\mathbf{w}}^{-U}\right)\right\| \\
&\leq \sup_{0\leq\tau\leq 1}\left\|\mathbf{I} - \eta_T\nabla^2 F_{\mathcal{S}\backslash U}\left(\mathbf{w}_\tau^{-U}\right)\right\|\left\|\mathbf{w}_T^{-U} - \hat{\mathbf{w}}^{-U}\right\| \leq \rho\|\mathbf{w}_T^{-U} - \hat{\mathbf{w}}^{-U}\|.
\end{aligned} \tag{67}$$

Apply it recursively,

$$\|\mathbf{w}_T^{-U} - \hat{\mathbf{w}}^{-U}\| \leq \rho^T\|\mathbf{w}_0^{-U} - \hat{\mathbf{w}}^{-U}\|, \text{ where } \rho = \max\{|1-\eta\lambda|, |1-\eta M|\} < 1. \tag{68}$$

By the assumption that $\lambda$-strongly convex and $L$-Lipschitzness, which implies that

$$\begin{aligned}
\frac{\lambda}{2}\left\|\mathbf{w}^{-U} - \hat{\mathbf{w}}^{-U}\right\|^2 &\leq F(\mathbf{w}^{-U}) - F\left(\hat{\mathbf{w}}^{-U}\right) \\
F(\mathbf{w}^{-U}) - F\left(\hat{\mathbf{w}}^{-U}\right) &\leq \frac{2L^2}{\lambda}.
\end{aligned} \tag{69}$$

Therefore, we have completed the proof as follows,

$$\|\mathbf{w}_T^{-U} - \hat{\mathbf{w}}^{-U}\| \leq \rho^T\frac{2L}{\lambda}, \text{ where } \rho = \max\{|1-\eta\lambda|, |1-\eta M|\} < 1. \tag{70}$$

□

Building upon the proofs presented in Lemma 9 and Lemma 10, we are now prepared to establish the generalization results stated in Theorem 6.

***Proof of Theorem 6.*** Let $\hat{\mathbf{w}}^{-U}$ be the empirical risk minimizer of the objective function $F_{\mathcal{S}\setminus U}(\mathbf{w})$, where $|U| = m$. Let $T$ be the total number of update steps. For any $z \in \mathcal{Z}$ the loss function $\ell(\mathbf{w}; z)$ is $\lambda$-strongly convex, $M$-smoothness and $L$-Lipschitz. For a minimizer $\mathbf{w}^*$ of the population risk in Equation (1) and $\tilde{\mathbf{w}}_{E,B}^{-U} = \mathbf{w}_{E,B} + \mathbf{a}_{E,B}^{-U} + \mathbf{n}$, we have that

$$\mathbb{E}\left[F(\tilde{\mathbf{w}}_{E,B}^{-U}) - F(\mathbf{w}^*)\right] = \mathbb{E}\left[F(\tilde{\mathbf{w}}_{E,B}^{-U}) - F(\hat{\mathbf{w}}^{-U})\right] + \mathbb{E}\left[F(\hat{\mathbf{w}}^{-U}) - F(\mathbf{w}^*)\right]. \tag{71}$$

Based on Lemma 9, we get

$$\mathbb{E}\left[F(\hat{\mathbf{w}}^{-U}) - F(\mathbf{w}^*)\right] \leq \frac{4L^2}{\lambda(n-m)}. \tag{72}$$

Then we note that the function $F(\mathbf{w})$ satisfies $L$-Lipschitz, therefore we can obtain,

$$\mathbb{E}\left[F(\tilde{\mathbf{w}}_{E,B}) - F(\mathbf{w}^*)\right] \leq \mathbb{E}[L\|\tilde{\mathbf{w}}_{E,B} - \hat{\mathbf{w}}^{-U}\|] + \frac{4L^2}{\lambda(n-m)}. \tag{73}$$

For the first term of Equation (73), we have

$$\mathbb{E}\left[\|\tilde{\mathbf{w}}_{E,B}^{-U} - \mathbf{w}_{E,B}^{-U} + \mathbf{w}_{E,B}^{-U} - \hat{\mathbf{w}}^{-U}\|\right] \leq \mathbb{E}\big[\underbrace{\|\tilde{\mathbf{w}}_{E,B}^{-U} - \mathbf{w}_{E,B}^{-U}\|}_{C_3}\big] + \mathbb{E}\big[\underbrace{\|\mathbf{w}_{E,B}^{-U} - \hat{\mathbf{w}}^{-U}\|}_{C_4}\big]. \tag{74}$$

Therefore, we now bound the first term $C_3$ and the second term $C_4$ of Equation (74) as follows:

**(1)** $C_4$: Based on Lemma 10 and $\mathbf{w}_{E,B}^{-U} = \mathbf{w}_T^{-U}$, we have that for $\eta \leq \frac{2}{\lambda+M}$,

$$\mathbb{E}\big[\|\mathbf{w}_{E,B}^{-U} - \hat{\mathbf{w}}^{-U}\|\big] \leq \rho^T \frac{2L}{\lambda}, \text{ where } \rho = \max\{|1 - \eta\lambda|, |1 - \eta M|\} < 1. \tag{75}$$

**(2)** $C_3$: Based on Theorem 4, we obtain,

$$\mathbb{E}[\|\tilde{\mathbf{w}}_{E,B}^{-U} - \mathbf{w}_{E,B}^{-U}\|] = \mathbb{E}[\|\mathbf{w}_{E,B} + \mathbf{a}_{E,B}^{-U} - \mathbf{w}_{E,B}^{-U}\|] + \mathbb{E}[\|\mathbf{n}\|] \leq \|\Delta_{E,B}^{-U}\| + \sqrt{d}c.$$
$$\text{where } \|\Delta_{E,B}^{-U}\| = \frac{2\eta^2 G}{1-q}\left(\frac{\rho^T - q^T}{\rho - q} - \frac{\rho^T - q^{2T}}{\rho - q^2}\right) + \frac{2\eta G}{|\mathcal{B}|}\frac{\rho^T - q^T}{\rho^B - q^B}\rho^{B-b(u)-1}q^{b(u)}. \tag{76}$$

And since the loss function $\ell(\mathbf{w}; z)$ satisfies $M$-smoothness and $\lambda$-strongly convex, we thus have that $0 < \rho < 1$. Plugging the above bound Equation (75) and Equation (76) into Equation (74), we have

$$\mathbb{E}\left[F(\tilde{\mathbf{w}}_{E,B}^{-U}) - F(\mathbf{w}^*)\right] \leq \frac{4L^2}{\lambda(n-m)} + \rho^T \frac{2L^2}{\lambda} + \left(\frac{\sqrt{d}\sqrt{2\ln(1.25/\delta)}}{\epsilon} + 1\right) \cdot L\|\Delta_{E,B}^{-U}\|. \tag{77}$$

Therefore, we have completed the proof. □

***Tightness Analysis of Theorem 6.*** Furthermore, the following settings, where $\rho$ is close to $q$, might render the bound on the approximation error vacuous and also lead to a high approximation error. We note that the first and third terms of $\|\Delta_{E,B}^{-U}\|$ can be equivalently written as $\mathcal{O}(\frac{\rho^T - q^T}{\rho - q} + \frac{\rho^T - q^T}{\rho^B - q^B} = \rho^{T-1}\sum_{t=0}^{T-1}(\frac{q}{\rho})^t + \rho^{T-B}\sum_{e=0}^{B}(\frac{q}{\rho})^{eB})$. These terms increase as $\rho$ approaches $q$, however are always upper bounded by $\mathcal{O}(\rho^{T-1} \cdot T + \rho^{T-B} \cdot T)$. In our analysis, $T = EB + B$ is a fixed constant, so this bound remains meaningful, although it may not be tight in some cases. Such as when $T \to \infty$, in this scenario $\rho$ is close to 1, the bound therefore is equivalent to $\mathcal{O}(\frac{2\eta G}{1-q} - \frac{2\eta G}{1-q^2} + \frac{2\eta G}{1-q^B})$. Fortunately, SGD typically does not need to find a local minima in infinite time, hence $\rho$ is always less than 1.

## D    SUPPLEMENTARY EXPERIMENTS

### D.1    HARDWARE, SOFTWARE AND SOURCE CODE

The experiments were conducted on the NVIDIA GeForce RTX 4090. The code were implemented in PyTorch 2.0.0 and leverage the CUDA Toolkit version 11.8. Our comprehensive tests were conducted on AMD EPYC 7763 CPU @1.50GHz with 64 cores under Ubuntu20.04.6 LTS.
Source Code: Our code is available at ○ Hessian Free Certified Unlearning

## D.2 Additional Verification Experiments

The verification experiments in this section consist of two parts: (1) We first investigate the reasons behind the randomness of previous works under the non-convex setting. We select forgetting samples under different random seeds, and show that previous works exhibit dependence for forgetting samples. This implies that the forgetting effect in prior studies benefits certain samples while negatively impacting others. (2) Second, we perform experiments on a larger-scale dataset and model with more metrics. Prior studies were unable to achieve this due to the costly Hessian computation.

**Sensitivity to the forgetting samples.** We conduct experiments on a simple CNN and the hyperparameter setup is consistent with the verification experiments I in the main paper. As shown in Figure 4, we provide intuitive insights into the reasons for the decline of prior works in non-convex settings. Specifically, we investigated the $L^2$ norm of the approximate parameter change $\|\mathbf{a}^{-U}\|$ and the norm of the exact parameter change across different selections of the forgetting sample. We observed that the performance of *NS* and *IJ* depends on the selection of the forgetting data points, i.e., *NS* and *IJ* can approximate actual changes well for some data while generating large approximation errors for others under the non-convex setting, as illustrated in Figure 4. The dependency of forgetting data points results in the decrease in distance and correlation coefficient metrics observed in Figure 1 (b). The observed results for *NS* and *IJ* are consistent with previous studies on influence functions (Basu et al. (2021; 2020)), where these methods display random behavior under the non-convex setting, resulting in a decrease in correlation with retraining. In contrast, our approach can effectively approximate the actual norm of parameter changes, regardless of the selection of the forgetting sample.

**Additional experiments.** We further evaluate using larger-scale model ResNet-18 (He et al. (2016)) which features 11M parameters with three datasets: CIFAR-10 (Alex (2009)) for image classification, CelebA (Liu et al. (2015)) for gender prediction, and LFWPeople (Huang et al. (2007)) for face recognition across 29 different individuals. Since previous second-order works (*NS* Sekhari et al. (2021)) and (*IJ* Suriyakumar et al. (2022)) are difficult to compute (Hessian) in this scenario, we instead use the following fast baselines, as described and similarly set up in Tarun et al. (2023):

- *FineTune*: In the case of finetuning, the learned model is finetuned on the remaining dataset.
- *NegGrad*: In the case of gradient ascent, the learned model is finetuned using the negative of update direction on the forgetting dataset.

**Configurations:** For *FineTune* and *NegGrad*, we adjust the hyperparameters and execute the algorithm until the accuracy on the forgetting dataset approaches that of the retrained model. Here are our detailed hyperparameter settings: (1) We conducted evaluation on ResNet-18 trained on CIFAR-10 with 50,000 samples. The learning stage is conducted for 40 epochs with step size of 0.001 and batch size of 256. For *FineTune*, 10 epochs of training are performed with step size of 0.001 and batch size of 256. For *NegGrad*, we run gradient ascent for 1 epoch with step size of $5 \times 10^{-5}$ and batch size of 2. (2) We conducted evaluation on ResNet-18 trained on LFW with 984 samples, for the classification of 29 facial categories. The learning stage is conducted for 50 epochs with step size of 0.004 and batch size of 40. For *FineTune*, 10 epochs of training are performed with step size of 0.004 and batch

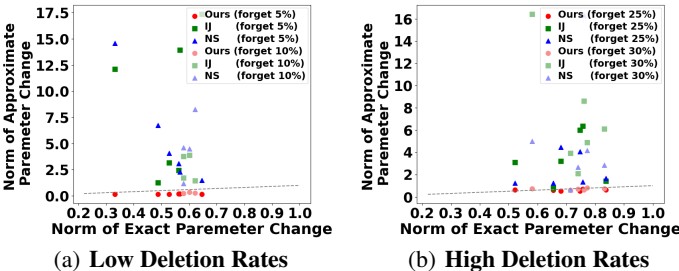

(a) **Low Deletion Rates**            (b) **High Deletion Rates**

Figure 4: **Verification experiments II**. Evaluation shows a comparison between the norm of approximate parameter change $\|\mathbf{a}^{-U}\|$ and norm of exact parameter change $\|\mathbf{w}_{E,B}^{-U} - \mathbf{w}_{E,B}\|$ across different random seeds. Intuitively, the *NS* and *IJ* methods are contingent on the selection of forgetting data. In contrast, our approach consistently approximates the actual values effectively.

size of 40. For *NegGrad*, we run gradient ascent for 1 epoch with step size of $4e^{-6}$ and batch size of 1. (3) We conducted evaluation on ResNet-18 trained on CelebA with 10,000 samples. The learning stage is conducted for 5 epochs with step size of 0.01 and batch size of 64. For *FineTune*, 2 epochs of training are performed with step size of 0.01 and batch size of 64. For *NegGrad*, we run gradient ascent for 1 epoch with a step size of 0.0005 and batch size of 1. We threshold the gradients within 10 for the above experiments. Finally, we uniformly unlearned 50 data points for each dataset. We denote the test data as $D_t$, the remaining data as $D_r$, and the forgetting data as $D_f$.

The following Table 3-5 are our evaluation results: we employ five widely-used metrics to evaluate the performance of unlearning methods: $Acc\_D_t$ (accuracy on $D_t$), $error\_D_t$ (error on $D_t$), $error\_D_r$ (error on $D_r$), $error\_D_f$ (error on $D_f$), $Distance$ ($L^2$ norm between unlearned and retrained model).

**Performances of our scheme:** We still maintain model indistinguishability from the retrained model on more complex tasks and larger models, where the distance is only 0.06 on CIFAR10 and 0.29 on CelebA. The results also show that the unlearned model of proposed method performs similarly to the retrained model in terms of the accuracy metric. Moreover, compared to these baselines, our method achieves forgetting at only the millisecond level, demonstrating the potential of our method on larger-scale models. Additionally, we noticed that when training is unstable, our method would lead to larger performance degradation, such as on the LFW dataset. Although our method has similar accuracy on the forgetting dataset, it suffers performance loss on the remaining dataset. This implies that our method is sensitive to the training process and hyperparameters in non-convex scenarios. We summarize the limitations in Appendix B.1, and provide possible schemes to address these problems.

**Model indistinguishability:** For *FineTune*, we can observe that even though the accuracy results are similar to the retrained model, the distance metric is very large (e.g. 2.21 on CIFAR-10 and 2.36 on CelebA). For *NegGrad*, we observe that the accuracy on forgetting dataset $D_f$ and remaining dataset $D_r$ would rapidly decrease simultaneously, but even in this case, the distances are is still lower than that of *FineTune* for all three datasets. This demonstrates that a small difference in accuracy does not indicate the model indistinguishability. Therefore, we use the $L^2$ norm and correlation metric as the primary measure of forgetting completeness, which are commonly employed in studies related to *influence function* (Basu et al. (2021; 2020)). These studies serve as inspiration for prior certified unlearning papers (Sekhari et al. (2021); Suriyakumar et al. (2022); Liu et al. (2023)) that we compare. Measuring forgetting through $L^2$ norm is also used in unlearning research, e.g., Wu et al. (2020); Izzo et al. (2021), but it necessitates retraining a model, which is impractical for real-world applications.

Table 3: **Verification experiments III.** Unlearning results on Resnet-18 trained on CIFAR-10 dataset.

| Method | $Acc\_D_t$ (%) | $Err\_D_t$ (%) | $Err\_D_r$ (%) | $Err\_D_f$ (%) | $Distance$ | Runtime (Sec) |
|---|---|---|---|---|---|---|
| FineTune | 79.34 | 20.66 | 15.92 | 22.00 | 2.21 | 106.76 |
| NegGrad | 40.61 | 59.39 | 59.15 | 70.00 | 0.11 | 0.56 |
| Retrain | 79.63 | 20.37 | 16.19 | 24.00 | — | 468.65 |
| **Ours** | 79.62 | 20.38 | 16.15 | 22.00 | **0.06** | **0.0006** |

Table 4: **Verification experiments IV.** Unlearning results on Resnet-18 trained on LFW dataset.

| Method | $Acc\_D_t$ (%) | $Err\_D_t$ (%) | $Err\_D_r$ (%) | $Err\_D_f$ (%) | $Distance$ | Runtime (Sec) |
|---|---|---|---|---|---|---|
| FineTune | 76.02 | 23.98 | 8.99 | 4.00 | 0.52 | 15.60 |
| NegGrad | 63.01 | 36.99 | 15.63 | 20.00 | 0.48 | 0.46 |
| Retrain | 80.89 | 19.11 | 4.07 | 26.00 | — | 77.43 |
| **Ours** | 71.92 | 28.08 | 14.91 | 22.00 | **0.33** | **0.001** |

Table 5: **Verification experiments V.** Unlearning results on Resnet-18 trained on CelebA dataset.

| Method | $Acc\_D_t$ (%) | $Err\_D_t$ (%) | $Err\_D_r$ (%) | $Err\_D_f$ (%) | $Distance$ | Runtime (Sec) |
|---|---|---|---|---|---|---|
| FineTune | 95.78 | 4.22 | 1.07 | 4.00 | 2.36 | 35.09 |
| NegGrad | 37.95 | 62.05 | 58.12 | 60.0 | 1.13 | 0.96 |
| Retrain | 95.72 | 4.28 | 1.20 | 4.00 | — | 87.72 |
| **Ours** | 95.57 | 4.43 | 1.12 | 2.00 | **0.29** | **0.0006** |

### D.3 ADDITIONAL APPLICATION EXPERIMENTS

Furthermore, we evaluate the performance on different datasets and models in real-world applications. Our evaluations are conducted from two perspectives: runtime and utility. Specifically, run-time focuses on the time spent precomputing unlearning statistics and the speedup of the unlearning algorithm compared to the retraining algorithm. Moreover, we evaluate the utility by the test accuracy of the unlearned model to ensure that generalization performance is not compromised. Finally, we record the area under the curve (AUC) for the MIA-L attack.

**Configurations:** We assess the performance on various datasets and models, as presented in Table 6. The values in parentheses indicate the difference in test accuracy compared to the retrained model. (1) We train a LR and simple CNN on MNIST with 1,000 training data 20% data points to be forgotten, which have setups identical to the aforementioned verification experiments I. We further evaluate on FMNIST with 4,000 training data and 20% data points to be forgotten using CNN and LeNet with a total of 61,706 parameters. The training is conducted for 30 epochs with a stepsize of 0.5 and a batch size of 256. (2) Moreover, we assessed our method on Resnet-18 (He et al. (2016)) trained on the CIFAR10 dataset for image classification with 50,000 data points, CelebA dataset for gender prediction with 10,000 data points, and LFWPeople dataset for face recognition with 984 data points. we uniformly add noise with a standard deviation of 0.01. The configurations align with those detailed in Table 3-5. In these circumstances, we have not evaluated *NS* and *IJ* due to out-of-memory.

**Application experiment results** show that our proposed method is computationally efficient and performs well in realistic environments, outperforming benchmarks of certified unlearning regarding all metrics by a significant margin, as demonstrated in Table 6. It achieves minimal computational and storage overhead while maintaining high performance in both convex and non-convex settings. We further investigate data removal with large-scale datasets using ResNet-18, as shown in Table 6. Our proposed Hessian-free method demonstrates significant potential for over-parameterized models, as previous second-order certified unlearning works are unable to compute the full Hessian matrix in such cases. Notably, for unlearning runtime, by precomputing statistical data before unlearning requests arrive, we only need to execute simple vector additions, achieving millisecond-level efficiency (0.6 ms) with minimal performance degradation in test accuracy. Finally, although our method has lower computation and storage complexity compared to previous works, it still requires $\mathcal{O}(n^2 d)$ precomputation and $\mathcal{O}(nd)$ storage which is expensive when facing both larger-scale datasets. We summarize the limitations in Appendix B.1, and provide possible schemes to address these problems.

Table 6: **Application experiments II.** We conduct two types of comparisons: (1) on small-scale models and datasets with previous second-order algorithms, where we compute the total computational/storage cost for all samples; and (2) on large-scale models and datasets with fast unlearning algorithms, where we compute the computational/cost overhead of per-sample forgetting.

| Method | Dataset | Model | Unlearning Computaion Runtime (Sec) | Speedup | PreComputaion Runtime (Sec) | Storage (GB) | MIA-L AUC | Distance | Test Accuracy (%) Unlearned model |
|---|---|---|---|---|---|---|---|---|---|
| NS | MNIST | LR | $5.09\times10^2$ | $1.34\times$ | $2.55\times10^3$ | 0.23 | 0.51 | 0.18 | 86.25 (-1.50) |
| | | CNN | $5.81\times10^3$ | $0.12\times$ | $2.91\times10^4$ | 1.78 | 0.51 | 1.78 | 83.50 (-10.25) |
| | FMNIST | CNN | $2.31\times10^4$ | $0.54\times$ | $1.16\times10^5$ | 1.78 | 0.54 | 2.41 | 57.69 (-22.25) |
| | | LeNet | $8.53\times10^4$ | $0.15\times$ | $4.27\times10^5$ | 14.18 | 0.54 | 1.10 | 62.00 (-18.50) |
| IJ | MNIST | LR | $0.23\times10^2$ | $29.64\times$ | $2.55\times10^3$ | 0.23 | 0.52 | 0.18 | 86.25 (-1.50) |
| | | CNN | $1.91\times10^2$ | $3.63\times$ | $2.91\times10^4$ | 1.78 | 0.51 | 3.12 | 82.75 (-11.00) |
| | FMNIST | CNN | $2.95\times10^1$ | $424\times$ | $1.16\times10^5$ | 1.78 | 0.51 | 0.94 | 75.69 (-4.25) |
| | | LeNet | $3.24\times10^1$ | $402\times$ | $4.27\times10^5$ | 14.18 | 0.54 | 0.57 | 76.75 (-3.75) |
| **HF** | MNIST | LR | **0.0073** | **93,376×** | **$2.20\times10^2$** | **0.03** | **0.51** | **0.15** | **87.50** (-0.25) |
| | | CNN | **0.0268** | **25,821×** | **$5.34\times10^2$** | **0.08** | **0.51** | **0.68** | **91.50** (-2.25) |
| | FMNIST | CNN | **0.127** | **98,468×** | **$3.66\times10^3$** | **0.32** | **0.51** | **0.74** | **77.85** (-2.09) |
| | | LeNet | **0.163** | **79,828×** | **$4.43\times10^3$** | **0.48** | **0.53** | **0.53** | **78.63** (-1.87) |

| Method | Dataset | Model | Unlearning Computaion Runtime (Sec) | Speedup | PreComputaion Runtime (Sec) | Storage (GB) | MIA-L AUC | Distance | Test Accuracy (%) Unlearned model |
|---|---|---|---|---|---|---|---|---|---|
| FineTune | CIFAR10 | | 106.76 | 4.39 × | – | – | 0.56 | 2.21 | 79.34 (-0.29) |
| | LFW | Resnet18 | 15.60 | 4.96 × | – | – | 0.53 | 0.52 | 76.02 (-4.87) |
| | CelebA | | 35.09 | 2.50 × | – | – | 0.56 | 2.36 | **95.78** (0.06) |
| NegGrad | CIFAR10 | | 0.56 | 836.88 × | – | – | 0.51 | 0.11 | 40.61 (-39.03) |
| | LFW | Resnet18 | 0.46 | 168.33 × | – | – | **0.46** | 0.48 | 63.01 (-17.88) |
| | CelebA | | 0.96 | 91.38 × | – | – | **0.50** | 1.13 | 37.95 (-57.77) |
| **HF** | CIFAR10 | | **0.0006** | **781,083×** | 184.53 | 0.043 | **0.51** | **0.06** | **79.62** (-0.01) |
| | LFW | Resnet18 | **0.001** | **77,430 ×** | 84.99 | 0.043 | 0.56 | **0.31** | 71.92 (-8.97) |
| | CelebA | | **0.0006** | **146,200×** | 31.90 | 0.042 | 0.54 | **0.29** | 95.57 (-0.15) |

### D.4 ADDITIONAL ADVERSARY EXPERIMENTS

**Membership Inference Attack against Learning (MIA-L)** aims to determine whether a specific example is part of the training set of the learned model. Carlini et al. (2022) introduced the Likelihood Ratio Attack (LiRA), which leverages the model's per-example difficulty scores and employs a well-calibrated Gaussian likelihood estimate to gauge the probability of a sample being part of training data. Finally, the adversary queries the confidence of the target model on a specific example and outputs a parametric likelihood-ratio test. **Membership Inference Attack against Unlearning (MIA-U)** seeks to predict whether a target sample is part of the learned model's training set but not in the corresponding unlearned (retrained) model's training set. Chen et al. (2021) proposed MIA-U to attack Scratch methods, specifically the Retrain and SISA (Bourtoule et al. (2021)) algorithms, where the adversary jointly utilizes the posteriors of the two models as input to the attack model, either by concatenating them or by computing difference. Specifically, it includes five methods to construct features for the attack model: Direct difference (DirectDiff), Sorted difference (SortedDiff), Direct concatenate (DirectConcat), Sorted concatenate (SortedConcat), and Euclidean distance (EucDist).

**Configuration:** (1) We investigate the Area Under the Curve (AUC) of MIA-U at different deletion rates. (2) We report the Receiver Operating Characteristic (ROC) curve of MIA-U in large-scale experiments as well. (3) Finally, as a complement to the results of the DirectDiff method reported in the main text, we present the ROC of the remaining four feature construction methods in MIA-U. We maintain the other hyperparameters consistent with previous experiments.

**Different deletion rates.** As illustrated in Figure 5, a clear correlation exists between the deletion rate and the effectiveness of MIA-U attacks. Specifically, at a low deletion rate of 1%, the AUC of MIA-U attacks reaches 91% for our proposed method and 89% for the retrain method in the LR model, indicating that unlearning algorithms may increase the risk of privacy breaches. When the deletion rate exceeds 5%, the performance of MIA-U attacks approaches random guessing (AUC = 50%), suggesting that removing a larger amount of data between successive model releases makes it increasingly difficult for MIA-U attacks to accurately infer the deleted individual information.

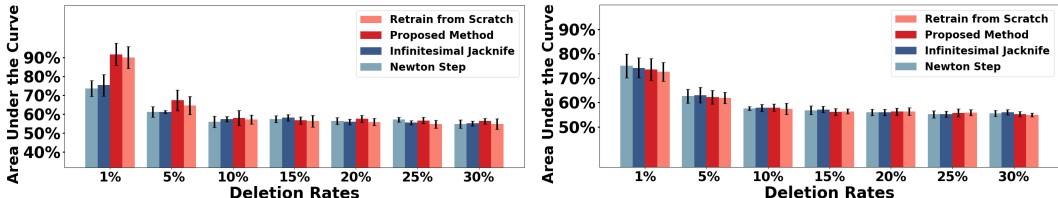

Figure 5: **Membership Inference Attack II**. The AUC values of MIA-U at different deletion rates. The feature construction method employed in MIA-U is DirectDiff, with error bars representing 95% confidence intervals. The target model in the first plot is LR, while the second plot corresponds to a CNN. The findings show that lower deletion rates lead to privacy leakage, whereas higher deletion rates diminish MIA-U's attack performance, bringing it closer to random guessing.

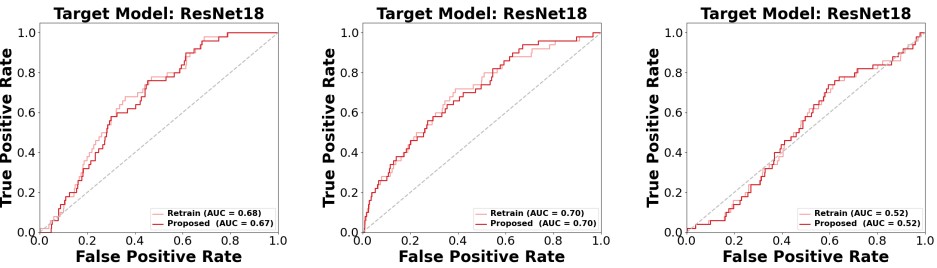

Figure 6: **Membership Inference Attack III**. ROC curves for MIA-U on ResNet18. The datasets used in the figure, from left to right, are CIFAR-10, LFW, and CelebA. For large-scale setups, our method (red line) performs consistently with the retraining algorithm (light red line) under MIA-U.

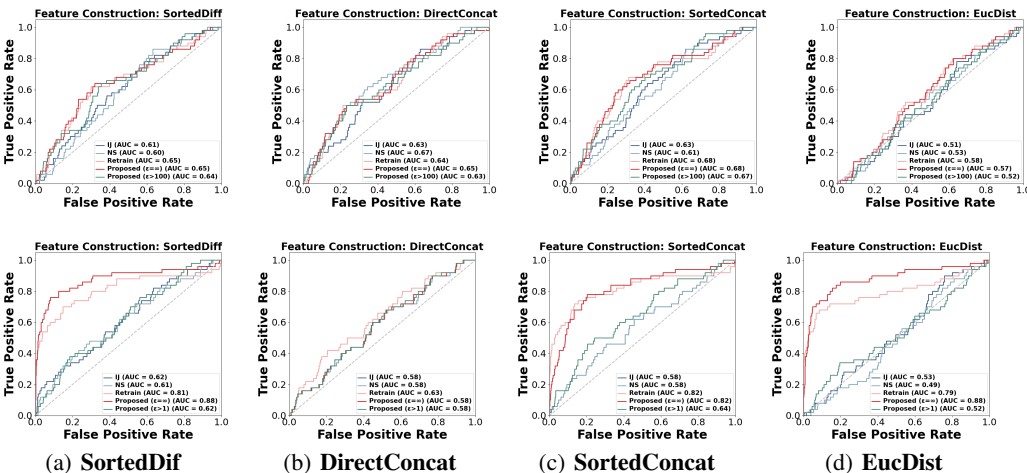

Figure 7: **Membership Inference Attack IV**. ROC curve for different feature construction methods. The target model in the first row is overfitted CNN, whereas the second row is well-generalized LR.

**Different feature construction methods.** As shown in Figure 7, we experiment with a well-generalized LR model and an overfitted CNN model, and the experimental results are consistent with the conclusion of Chen et al. (2021): sorting the posteriors improves attack performance; concatenation-based methods achieve higher attack performance on the overfitted CNN; and difference-based methods perform better on the well-generalized LR. In addition, we observe that perturbing the unlearning model can effectively defend the EucDist method. However, for other feature construction methods, adding noise to perturb the model after a certain level does not reduce the AUC of the attack model. This reveals a fundamental dilemma of most certified data removal processes developed so far: Although the goal is to achieve performance similar to a gold-standard retrained algorithm, the retrained model itself remains vulnerable to MIA-U, exposing the failure modes of certified unlearning. Nonetheless, as shown in Figure 5, removing at least 5% of the data when releasing the unlearned model can mitigate privacy leakage. However, we are not aware of any recent certified data removal mechanisms that can satisfy the deletion capacity exceeding 5% of the training set. Fortunately, one of the defense mechanisms proposed by Chen et al. (2021), which employs Differentially Private Stochastic Gradient Descent (DP-SGD), can effectively mitigate MIA-U but requires careful trade-off amongst privacy, unlearning efficiency, and model utility.

## E    SENSITIVITY ANALYSIS AND ABLATION STUDIES

In this section, we investigate the effects of different step sizes, epochs, stepsize decay rates, and gradient clipping on the proposed approach. We conducted simulations in both convex and nonconvex scenarios, using LR for the convex case and 2-layer CNN for the non-convex case, respectively.

**Takeaway.** Let's first provide a brief summary of the main results of ablation studies.

**(1) Impact of Step Sizes.** Firstly, a smaller step size enables our model to be closer to the retrained model. However, when the step size exceeds a certain threshold, the bound will become meaningless and result in significant approximation errors. **(2) Impact of Epochs.** Secondly, when the step size is sufficiently small, the approximation error is insensitive and exhibits slow growth with respect to the number of epochs, i.e. excessive iterations in this scenario do not lead to significant errors. **(3) Impact of Stepsize Decay Rates.** Thirdly, our results indicate that as the decay rate $q$ decreases, the approximation error diminishes. Specifically, The stepsize decay strategy introduces a bound on approximation error, preventing a continuous increase in error as the size of epochs grows. **(4) Impact of Gradient Clipping.** To prevent gradient explosion, we introduced gradient clipping. We show that gradient clipping, which scales the gradient based on its norm, has a similar effect to using a smaller step size. Although, in theory, clipping the gradient may introduce errors, our experiments indicate that this impact is negligible when the threshold is set to a large value.

### E.1 IMPACT OF STEP SIZES

In this section, we investigated (i) the impact of step sizes and (ii) the impact of small stepsizes with more epochs. Additionally, we discussed the connection between learning generalization and our unlearning method.

**Configurations:** (i) We investigate the impact of step sizes $\{0.005, 0.01, 0.05, 0.1, 0.3\}$ on distance metrics in Table 7 and Table 8. For LR, training was performed for 15 epochs with a stepsize of 0.05 and batch size of 32. For CNN, training was carried out for 20 epochs with a step size of 0.05 and a batch size of 64. (ii) Furthermore, in Figure 9, to investigate the effect of epochs under small step sizes, LR was trained for 100 epochs and 200 epochs with a step size of 0.01 (initially 0.05 with 15 epochs). For CNN, training for 50 epochs and 500 epochs with a step size of 0.025 (originally 0.05 with 20 epochs) was performed.

**Impact of step sizes.** As shown in Table 7 and Table 8, a smaller step size enables our model to be closer to the retrained model. Specifically, as Theorem 4 demonstrates that, in this process, the error arises from the scaled remainder term, i.e., the term $o(\mathbf{w}_{e,b}^{-u} - \mathbf{w}_{e,b})$ is scaled by the subsequent $\mathbf{M}_{e,b(u)}$. Therefore, the approximation error mainly consists of two components: (1) the step size $\eta_{e,b}$ and (2) the maximum eigenvalue $\lambda_1$ of $\mathbf{H}_{e,b}$. A smaller step size $\eta$ contracts the eigenvalues of the matrix $\eta_{e,b}\mathbf{H}_{e,b}$, thereby reducing the spectral radius $\rho$ of the matrix $\mathbf{I} - \eta_{e,b}\mathbf{H}_{e,b}$, and ultimately reducing the approximation error.

Furthermore, when the step size is smaller than a certain threshold, increasing the step size does not lead to significant errors. However, when the step size exceeds this threshold (making $\rho$ not less than or close to 1), the errors will grow exponentially, as demonstrated in Table 7 and Table 8 for step size $\eta \geq 0.3$. In this scenario, our method leads to a complete breakdown of the model, rendering it unusable. However, it is noteworthy that theoretical predictions (Wu et al. (2018)) and empirical validation (Gilmer et al. (2022)) suggested that, across all datasets and models, successful training occurs only when optimization enters a stable region of parameter space where $\lambda_1 \cdot \eta \leq 2$. Therefore, achieving good optimization results typically does not require a strategy of using aggressive stepsizes.

**The connection between learning and unlearning.** There is a connection between the sharpness of learning generalization studies and the unlearning method that we propose. Specifically, these studies aim to explore the sharpness of the Hessian of loss function $\mathbf{H}_{e,b}$, and the term 'sharpness' refers to the maximum eigenvalue of the Hessian, denoted as $\lambda_1$ in some studies, such as Gilmer et al. (2022). A flat loss space is widely recognized as beneficial for gradient descent optimization, leading to improved convergence. Importantly, Gilmer et al. (2022) demonstrated the central role that $\lambda_1$ plays in neural network optimization, emphasizing that maintaining a sufficiently small $\lambda_1$ during optimization is a necessary condition for successful training (without causing divergence) at large step sizes, as shown in Figure 8. Models that train successfully enter a region where $\lambda_1 \cdot \eta \leq 2$ mid training (or fluctuate just above this bound). In our paper, we leverage the second-order information from the Hessian at each iteration during the training process to compute the approximators and achieve forgetting. As indicated in Theorem 4, a small $\lambda_1$ of the Hessian is beneficial for reducing

Table 7: **Impact of Step Sizes** on Convex Setting. Stepsize 0.05 represents the hyperparameter choice of experiments for LR in the main paper. It can be observed that as the step size decreases, the unlearned model is closer to the retrained model. (Seed 42)

| Stepsize | Distance | | | |
|---|---|---|---|---|
| | 1% | 5% | 20% | 30% |
| 0.005 | 0.018 | 0.040 | 0.092 | 0.128 |
| 0.01 | 0.023 | 0.0518 | 0.127 | 0.178 |
| **0.05** | 0.027 | 0.064 | 0.149 | 0.210 |
| 0.1 | 0.028 | 0.066 | 0.154 | 0.217 |
| 0.3 | 5.581 | 17.48 | 28.45 | 40.64 |

Table 8: **Impact of step sizes** on non-Convex setting. Stepsize 0.05 represents the hyperparameter choice of experiments for CNN in the main paper. It can be observed that as the step size decreases, the unlearned model is closer to the retrained model. (Seed 42)

| Stepsize | Distance | | | |
|---|---|---|---|---|
| | 1% | 5% | 20% | 30% |
| 0.005 | 0.004 | 0.015 | 0.058 | 0.084 |
| 0.01 | 0.017 | 0.069 | 0.257 | 0.372 |
| **0.05** | 0.167 | 0.380 | 0.584 | 0.825 |
| 0.1 | 0.437 | 0.795 | 1.722 | 2.72 |
| 0.3 | 122.2 | 393.4 | 1647 | 2455 |

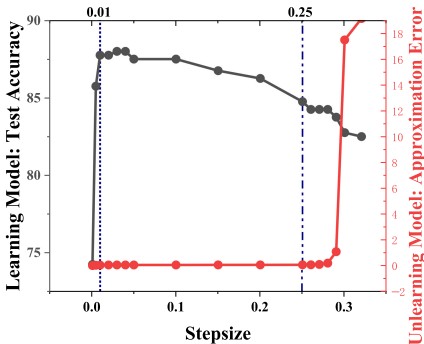

Figure 8: **The connection between learning and unlearning.** We conduct experiment on MNIST and keep the other hyperparameters fixed and adjust the learning rate from 0.001 to 0.31. Under a fixed iteration budget, we have: (1) When the step size is small (less than the threshold $\eta_1 = 0.01$), the training is often insufficient, leading to lower test accuracy. (2) Increasing the step size ensures sufficient training, but when the learning rate becomes too large, overfitting and instability lead to a decrease in test accuracy. We also observe the following phenomena in the unlearning process: (1) When the step size is below a threshold $\eta_2 = 0.25$, the growth of approximation error is extremely slow. (2) However, once the step size exceeds this threshold, the error grows exponentially.

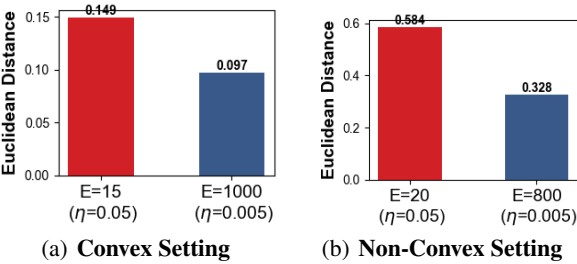

(a) **Convex Setting**          (b) **Non-Convex Setting**

Figure 9: **Impact of Small Step Size with More Epochs.** (a)(b) correspond to LR and CNN with 20% of the data to be forgotten, respectively. The red bar represents the distance metric when using a larger step size; the blue bar represents the distance metric with a small step size during the larger epochs. It can be observed that even when training more epochs with smaller stepsize, the approximation error remains lower compared to training with larger stepsize. (Seed 42)

the approximation error. Therefore, effectively reducing $\lambda_1$ during optimization, so that the training enters a flatter region, will not only improve the generalization ability of the learning stage model but also reduce the approximation error in the unlearning stage. As suggested in Gilmer et al. (2022), strategies like stepsize warm-up or initialization strategies for architectures can be employed to stabilize learning and potentially reduce unlearning errors by decreasing $\lambda_1$, enabling training at larger step sizes.

**Impact of small step size with more epochs.** Finally, as shown in Figure 9, with a small step size (0.005) during model training, we observed a lower approximation error. Even with large epoch sizes, the approximation error remains lower compared to cases with a large step size, such as 0.05. Therefore, selecting an appropriately small step size ensures smaller approximation errors, as these errors typically converge to a smaller value as the number of epochs increases.

### E.2    IMPACT OF EPOCHS

In this subsection, we began investigating the distance metric as the number of epochs increases.

**Configurations:** We investigated the distance by varying the epoch from 1 to 100. We conducted experiments with a high deletion rate of 20% and a low deletion rate of 5%. The selection of other hyperparameters remains consistent with the ones used previously.

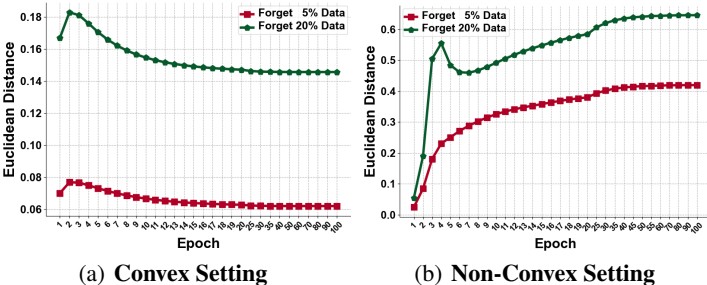

(a) **Convex Setting**       (b) **Non-Convex Setting**

Figure 10: **Impact of Epoch.** (a) (b) illustrate that in the distance for LR and CNN by varying epoch from 1 and 100. As shown in the Figure (a) (b), the error accumulates with the increase in epochs. However, with an appropriate choice of step size, such growth is acceptable (Seed 42).

**Impact of epochs.** As shown in Figure 10, the distance exhibits convergent behavior, ultimately stabilizing at a constant value. This phenomenon is consistent with our theoretical findings, as stated in Theorem 4, which demonstrates that the approximation error decreases with an increasing number of epochs for convex models. The contractive mapping ensures a gradual reduction in the final error. In the case of non-convex models, the error tends to increase with the number of epochs, ultimately stabilizing, a phenomenon that can be explained by the degeneracy of the Hessian in over-parameterized models (Sagun et al. (2018)), as we will discuss in further detail later.

### E.3 IMPACT OF STEPSIZE DECAY

These experiments aim to study the impacts of stepsize decay. We consider two scenarios: one with stepsize decay ($q = 0.995$) and the other without stepsize decay ($q = 1$).

**Impact of decay rates of stepsize:** We can observe from Figure 11 that the Euclidean distance for the stepsize decay strategy is generally smaller compared to that without the stepsize decay. This aligns with our earlier analysis of step size in Appendix E.1, i.e., a smaller step size leads to a smaller approximation error, and choosing an appropriate stepsize decay rate ensures a continuous reduction in step size, preventing the unbounded growth of the approximation error.

Additionally, as in Figure 11 (b), It can be observed that the distance metric tends to stabilize even without stepsize decay. This result can be explained by the conclusions in Sagun et al. (2018; 2016). Specifically, Sagun et al. (2018; 2016) analyzed the spectral distribution of the Hessian eigenvalues during CNN training. It reveals that, initially, the Hessian eigenvalues are symmetrically distributed around 0. As training progresses, they converge towards 0, indicating Hessian degeneracy. Toward the end of training, only a small number of eigenvalues become negative, and a large portion of the Hessian eigenvalues approach 0 when the training approaches convergence. Recall that our previous analysis of the stepsize in Appendix E.1, where we demonstrate that the approximation error is mainly determined by $\rho$ of $\mathbf{I} - \eta_{e,b}\mathbf{H}_{e,b}$, and $\rho$ is determined by $\eta_{e,b}$ and $\lambda_1$. As the number of iterations increases, the eigenvalues of the Hessian $\mathbf{H}_{e,b}$ tends to 0. This implies a decrease in $\lambda_1$, which leads to a slower increase in error. Therefore, in Figure 11 (b), the distance gradually stabilizes even without a decay strategy due to the decrease in $\lambda_1$ during training, and it still increases due to the existence of some larger outlier eigenvalues.

### E.4 IMPACT OF GRADIENT CLIPPING

Stochastic gradient algorithms are often unstable when applied to functions that lack Lipschitz continuity and/or bounded gradients. It is well-established that exploding gradients pose a significant challenge in deep learning applications (Hanin (2018)). Since our proposed method relies on backtracking historical gradients, the approach may fail completely when gradients tend toward infinity. To circumvent this issue, we incorporated gradient clipping into all the aforementioned experiments. Gradient clipping is a widely adopted technique that ensures favorable performance in deep network training by restricting gradient norms from becoming excessively large (Mai & Johansson (2021); Chen et al. (2020); Zhang et al. (2020)). However, this operation indicates that

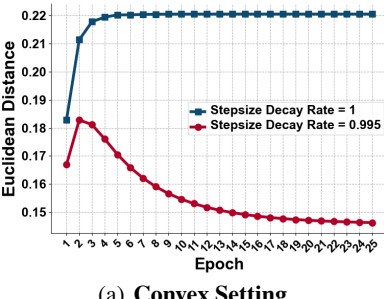
(a) **Convex Setting**

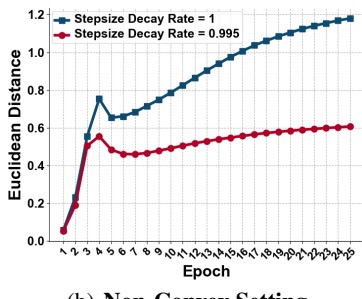
(b) **Non-Convex Setting**

Figure 11: **Impacts of the stepsize decay rate** on the Euclidean distances for (a) the LR and (b) the CNN with 20% data to be forgotten during 25 epochs. The red line represents learning with a decay rate of 0.995, while the blue line represents without decay. It can be observed that stepsize decay effectively reduces the approximation error of the unlearned model. (Seed 42)

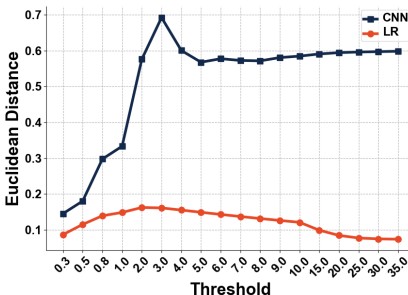

Figure 12: **Impacts of the gradient clipping** on the distance metric for the LR (orange) and the CNN (blue) with 20% data to be forgotten. (Seed 42)

the gradients in the Taylor expansion of Equation (2) are not the true gradients but rather the clipped gradients, which introduces an approximation error. Our subsequent experiments will demonstrate that this error can be rendered negligible. It is worth noting that our method does not inherently rely on gradient clipping but instead imposes requirements on the stability of the learning process.

**Configurations:** We investigate the impact of gradient clipping with thresholds set at {0.3, 0.5, 0.8, 1, 2, 3, 4, 5, 6, 7, 8, 9, 10, 15, 20, 25, 30, 35}. We remove 20% of the data while keeping other hyperparameter settings consistent with those described in the main text.

**Impact of gradient clipping.** As evidenced in Figure 12, the implementation of small clipping thresholds results in a scaling down of the gradient proportional to its norm, producing an effect similar to using a smaller step size. Consequently, the application of lower clipping thresholds maintains the error at a relatively minimal level. However, as the gradient clipping threshold surpasses a certain range, the error introduced by the clipping process becomes the predominant factor, culminating in an escalation of the overall error. Nonetheless, most gradients no longer require clipping during the optimization process. Therefore, the induced error diminishes until the threshold reaches a sufficiently elevated value, at which point no supplementary error is introduced. Once this threshold is attained, gradient clipping ceases to exert an influence on the performance of our proposed unlearning algorithm.

