# OpenReview forum: "Hessian-Free Online Certified Unlearning"
_ICLR.cc/2025/Conference — ICLR 2025 Poster_

### Official Review · Reviewer_t2wR · 2024-10-31

**Soundness:** 3
**Presentation:** 3
**Contribution:** 3
**Rating:** 6
**Confidence:** 2

**Summary:**

This paper proposes a Hessian-free machine unlearning algorithm. The authors theoretically analyze the approximation error for both convex and non-convex loss functions and prove the generalization theory for strongly convex loss functions. Extensive experiments demonstrate the efficiency of the proposed algorithm compared to other Hessian-based algorithms.

**Strengths:**

1. The authors analyze the training trajectory and propose a machine unlearning algorithm, which is practical and innovative.
2. Compared to other Hessian-based algorithms, the proposed Hessian-free algorithm is efficient, especially for high-dimensional problems.
3. The authors conducted comprehensive experiments to validate the effectiveness of their proposed algorithm, and the experimental results are well presented.

**Weaknesses:**

Although the authors provide the approximation error analysis, there is no theoretical guarantee for the generalization performance of the unlearning model in the non-convex case.

**Questions:**

1. The authors propose using HVP to avoid directly calculating the Hessian matrix and reduce computational complexity, as discussed in Section 4.4. Could other algorithms discussed in Section 4.4 also benefit from HVP? For example, for IJ, $H^{-1}\nabla \ell $ can be approximately computed using $K$ steps of the conjugate gradient method, where each step HVP can be applied. Could this approach enable IJ to achieve lower complexity and experiment time, considering the entire process of precomputation and unlearning?  In this case, how does the proposed algorithm compare to IJ?
2. The authors discuss in Appendix E that a small step size leads to a smaller approximation error. However, a small step size may result in insufficient model training. Could the authors further explain the trade-off?

---

> ### Author Response · Authors · 2024-11-21
> **Official Comment by Authors**
>
> We sincerely appreciate your support and constructive suggestions. We have made every effort to revise the manuscript based on your valuable suggestions.
>
> We provide our responses to comments, denoted by **[W]** for weaknesses, **[Q]** for questions, **[L]** for lines in our manuscript, and **[R]** for references in the General Response.
>
> **W1:** *There is no theoretical guarantee for the generalization of unlearning model in the non-convex case.*
>
> >Our contributions for non-convex assumption primarily focus on unlearning domain. For theoretical analysis involving learning process conclusions, we inherit convexity assumption from prior unlearning works, adopting simple and rigorously assumed conclusions such as Lemma 9.
> >
> >We appreciate reviewer’s constructive comments. Given that unlearning community are starting to explore generalization in non-convex settings, as a future extension, we plan to provide non-convex theoretical conclusions in later versions or follow-up works.
>
> **Q1:** *Could other algorithms discussed in Section 4.4 benefit from HVP?*
>
> >We sincerely thank reviewer for their insightful questions and provide following clarifications.
> >
> >- The key reason is that, to handle deletion requests from unknown users, both NS and IJ require explicit pre-computing of Hessian and its inverse before an unlearning request arrives, which makes HVP impossible.
> >
> >  Recall the technique details of NS and IJ.
> >
> >  For NS, the update step is: $ \frac{1}{n-m} {   \Big( \frac{1}{n-m}\sum_{i=1}^n   \nabla^2 \ell (\hat{\mathbf{w}}; z_i) -\sum_{j \in U}  \nabla^2 \ell (\hat{\mathbf{w}};u_j )\Big) }^{-1}\sum_{j \in U} \nabla \ell (\hat{\mathbf{w}}; u_j).$
> >
> >  For IJ, the update step is:  $ \frac{1}{n} {(\frac{1}{n}\sum_{i=1}^n \nabla^2 \ell (\hat{\mathbf{w}}; z _ i) ) }^{-1}  \nabla \ell (\hat{\mathbf{w}}; u _ j).$
> >
> >  However, for Hessian $\sum_{j \in U} \nabla^2 \ell (\hat{\mathbf{w}}; u _ j)$ in NS and gradient $\nabla \ell (\hat{\mathbf{w}}, u_j)$ in both NS and IJ, **(i)** forgetting sample $u _ j$ is unknown before deletion request arrives, and **(ii)** model $\hat{\mathbf{w}}$ will be deleted after processing a deletion request, while subsequent unlearned model is also unknown since forgetting sample $u_j$ is unpredictable. Therefore, explicit pre-computing is required for $\sum_{i=1}^n \nabla^2 \ell (\hat{\mathbf{w}}; z_i)$ in NS and ${\left( \sum_{i=1}^n \nabla^2 \ell (\hat{\mathbf{w}}; z_i) \right)}^{-1}$ in IJ.
> >
> >  This is also a unique advantage of our method, which possesses additivity in Theorem 1 **(L245)**, enabling it to efficiently handle multiple deletion requests  in online manner. We will make this point clearer in revised manuscript to better highlight our contributions.
> >
> >- In addition, inaccurate techniques (e.g., as mentioned by reviewer, it could be approximated by using methods like least squares) would also render theoretical bounds not as strong as techniques proposed in NS and IJ. These operations require previous works to re-derive bound, resulting in loss of original $\mathcal{O}(m^2/n^2)$ approximation error advantage.
> >
> >Given the above reasons, approximate techniques are more suitable for non-privacy scenarios involving heuristic methods that do not require certified theoretical guarantees, such as bias removal based on influence function in **R8** .
>
> **Q2:** *Could authors further explain trade-off between smaller approximation error and insufficient model training caused by step size?*
>
> >We greatly appreciate the insightful questions, which has inspired us with new insights. Below, we provide explanations and present new experiments to support it.
> >
> >As we demonstrate in Appendix **L1806-L1820**:
> >
> >- Theoretical predictions **R6** and empirical validation **R7** suggest that successful training occurs only when optimization enters a stable region of parameter space, where $\lambda  _  1 \eta _ 1 < 2$ (with $\lambda  _  1$ being  largest eigenvalue of Hessian).
> >- We also observed when step size is below certain threshold $\eta  _  {\text{2}}$, increasing step size does not lead to unacceptable errors, e.g., in Table 7, increasing step size from 0.01 to 0.1 results in error increase of only 0.005. However, when step size exceeds 0.1 and increases to 0.3, error becomes uncontrollably large, increasing by 5.553.
> >
> >Combining the conclusions and observations, there exists a range on step size that ensures successful training during learning phase while preventing errors from escalating in unlearning phase. Our further experiments support this: maintaining a threshold that prevents fluctuations in error often allows for sufficient training (successful training as state in **R6**) at appropriate step size, as our method is less affected by step size when below a threshold, there is typically no noticeable tradeoff. For detailed code and results, please refer to anonymous [repository link](https://github.com/Anonymous202401/If-Recollecting-were-Forgetting) provided in **L107**.

---

> > ### Comment · Reviewer_t2wR · 2024-11-22
> > **Response to Rebuttal**
> >
> > I appreciate the authors' detailed response. I appreciate the explanation regarding why HVP cannot be utilized by other unlearning methods like NS and IJ. I also appreciate the explanation of the tradeoff between approximation error and model training. The experiments on the model's performance under different step sizes are very thoughtful.

---

> > > ### Author Response · Authors · 2024-11-24
> > > **Reply to Reviewer t2wR**
> > >
> > > We appreciate the meaningful discussion with the reviewers, which has helped us further clarify and enhance the contributions of our work  that were previously overlooked. Since Reviewer CNQV also raised questions regarding the HVP and Reviewer kjAJ suggested this as a contribution worth highlighting, we added new descriptions to the revised manuscript to further emphasize our contributions. The specific changes are as follows:
> > >
> > > > - We added more descriptions of previous works in the Introduction **(L68, L90)**.
> > > > - We included the advantage of Hessian-free methods in handling multiple deletion requests online, which previous Hessian-based works cannot achieve, in Theorem 1 (Additivity) **(L252-254)**.
> > > > - we explained why previous Hessian-based works fail to use HVP in Section 4.4,  **(L370-L375)**.
> > >
> > > If the reviewer has any further **Q**uestions or **W**eakness that need to be addressed, we would be glad to provide any clarification.

---

### Official Review · Reviewer_ia1D · 2024-11-05

**Soundness:** 2
**Presentation:** 1
**Contribution:** 2
**Rating:** 6
**Confidence:** 3

**Summary:**

This paper proposes a Hessian-free approach to certified machine unlearning that aims to improve computational efficiency and scalability in removing specific data points from a model without full retraining. Instead of relying on direct Hessian computations, which are computationally prohibitive in high-dimensional and non-convex settings, the method approximates the impact of data removal through affine stochastic recursions that analyze model update discrepancies. The method achieves computational gains, reducing unlearning time to $\mathcal{O}(md)$ and storage to $\mathcal{O}(nd)$, outperforming existing second-order methods.

**Strengths:**

- Both online learning and certified unlearning are highly significant research areas in machine learning.
 The paper is the first to introduce a Hessian-free approach to certified unlearning, which is a notable change from the dominant reliance on Hessian-based methods in second-order unlearning.

- Experimental validation shows unlearning runtime in milliseconds, robust generalization guarantees, and privacy improvements against membership inference attacks with an added noise mechanism.

- The authors also include relevant code and pseudo-code in the appendix, which is helpful for reproducibility.

**Weaknesses:**

1. The paper’s theoretical guarantees hinge on assumptions of convexity and smoothness (Assumption 1), which restricts the scope of the analysis to settings that are arguably idealized for real-world applications involving non-convex (e.g. deep learning) models. Thus, the authors likely overstate their contributions.

2. I would say, Hessian-free optimization for second-order (and even higher-order) algorithms is an active research area. Beyond the Machine Unlearning domain discussed in detail on page 3 and in Appendix A, the authors should connect the ideas presented here with a broader body of Hessian-free optimization work in classical parametric optimization. A numerical comparison, if feasible, is also encouraged, as this may better highlight the novelty of this paper’s contributions beyond merely applying existing Hessian-free methods to the Machine Unlearning field.

3. While the authors present some experimental results, they lack diversity in dataset selection and only test the approach on ~5 datasets. The efficacy of this unlearning mechanism remains unclear in large-scale or high-dimensional applications where computational efficiency is critical.


4. The writing quality of this work is limited and the presentation should be improved, for instance

- the (2) is simply one case of (1) by replacing the $\mathcal{D}$ with empirical distribution. i.e., the authors could remove (2) for simplicity, or just start from the (1)

- I think the (3) should be written as $\mathbf{w}\_{e, b+1} \leftarrow \mathbf{w}\_{e, b}-\eta\_{e, b} \sum\_{i \in \mathcal{B}_{e, b}} \nabla \ell\left(\mathbf{w}\_{e, b} ; z\_i\right),$ since the linear scaling rule (Goyal et al. 2017) is introduced later in line 163.

- line 156 and algo. 1: as your notation, the total epochs and batches would be $E+1$ and $B+1$. So as your complexity in section 4.4

- when removing the $u\_j$, why the normalization constant of $\eta$ in your (5) is $ \mathcal{B}\_{e, b(u\_j)}$ instead of $ \mathcal{B}\_{e, b(u\_j)}-1$?

- definition 1: how can you ask a learning algorithm within the (solution) parameter space $\mathcal{W}$? Please revise the definition or rephrase your wording

- in your lemma 2, I don't think there exists a valid $G$ such that $G=\max \left\\|\nabla \ell\left(\mathbf{w}_{e, b} ; z\right)\right\\|<\infty$. This should be a consequence of the assumption that the grad of $l$ is uniformly upper bounded. Otherwise, this could be derived by your assumption 1, which is imposed later

- Theorem 4 should be $B=\left\lceil \frac{n}{|\mathcal{B}|} \right\rceil$

- if the intent is for the product to go in reverse order in line 10 of algo. 1, i.e. $\prod_{k=E}^e \prod_{b=B-1}^{b(u)+1}$, the notation should ideally be clarified in the text to avoid misunderstandings

- font size in figures e.g., 1, 2, 7, is too small and needs to be enlarged for clarity

- figure 4 caption: "a comparison comparison between"

**Questions:**

Please refer to weaknesses.

---

> ### Author Response · Authors · 2024-11-21
> **Official Comment by Authors (1)**
>
> We thank the reviewer for all of your valuable comments. We sincerely hope that this revised manuscript has addressed all your comments and suggestions. We are more than willing to provide further clarification or address any additional concerns.
>
> We provide our responses to the comments, denoted by **[W]** for weaknesses, **[Q]** for questions, **[L]** for lines in our manuscript, and **[R]** for references.
>
> **W1:** *Theoretical guarantees hinge on assumptions of convexity.*
>
> >- Unlike previous works, the key distinction of our method lies in its ability to handle non-convex scenarios. This is achieved by avoiding the need for Hessian inversion, meaning that strong convexity is not required to ensure the positive definiteness of the Hessian. Therefore, our unlearning method and its theoretical guarantees do not rely on the assumption of convexity.
> >- In deriving other performance analyses involving learning process conclusions, such as generalization guarantee, we inherit the convexity assumption from prior unlearning works. As noted in **Line 97** and **L308-L311**, we clarified in the initial manuscript that our proposed method and  theoretical unlearning guarantees apply to both convex and non-convex settings, but do not extend to the theoretical analysis of the learning process.
> >
> >In summary, our contributions for non-convex assumption primarily focus on the unlearning domain. For the theoretical analysis involving the learning process conclusions, we have built upon previous unlearning work, adopting simple and rigorously assumed conclusions such as Lemma 9. As a future extension, we plan to provide non-convex theoretical conclusions for generalization guarantee in later versions or follow-up works.
>
> **W2:** *Authors should connect the ideas presented with a broader body of Hessian-free optimization work in classical parametric optimization.*
>
> >We appreciate the reviewer’s suggestion. Here, we briefly clarify the connection between our ideas and Hessian-free optimization work. Our unlearning method and previous Hessian-free optimization approaches are parallel lines of research, such as using conjugate gradient with HVP to perform sub-optimization. Different algorithms that use many of the same key principles have appeared in the literature of various communities under different names such as Newton-CG, CG-Steihaug, Newton-Lanczos, and Truncated Newton, as stated in **R11**. But in any case, the learning and unlearning processes are distinct from each other. The commonality lies in leveraging automatic differentiation tools to compute HVP, avoiding explicit computation of the Hessian, as this provides a numerically stable and precise way to compute the desired directional derivative.
>
> **W3:** *Lack diversity in dataset selection and only test the approach on ~5 datasets.*
>
> >- We greatly appreciate the valuable suggestion. In addition to MNIST, FMNIST, CIFAR10, CelebA, and LFW, we have conducted experiments on additional datasets for sufficient diversity in dataset selection, including HAPT, Adult, Wine, and Obesity. Our initial manuscript included digit, clothing, object, gender, and human face classification, and we further provided experiments on human activity recognition, income, wine quality, and estimation of obesity levels. These will demonstrate sufficient diversity in our dataset selection. The results show that we achieve good performance across different datasets, demonstrating the advantages of our algorithm. Please refer to the anonymous [repository link](https://github.com/Anonymous202401/If-Recollecting-were-Forgetting) provided in **L107**  for detailed code and figures.
> >- Besides, we would like to clarify our experimental contributions. Previous theoretical works **R1, R3** did not include experiments, and **R2** only conducted  experiments with binary LR. Even the recent certified unlearning works **R4** and **R5** have experimented with convex models. The datasets used in these works were limited to binary MNIST and CIFAR10. While providing different theoretical insights, the extensive experiments led to space limitations in our writing. This is also why the reviewer raised the concern about the font size in the figure 1 being too small. We have followed the suggestion and increased the font size. However, for the additional experiments suggested by reviewer on the more datasets, we have to place these experiments in the appendix.

---

> > ### Author Response · Authors · 2024-11-21
> > **Official Comment by Authors (2)**
> >
> > **W4:**  *Presentation should be improved.*
> >
> > Thanks for your careful checks. We tried our best to improve the manuscript and made some changes to the manuscript. And here we did not list the changes but marked in blue in the revised paper. Below are our detailed responses.
> >
> > >**W4.1:** *Considering population risk (1) and empirical risk (2); Author could remove (2) for simplicity.*
> > >
> > >>We respectfully disagree with the suggestion to remove the definition of empirical risk in equation (2). Providing rigorous definitions of both population risk (1) and empirical risk (2) is a common practice in many works, including the unlearning literature **R1, R2, R3**. These definitions also facilitate the analysis of generalization guarantees and the empirical risk minimizer in the subsequent sections.
> > >
> > >**W4.2:** *The (3) should be rewritten since linear scaling rule is introduced.*
> > >
> > >>We did not violate the linear scaling rule in **R9**. We wrote $w_{e,b+1} = w_{e,b} + \frac{\eta_{e,b}}{|\mathcal{B} _ {e,b}|} \sum \nabla \ell(w_{e,b})$ because the definition of  batch size $|\mathcal{B} _ {e,b}|$ is needed in later  analysis.
> > >
> > >**W4.3:** *If intent is for the product to go in reverse order in line 10 of algo. 1, the notation should ideally be clarified in the text.*
> > >
> > >> We greatly appreciate the reviewer’s suggestion. To avoid any misunderstanding, we have revised the manuscript and added the necessary explanation in Appendix C.1.
> > >
> > >**W4.4:** *In your Lemma 2, I don't think there exists a valid $G$.*
> > >
> > >> Thanks for your comment. We would like to explain that we can obtain a valid $G$ typically by directly computing the norm of the gradient. If we need gradient clipping after computing the norm  to prevent divergence, then $G$ can be the clipping threshold.
> > >
> > >**Q4.4:** *When removing the $u  _  j$, why the normalization constant remain unchanged?*
> > >
> > >> The constant remains unchanged to keep the ratio between the step size and the batch size constant. The specific reason is as follows:
> > >
> > >> - From the perspective of the linear scaling rule, the same ratio of $\eta$ to $|\mathcal{B} _ {e,b}|$. ensures there is no loss of accuracy.
> > >> - From the perspective of reweighting in Appendix C.1, this ensures that the weighting of the remaining samples during each model update remains consistent.
> > >
> > >**Q4.5:** *How can you ask a learning algorithm within the (solution) parameter space?*
> > >
> > >> Thank you for your suggestions. We have fixed the typo in the definition and changed **L178** to $\Omega: \mathcal{Z}^n \rightarrow \mathcal{W}$.
> > >
> > >**W4.6:** *Typo in (1) **L156** for total epochs, batches   (2) **L289** for $B=\left\lceil\frac{n}{|\mathcal{B}|}\right\rceil$ (3) **L1562** for Figure 4 caption.*
> > >
> > >>Thanks for your correction. We have
> > >
> > >>- updated **L156** and Algorithm, changing $E$ and $B$ to $E+1$ and $B+1$.
> > >>- added ceiling symbol to $B$.
> > >>- removed redundant "comparison" in Appendix Fig. 4 caption.

---

> ### Author Response · Authors · 2024-11-25
> **Follow-up on rebuttal and a kind reminder**
>
> Dear Reviewer ia1D,
>
> We would like to express our gratitude for your constructive suggestions and thoughtful reviews, which have proven invaluable in enhancing the quality of our paper. As a follow-up to our rebuttal, we would like to kindly remind you that the deadline for discussion closure is rapidly approaching.
>
> During this open response period, we aim to engage in discussions, address any further inquiries, and further enhance the overall quality of our paper. We would appreciate it if you could confirm whether you have had the opportunity to review our rebuttal, in which we made concerted efforts to address all of your concerns. It is of utmost importance to us that our responses have been thorough and persuasive.  We have also attached tables summarizing the experimental results for diverse dataset selection for your convenience.
>
> Should you require any additional information or clarification, please do not hesitate to let us know. Thank you once again for your time and valuable consideration.
>
> Best regards,
>
> Authors
>
> (Submission Number: 830)
>
>
> | Remove 5% Wine Dataset | Distance (±std)  | Unlearning time  (Sec) | Storage  (MB) | Precomputing time (Sec) |
> | ---------------------- | ---------------- | ---------------------- | ------------- | ----------------------- |
> | **IJ**                 | 1.33 (±2.68)     | 0.31                   | 9.12          | 155.01                  |
> | **NU**                 | 0.71 (±0.66)     | 7.55                   | 9.12          | 153.16                  |
> | **Proposed**           | **0.04 (±0.01)** | **0.0004**             | **1.06**      | **11.56**               |
>
> | Remove 10% Wine Dataset | Distance (±std)  | Unlearning time (Sec) | Storage (MB) | Precomputing time (Sec) |
> | ----------------------- | ---------------- | --------------------- | ------------ | ----------------------- |
> | **IJ**                  | 0.65 (±0.48)     | 0.31                  | 9.12         | 155.01                  |
> | **NU**                  | 0.88 (±1.61)     | 14.34                 | 9.12         | 153.16                  |
> | **Proposed**            | **0.09 (±0.02)** | **0.0004**            | **1.06**     | **11.56**               |
>
>
>
> | Remove 5% Obesity Dataset | Distance (±std)  | Unlearning time  (Sec) | Storage  (MB) | Precomputing time (Sec) |
> | ------------------------- | ---------------- | ---------------------- | ------------- | ----------------------- |
> | **IJ**                    | 0.93 (±0.24)     | 1.98                   | 14.21         | 2,240.16                |
> | **NU**                    | 0.91 (±0.21)     | 106.50                 | 14.21         | 2,228.83                |
> | **Proposed**              | **0.58 (±0.15)** | **0.0004**             | **13.23**     | **102.06**              |
>
> | Remove 10% Obesity Dataset | Distance (±std)  | Unlearning time  (Sec) | Storage  (MB) | Precomputing time (Sec) |
> | -------------------------- | ---------------- | ---------------------- | ------------- | ----------------------- |
> | **IJ**                     | 1.18 (±0.19)     | 1.98                   | 14.21         | 2,240.16                |
> | **NU**                     | 1.35 (±0.49)     | 212.66                 | 14.21         | 2,228.83                |
> | **Proposed**               | **0.83 (±0.11)** | **0.0004**             | **13.23**     | **102.06**              |

---

> ### Comment · Reviewer_ia1D · 2024-11-25
> **Thank you for your rebuttal**
>
> I understand that your new algorithm can be applied to *both* convex and non-convex settings, but much of the analysis is built on convexity, which does present a limitation. It would be nice if you could further discuss that in your work.
>
> Regarding **W4.4**, my main point is that Assumption 1 should be stated before Lemma 2 rather than in its current position. Otherwise, the upper bound may not hold (unless you explicitly assume it, but the lemma appears to be presented as a result rather than a condition).
>
> That said, given the overall responses, I have revised my stance.

---

> ### Author Response · Authors · 2024-11-27
> **Acknowledgment of the Reviewer’s Constructive Comments**
>
> We appreciate the reviewer’s understanding of the contribution our unlearning method makes in bridging the gap in implementing non-convex and non-convergence conditions. We also sincerely thank the reviewer for improving the rating.  Below is our response to address your remaining concerns.
>
> > - We acknowledge that, while we propose a new method that does not rely on convexity, our analysis builds upon the theoretical framework of previous works for comparison purposes. These prior certified unlearning works, both in terms of methodology and theoretical analysis, were limited to convex settings, which is why much of our theoretical results rely on convexity—except for our unlearning-related conclusions. We briefly discuss the limitation in the revised manuscript **(L308, L934)**. In the future, we plan to extend the existing theoretical framework, though this may require revisiting existing methods and frameworks, or introducing new assumptions to address more complex scenarios.
> >
> > - In response to the reviewer’s concern on **W4.4**, we have, following the reviewer’s suggestion, stated Assumption 1 before Lemma 2 in order to ensure that the upper bound holds.
>
>  If the reviewer has any further **Q**uestions or **W**eakness that need to be addressed, we would be glad to provide any clarification.

---

### Official Review · Reviewer_CNQV · 2024-11-06

**Soundness:** 2
**Presentation:** 3
**Contribution:** 3
**Rating:** 6
**Confidence:** 3

**Summary:**

The paper proposes a new unlearning algorithm which extracts second-order information in a Hessian-free manner without the need to assume strong convexity. The key idea is to track and remove the impact of a specific sample in the entire update trajectory of the model, which is called affine stochastic recursion between the retrained and learned models in this work. It provides theoretical guarantees on generalization, deletion capacity, and space/time complexities. Experiments are conducted to demonstrate the superiority of the proposed algorithm.

**Strengths:**

Strengths include:
1) The paper proposes affine stochastic recursion to pinpoint the overall impact of a specific sample on the learned model.
2) It makes the unlearning process efficient via the Hessian-free computation.
3) It provides theoretical guarantees that cover generalization, deletion capacity, and space/time complexities.
4) Experiments are provided to demonstrate the advantages of the proposed unlearning algorithm.

**Weaknesses:**

Weaknesses include:
1) The recollection matrix M looks incorrect, based on the derivation given in Appendix C.1
2) Limitations of the algorithms are not discussed. For example, the proposed algorithm may not perform well on large-scale datasets given the quadratic time complexity in data size.

**Questions:**

1) Could you explain why NS and IJ can't utilize HVP? Although they involve Hessian inverse, it could be approximated by using like least-square which essentially does HVP as well.
2) Regarding the correlation metric on loss change, could you tell us what stopping rule you use while calculating those loss changes across different algorithms? I feel this is important for gauging performance.
3) It is unclear whether fine-tuning was used for each of the algorithms in experiments.

---

> ### Author Response · Authors · 2024-11-21
> **Official Comment by Authors**
>
> We sincerely appreciate your support and constructive suggestions. We have made every effort to revise the manuscript based on your valuable and constructive suggestions.
>
> We provide our responses to the comments, denoted by **[W]** for weaknesses, and **[Q]** for questions. The references, denoted by **[R]**, are provided in the General Response. And we use **[L]** to represent lines in our manuscript.
>
> **W1:** *The recollection matrix M looks incorrect.*
>
> >We thank the reviewer's comments, and would like to clarify that the recollection matrix $\mathbf{M}$ is correct because its product requires multiplying $\mathbf{I}-\eta \mathbf{H}$ of the most recent update with those of the past updates.., such as $(\mathbf{I} - \eta  _  {E,B} \mathbf{H}  _  {E,B})(\mathbf{I} - \eta  _  {E,B-1}\mathbf{H}  _  {E,B-1})...(\mathbf{I} - \eta  _  {E,b(u)+1} \mathbf{H}  _  {E,b(u)+1})$. This means the multiplication should begin with the most recent update and continue backward through the previous gradient updates. Can we politely ask what makes the reviewer think that $\mathbf{M}$ seems incorrect?  We greatly appreciate the reviewer’s concern, and to avoid any misunderstandings, we have revised the manuscript and added the necessary explanation in Appendix C.1.
>
> **W2:** *Limitations of the algorithms are not discussed, and the proposed algorithm may not perform well on large-scale datasets.*
>
> >In Appendix B.1 of our initial submission, we provide a detailed description of the limitations of our algorithm and corresponding solutions, including the reviewer’s concern about large-scale datasets. For example, we can consider maintaining vectors for $k$ users instead of $n$ data points ($k \ll n$), which would significantly reduce both computation time and storage, without relying on quadratic time complexity in the data size.
>
> **Q1:** *Why NS and IJ can't utilize HVP?*
>
> >We sincerely thank the reviewers for their insightful questions and have provided the following clarifications in response.
> >
> >- The key reason is that, to handle deletion requests from unknown users, both NS and IJ algorithms require explicit pre-computing of the Hessian and its inverse before an unlearning request arrives, which makes HVP impossible.
> >
> >  Let us recall the technique details of NS and IJ.
> >
> >  For NS, the update step is: $ \frac{1}{n-m} {   \Big( \frac{1}{n-m}\sum_{i=1}^n   \nabla^2 \ell (\hat{\mathbf{w}}; z_i) -\sum_{j \in U}  \nabla^2 \ell (\hat{\mathbf{w}};u_j )\Big) }^{-1}\sum_{j \in U}   \nabla \ell (\hat{\mathbf{w}}; u_j).$
> >
> >  For IJ, the update step is:  $ \frac{1}{n} {   (\frac{1}{n} \sum_{i=1}^n   \nabla^2 \ell (\hat{\mathbf{w}}; z _ i) ) }^{-1}  \nabla \ell (\hat{\mathbf{w}}; u _ j).$
> >
> >  However, for the Hessian $\sum_{j \in U} \nabla^2 \ell (\hat{\mathbf{w}}; u _ j)$ in NS and the gradient $\nabla \ell (\hat{\mathbf{w}}, u_j)$ in both NS and IJ, **(i)** the forgetting sample $u _ j$ is unknown before the deletion request arrives, and **(ii)** the model $\hat{\mathbf{w}}$ will be deleted after processing a deletion request, while the subsequent unlearned model is also unknown because the forgetting sample $u_j$ is unpredictable.
> >
> >  This is also a unique advantage of our method, which possesses additivity in Theorem 1 **(L245)**, enabling it to efficiently handle multiple deletion requests  in an online manner. We will make this point clearer in the revised manuscript to better highlight our contributions.
> >
> >- In addition, inaccurate techniques (e.g., as mentioned by the reviewer, it could be approximated by using methods like least squares) would also render the theoretical bounds of previous works not as strong as the techniques proposed in NS and IJ. These operations require the previous work to re-derive the upper bound, resulting in the loss of the original $\mathcal{O}(m^2/n^2)$ approximation error advantage.
> >
> >Given the above reasons, these approximate techniques are more suitable for non-privacy scenarios involving heuristic methods that do not require certified theoretical guarantees, such as bias removal based on influence function in **R8** .
>
> **Q2:** *What stopping rule do you use while calculating those loss changes across different algorithms?*
>
> >We trained the model normally until the accuracy no longer increased significantly (e.g., when the model accuracy stabilized or started to decline), at which point we considered the model to have nearly converged, and the training needed to stop.
>
> **Q3:** *Whether fine-tuning was used for each of the algorithms in experiment?*
>
> >We only used fine-tuning in Figure 3 of Appendix B.4. All other experiments are based on Algorithm 1 without fine-tuning. We briefly introduce fine-tuning in Appendix B.4 because we believe our method can serve as a heuristic algorithm in non-privacy scenarios, where theoretical certified guarantees are not necessary.

---

> > ### Comment · Reviewer_CNQV · 2024-11-25
> > **Recollection matrix**
> >
> > My understanding is that recursion over Eq. (35) yields the following:
> > $$
> > \\begin{array}{l}
> > \quad \mathbf{w}\_{E,B}^{-u} - \mathbf{w}\_{E,B}  \newline \approx
> > \prod\_{b=B-1}^{0}(\mathbf{I}-\eta\_{E,b}\mathbf{H}\_{E,b}) (\mathbf{w}\_{E-1,B}^{-u} - \mathbf{w}\_{E-1,B}) +
> > \prod\_{b=B-1}^{b(u)+1}(\mathbf{I}-\eta\_{E,b}\mathbf{H}\_{E,b})\frac{\eta\_{E,b(u)}}{|\mathcal{B}\_{E,b(u)}|} \nabla l(\mathbf{w}\_{E,b(u)};u) \newline \approx \prod\_{b=B-1}^{0}(\mathbf{I}-\eta\_{E,b}\mathbf{H}\_{E,b}) \prod\_{b=B-1}^{0}(\mathbf{I}-\eta\_{E-1,b}\mathbf{H}\_{E-1,b}) (\mathbf{w}\_{E-2,B}^{-u} - \mathbf{w}\_{E-2,B})\newline\quad+\prod\_{b=B-1}^{0}(\mathbf{I}-\eta\_{E,b}\mathbf{H}\_{E,b})\prod\_{b=B-1}^{b(u)+1}(\mathbf{I}-\eta\_{E-1,b}\mathbf{H}\_{E-1,b})\frac{\eta\_{E-1,b(u)}}{|\mathcal{B}\_{E-1,b(u)}|} \nabla l(\mathbf{w}\_{E-1,b(u)};u)\newline\quad+
> > \prod\_{b=B-1}^{b(u)+1}(\mathbf{I}-\eta\_{E,b}\mathbf{H}\_{E,b})\frac{\eta\_{E,b(u)}}{|\mathcal{B}\_{E,b(u)}|} \nabla l(\mathbf{w}\_{E,b(u)};u)\approx\cdots\newline\approx\prod\_{e=E}^{1}\prod\_{b=B-1}^{0}(\mathbf{I}-\eta\_{e,b}\mathbf{H}\_{e,b}) (\mathbf{w}\_{0,B}^{-u} - \mathbf{w}\_{0,B})\newline\quad+\prod\_{e=E}^{2}\prod\_{b=B-1}^{0}(\mathbf{I}-\eta\_{e,b}\mathbf{H}\_{e,b})\cdot\prod\_{b=B-1}^{b(u)+1}(\mathbf{I}-\eta\_{1,b}\mathbf{H}\_{1,b})\frac{\eta\_{1,b(u)}}{|\mathcal{B}\_{1,b(u)}|} \nabla l(\mathbf{w}\_{1,b(u)};u)\newline\quad+\prod\_{e=E}^{3}\prod\_{b=B-1}^{0}(\mathbf{I}-\eta\_{e,b}\mathbf{H}\_{e,b})\cdot\prod\_{b=B-1}^{b(u)+1}(\mathbf{I}-\eta\_{2,b}\mathbf{H}\_{2,b})\frac{\eta\_{2,b(u)}}{|\mathcal{B}\_{2,b(u)}|} \nabla l(\mathbf{w}\_{2,b(u)};u)\newline\quad+\cdots\newline\quad+\prod\_{e=E}^{E}\prod\_{b=B-1}^{0}(\mathbf{I}-\eta\_{e,b}\mathbf{H}\_{e,b})\cdot\prod\_{b=B-1}^{b(u)+1}(\mathbf{I}-\eta\_{E-1,b}\mathbf{H}\_{E-1,b})\frac{\eta\_{E-1,b(u)}}{|\mathcal{B}\_{E-1,b(u)}|} \nabla l(\mathbf{w}\_{E-1,b(u)};u)\newline\qquad\qquad\qquad\qquad\qquad\qquad\quad\\;+
> > \prod\_{b=B-1}^{b(u)+1}(\mathbf{I}-\eta\_{E,b}\mathbf{H}\_{E,b})\frac{\eta\_{E,b(u)}}{|\mathcal{B}\_{E,b(u)}|} \nabla l(\mathbf{w}\_{E,b(u)};u)\newline\approx\prod\_{e=E}^{0}\prod\_{b=B-1}^{0}(\mathbf{I}-\eta\_{e,b}\mathbf{H}\_{e,b}) (\mathbf{w}\_{0,0}^{-u} - \mathbf{w}\_{0,0})\newline\quad+\prod\_{e=E}^{1}\prod\_{b=B-1}^{0}(\mathbf{I}-\eta\_{e,b}\mathbf{H}\_{e,b})\cdot\prod\_{b=B-1}^{b(u)+1}(\mathbf{I}-\eta\_{0,b}\mathbf{H}\_{0,b})\frac{\eta\_{0,b(u)}}{|\mathcal{B}\_{0,b(u)}|} \nabla l(\mathbf{w}\_{0,b(u)};u)\newline\quad+\prod\_{e=E}^{2}\prod\_{b=B-1}^{0}(\mathbf{I}-\eta\_{e,b}\mathbf{H}\_{e,b})\cdot\prod\_{b=B-1}^{b(u)+1}(\mathbf{I}-\eta\_{1,b}\mathbf{H}\_{1,b})\frac{\eta\_{1,b(u)}}{|\mathcal{B}\_{1,b(u)}|} \nabla l(\mathbf{w}\_{1,b(u)};u)\newline\quad+\prod\_{e=E}^{3}\prod\_{b=B-1}^{0}(\mathbf{I}-\eta\_{e,b}\mathbf{H}\_{e,b})\cdot\prod\_{b=B-1}^{b(u)+1}(\mathbf{I}-\eta\_{2,b}\mathbf{H}\_{2,b})\frac{\eta\_{2,b(u)}}{|\mathcal{B}\_{2,b(u)}|} \nabla l(\mathbf{w}\_{2,b(u)};u)\newline\quad+\cdots\newline\quad+\prod\_{e=E}^{E}\prod\_{b=B-1}^{0}(\mathbf{I}-\eta\_{e,b}\mathbf{H}\_{e,b})\cdot\prod\_{b=B-1}^{b(u)+1}(\mathbf{I}-\eta\_{E-1,b}\mathbf{H}\_{E-1,b})\frac{\eta\_{E-1,b(u)}}{|\mathcal{B}\_{E-1,b(u)}|} \nabla l(\mathbf{w}\_{E-1,b(u)};u)\newline\qquad\qquad\qquad\qquad\qquad\qquad\quad\\;+
> > \prod\_{b=B-1}^{b(u)+1}(\mathbf{I}-\eta\_{E,b}\mathbf{H}\_{E,b})\frac{\eta\_{E,b(u)}}{|\mathcal{B}\_{E,b(u)}|} \nabla l(\mathbf{w}\_{E,b(u)};u).
> > \\end{array}
> > $$
> > Thus, we have that
> > $$
> > \mathbf{w}\_{E,B}^{-u} - \mathbf{w}\_{E,B} \approx \sum\_{e=1}^{E+1}\mathbf{M}\_{e,b(u)}\nabla l(\mathbf{w}\_{e-1,b(u)};u),
> > $$
> > where
> > $$
> > \mathbf{M}\_{e,b(u)}=\frac{\eta\_{e-1,b(u)}}{|\mathcal{B}\_{e-1,b(u)}|}\prod\_{k=E}^{e}\prod\_{b=B-1}^{0}(\mathbf{I}-\eta\_{k,b}\mathbf{H}\_{k,b})\cdot\prod\_{b=B-1}^{b(u)+1}(\mathbf{I}-\eta\_{e-1,b}\mathbf{H}\_{e-1,b})
> > $$ with
> > $\prod\_{k=E}^{e}\prod\_{b=B-1}^{0}(\mathbf{I}-\eta\_{k,b}\mathbf{H}\_{k,b})=:\mathbf{I}
> > $ for $e=E+1$.
> > This is different from Eq.(36), if my derivation is correct.

---

> ### Author Response · Authors · 2024-11-25
> **Sincere Gratitude for the Correction from Reviewer CNQV**
>
> Thanks  a lot for Reviewer CNQV careful proofreading and for the time you have dedicated to our work. After considering your feedback, we realized that there was indeed an unintended error in the notation of the $\mathbf{M} _ {e, b(u)}$ in Eq. 10. Specifically, we mistakenly formulated the recursive product $\mathbf{M} _ {e, b(u)}$. We have made the necessary corrections to the related content (**L221**, and **L1075**, **L1180**  in the appendix) marked in blue in the revised paper. This correction does not affect the other conclusions of our work. Below are the details of our revisions:
>
> > We removed the unnecessary and cumbersome product symbols. In the latest manuscript, we define the recollection matrix as  $\mathbf{M} _ {e,b(u)} := \frac{\eta _ {e,b(u)}}{|\mathcal{B} _ {e,b(u)}|} \hat{\mathbf{H}} _ {E,B-1 \rightarrow e,b(u)+1 }$, where $\hat{\mathbf{H}} _ {E,B-1 \rightarrow e,b(u)+1 }= (\mathbf{I}- {\eta _ {E,B-1}}\mathbf{H} _ {E,B-1})\cdot(\mathbf{I}- {\eta _ {E,B-2}}\mathbf{H} _ {E,B-2})...(\mathbf{I}- {\eta _ {e,b(u)+1}}\mathbf{H} _ {e,b(u)+1})$ which represents the product of $\mathbf{I}-\eta \mathbf{H}$ from $E$-th epoch's $B-1$-th update to $e$-th epoch's $b(u)+1$-th update.  The improved notation of $\mathbf{M} _ {e,b(u)}$ in the revised version now effectively and clearly conveys its meaning.
>
> We sincerely appreciate your thorough proofreading once again, as well as the meaningful suggestions and questions you raised earlier.  If there are any other questions, we would be more than glad to address any further requests from the reviewer.

---

### Official Review · Reviewer_kjAJ · 2024-11-08

**Soundness:** 2
**Presentation:** 3
**Contribution:** 2
**Rating:** 6
**Confidence:** 3

**Summary:**

This paper addresses the challenge of certified unlearning, where models are required to forget information at the request of data providers. The authors introduce a novel approach leveraging details tracked during model training to approximate how the training process would have proceeded without the data marked for deletion. Notably, the proposed method circumvents the need for full Hessian computations or inversion by using Hessian-vector products for second-order information. Additionally, it does not assume that the original model is an empirical risk minimizer. The authors' theoretical analysis argues that their method offers enhanced unlearning guarantees, efficient storage and precomputation, faster data deletion, and improved generalization bound, particularly for overparameterized models. Empirical results support the approach's claim of rapid unlearning execution.

**Strengths:**

1. The idea of tracking algorithmic updates during training to facilitate unlearning is straightforward yet impactful, with sound theoretical analysis for unlearning privacy guarantee and descent empirical evaluation.
2. Removing the assumption that the initial learned model must be an empirical risk minimizer is significant for the practical applicability of the method.
3. The claimed efficiency improvements are particularly relevant for overparameterized deep models where the batch size is comparable to the number of training epochs, probably covering a substantial portion of common machine learning models.

**Weaknesses:**

1) The claim regarding the efficiency of precomputation and storage (lines 378–387) hinges on two critical assumptions: (i) the model's parameter size $d$ is significantly greater than the training data size $n$, and (ii) the number of epochs $E$ is of the same order as the batch size $|B|$. While the authors state (ii) as typical ("Typically, $E$ and $|B|$ are of the same order"), a reference would strengthen this claim. Moreover, the assumption may not hold in scenarios such as online learning or streaming applications. Qualifying its generality and highlighting this as an assumption similar to Line 299 would be beneficial and avoid misleading readers.

2) The generalization analysis in Section 4.2 considers strong convex loss functions and focuses on excess risk bound. Excess risk bound consist of two terms: the first term comes from (a) the excess risk of the empirical risk minimizer, and the second term comes from (b) unlearning error, (c) optimization error, and (d) the noise for obfuscation (Line 14 of Algorithm 1). There are at least two problems with this analysis and theorem statement.

2.1 In strong convex settings, the assumption that $E$ and $|B|$ are comparable can seem more questionable, especially as the excess risk bound expressed in big-O notations (Line 319) uses this assumption. It might be okay to make this assumption for controlling (b), but $E$ and $B$ also affects (c), i.e. whether the term $C_4$ in Equation 74 is small enough, which conceptually (roughly) translates to whether $w_{E, B}$ is a good empirical minimizer or not. It seems problematic to assert that the number of epoch required for convergence to empirical risk minimizer is the same order of magnitude as batch size.

2.2 The first term comes from Lemma 9, which cites Shalev-Shwartz et al. (2009). The latter assumes i.i.d. of samples, which should translate to i.i.d. of the set $U$.

**Questions:**

1. In the abstract (Line 017) and introduction (Line 073), the authors claim that previous work requires strong convexity, while in Line 66 the authors said previous work requires convexity. It seems to me that convexity suffices in Sekhari et al. (2021) because unlearning for a convex loss function can be reduced to a problem with strong convex loss. Could the authors clarify what assumptions are needed in previous work?
2. In Assumption 1 (Line 299), is the loss assumed to be jointly convex in both $z$ and $w$, only $w$, or only $z$? Similar questions for Lipschitzness and smoothness.

---

> ### Author Response · Authors · 2024-11-21
> **Official Comment by Authors**
>
> We sincerely appreciate your support and constructive suggestions. We have made every effort to revise the manuscript based on your valuable and constructive suggestions.
>
> We provide our responses to the comments, denoted by **[W]** for weaknesses, and **[Q]** for questions. The references, denoted by **[R]**, are provided in the General Response. And we use **[L]** to represent lines in our manuscript.
>
> **W1:** *Qualifying the generality of complexity and highlighting this as an assumption similar to Line 299 would be beneficial and avoid misleading readers.*
>
> >Thank you for your insights. Based on the reviewer's suggestions, we have tried our best to revise the manuscript for descriptions and assumptions similar to **L299**, and provide additional references, such as **R10**, to demonstrate that the ratio of $E$ to $B$ is generally small. Additionally, the feedback has inspired us to provide a more precise description of the impact of $E$ and $|B|$ on complexity in our revised version. In fact, the ratio of $E$ to $|B|$ is much smaller than $d$. Therefore, our method works for most scenarios, except in the cases where SGD is over-iterating the epoch to levels in the dimension $d$.
>
> **W2:** (1) *Shalev-Shwartz et al. (2009) assumes i.i.d. of samples, which should translate to i.i.d. of the set.* **AND** (2)  *The term C4 in Equation 74 is small enough which conceptually (roughly) translates to whether $w _ {E,B}$ is a good empirical minimizer or not. It seems problematic to assert that the number of epochs required for convergence to empirical risk minimizer is the same order of magnitude as batch size.*
>
> >- For (1): We greatly appreciate the constructive suggestions from the reviewer, and we have completed the missing details in Lemma 9 in the revised version.
> >
> >- For (2): We do not require $w_{E,B}$ to be an empirical risk minimizer in Theorem 6 of Section 4.2. We appreciate the reviewers for spending time on the proofs in our paper and for outlining our proof sketch in the review.
> >
> > We would like to further clarify that (a) represents the excess risk between the retrained empirical risk minimizer $\hat{w}^{-U}$ and ${w}^*$, while (c) denotes the optimization error between the retrained model ${w}^{-U} _ {E,B}$ and $\hat{w}^{-U}$. Using (a) and (c), we can bound the excess risk of the retrained model ${w}^{-U} _ {E,B}$ relative to ${w}^*$,  Note that at this point, we no longer require convergence to the empirical risk minimizer. Furthermore, given (b) and (d), which represent the approximation error and noise of the unlearning model $\tilde{w}^{-U} _ {E,B}$, we can bound the excess risk of the unlearned model $\tilde{w}^{-U} _ {E,B}$ relative to ${w}^*$ in Theorem 6. Therefore, in Theorem 6, we do not need the term C4 to be sufficiently small as it also satisfies  $\mathcal{O}(\rho^n)$, and we do not require $w  _  {E,B}$ to be an empirical risk minimizer.  This is a key distinction that sets our Theorem 6 apart from previous works, as the generalization guarantees in earlier works are derived through implicit assumption about the empirical risk minimizer.
>
> **Q1:** *Convexity suffices in **R1** because unlearning for a convex loss can be reduced to a problem with strong convex loss.* **AND** *Could the authors clarify what assumptions are needed in previous work?*
>
> >- First of all, previous algorithms requires strong convexity. The reason that  previous work requires strong convexity is to guarantee that the Hessian is positive definite, thus ensuring its invertibility.  Simply claiming to require convexity is insufficient because a semi-positive definite Hessian still cannot guarantee invertibility, although **R1** can use regularization techniques (such as L2 regularization) to make a logistic regression loss function strongly convex.
> >- We thank the reviewer's comments and would like to clarify that the assumptions in Assumption 1 is consistent with previous works **R1, R2**. In addition to this, previous works require the implicit assumption of a unique empirical risk minimizer.
>
> **Q2:** *Is the loss assumed to be jointly convex in both $z$ and $w$?*
>
> >All assumptions in **L299**  are made solely with respect to $w$. We appreciate the reviewer’s valuable question and we provide a more detailed explanation of the definition of Assumption 1 in the appendix (**L1389-L1403**) in revised manuscript.

---

> > ### Comment · Reviewer_kjAJ · 2024-11-22
> >
> > Thanks for the authors' detailed response. I am convinced by the response to W2 and I appreciate that the authors incorporate feedback from W1 into the paper.
> >
> > For Q1, I would still argue that for convex functions, the fact that [R1] uses regularization as a technique shouldn't be interpreted as their work requires the strong convexity assumption; after all, their unlearning and generalization guarantees are stated for the original convex function before adding regularization. For Q2, I would still suggest that the authors use "convex in w" in the main body for the statement to be correct/ precise.
> >
> > Similar to Reviewers t2wR and CNQV, I also miss the reasons why neither [R1] nor [R2] can be implemented using only Hession-vector products by using, for example, the conjugate gradient method. After learning how prior works _must_ be Hessian-based, I am now able to see the contributions underlying the Hessian-free property more clearly. I suggest the authors also highlight it in the paper, as this point is repeatedly missed.

---

> > > ### Author Response · Authors · 2024-11-24
> > > **Reply to Reviewer kjAJ**
> > >
> > > We appreciate the reviewer’s feedback, especially the suggestions regarding the description of prior works. We agree with the perspective that a convex loss function can be reduced to a problem with a strongly convex loss. To avoid similar confusion as raised in **Q1**, we have revised the manuscript to modify the assumption of strong convexity in prior work to convexity in the abstract (**L17**), introduction (**L73**) as suggested by the reviewer, as well as in the related works section (**L130**). Additionally, we have followed the reviewer's suggestion to add "in $\mathbf{w}$" to Assumption 1 (**L299**) for the statement to be precise.
> > >
> > > We also appreciate the reviewer’s suggestion to highlight the contributions of our Hessian-Free approach, which was omitted and missed in our initial manuscript.  In the revised version, we have explicitly emphasized the advantages of our Hessian-Free approach and its comparison to Hessian-based methods in handling multiple deletion requests. This revision better highlights the contribution of our work, as suggested by the reviewer. The specific revisions are as follows:
> > >
> > > > - We added more descriptions of previous works in the Introduction **(L68, L90)**.
> > > > - We included the advantage of Hessian-free methods in handling multiple deletion requests online, which previous Hessian-based works cannot achieve, in Theorem 1 (Additivity) **(L252-254)**.
> > > > - we explained why previous Hessian-based work fails to use HVP in Section 4.4 **(L370-L375)**.
> > >
> > > If the reviewer has any further **Q**uestions or **W**eakness that need to be addressed, we would be glad to provide any clarification.

---

### Author Response · Authors · 2024-11-21
**General Response**

Dear Senior Area Chairs, Area Chairs, and Reviewers,

We thank all reviewers for their constructive and valuable comments and appreciate the time spent on our manuscript. We have provided responses to the comments to fully address all reviewer concerns, denoted by **[W]** for weaknesses, **[Q]** for questions, **[L]** for lines in our manuscript, and **[R]** for references.  We made minor adjustments to the manuscript and use **[C in L]** to denote the changes in lines in our revised manuscript. We have provided our additional experimental results in an anonymous GitHub repository, and the link is provided in **L107**.

Here's a summary of our responses:

**Reviewer kjAJ:**

>- We adopt the reviewer’s suggestion to provide descriptions and references regarding complexity to avoid any potential misunderstandings in the revised version. **(C in L266, L380)**
>- We explain the analysis of generalization performance and the statement of the theorem in Section 4.2.
>- We clarify the misunderstanding regarding the analysis of optimization error in the performance analysis.
>- We adopt the reviewer’s suggestion and added the missing description in Lemma 9. **(C in L1407 of Appendix)**
>- We clarify the assumptions between previous works and our work, and provide a more detailed definitions of assumptions to avoid confusion.

**Reviewer CNQV:**

>- We correct the errors of notation in Matrix $\mathbf{M} $ based on the reviewer’s comments. **(C in L221)**
>- We clarify that our work have discussed a limitations analysis and propose corresponding improvement methods.
>- We explain the reasons why HVP with some approximation techniques are not applicable to previous studies .
>- We explain the setup of the experiments and provide additional explanations in the revised version. **(C in L1031 of Appendix)**

**Reviewer ia1D:**

>- We strive to clarify that our non-convexity claims are only applicable to unlearning methods and their associated theoretical analysis, and we explain why one of our contributions is bridging the gap in non-convex scenarios for certified unlearning methods.
>- We clarify that the optimization methods used in the learning process are unrelated to our approach and explain the connection between existing Hessian-free optimization methods and our unlearning work.
>- We incorporate the reviewer’s suggestion for experiments on more datasets and provide additional clarifications regarding our experimental contributions. **(Additional experiments provided)**
>- We clarify some suggestions from the reviewer regarding definitions and explain why we do not modify these definitions.
>- Based on the reviewer’s suggestions, we modify the typos and inaccurate statements in the paper. **(C in L156, L289, L178, and L1562 of Appendix)**

**Reviewer t2wR:**

>- We explain the assumptions of our algorithm regarding generalization error and listed it as an direction for future work.
>- We explain the reasons why HVP with some approximation techniques for previous unlearning work are not applicable.
>- We clarify the trade-off between learning rate and both learning and unlearning, incorporating the reviewer’s suggestion for trade-off experiments. **(Additional experiments provided)**

**References:**

> [R1] Remember what you want to forget: Algorithms for machine unlearning. Sekhari et al., NeurIPS 2021.
>
> [R2] Algorithms that approximate data removal: New results and limitations. Suriyakumar et al., NeurIPS 2022.
>
> [R3] Certified minimax unlearning with generalization rates and deletion capacity. Liu et al., NeurIPS 2023.
>
> [R4] Langevin Unlearning: A New Perspective of Noisy Gradient Descent for Machine Unlearning. Chien et al.,  NeurIPS 2024.
>
> [R5] Certified Machine Unlearning via Noisy Stochastic Gradient Descent. Chien et al.,  NeurIPS 2024.
>
> [R6] How SGD selects the global minima in over-parameterized learning: A dynamical stability perspective. Wu et al., NeurIPS 2018.
>
> [R7]  A loss curvature perspective on training instabilities of deep learning models. Gilmer et al., NeurIPS 2022.
>
> [R8] Fast Model Debias with Machine Unlearning. Chen et al., NeurIPS 2023.
>
> [R9] An Empirical Model of Large-Batch Training. McCandlish et al., NeurIPS 2023.
>
> [R10] A disciplined approach to neural network hyper-parameters: Part 1 -- learning rate, batch size, momentum, and weight decay. Leslie N. Smith, ArXiv 2018.
>
> [R11] Training Deep and Recurrent Networks with Hessian-Free Optimization. James Martens and Ilya Sutskever. 2012.

**Additional Revisions:**

> 1. We shortened the title to allow for more detailed descriptions in Section 1.
>
> 2. We revised the imprecise expression in Appendix C.1.

---

### Meta-Review · Area_Chair_p2KV · 2024-12-19

**Metareview:**

The paper studies the problem of certified unlearning in machine learning scenarios. It proposes a "Hessian-free" approach, removing the need for explicit Hessian computation and inversion. As a result, the paper also removes the requirement that the objective function be strongly convex, and for the unlearning part can even handle nonconvex loss functions. For the learning portion of the results, the paper builds on prior work, which requires convexity. Overall, the paper presents interesting contributions in an important and growing area of machine learning.

**Additional Comments On Reviewer Discussion:**

The rebuttal phase was quite productive, where the authors and reviewers engaged in a discussion. The authors addressed all the questions/concerns, revised the paper, and corrected some small issues identified in a subset of the reviews. As a result, all reviews converged towards acceptance.

---

### Decision · Program_Chairs · 2025-01-22

Accept (Poster)